# The potassium channel K$_{2P}$2.1 shapes the morphology and function of brain endothelial cells via actin network remodeling

Stefanie Lichtenberg [1,2,14], Laura Vinnenberg[1,14], Falk Steffen [3], Isabelle Plegge[4], Nicholas Hanuscheck[3], Vera Dobelmann[1], Joel Gruchot[1], Christina B. Schroeter [1], Haribaskar Ramachandran[5], Beatrice Wasser[3], Derya Bachir[1], Christopher Nelke[1], Jonas Franz [6,7], Christoph Riethmüller[7], Stefan Tenzer [8], Ute Distler [8], Christina Francisca Vogelaar [3], Kristina Kusche-Vihrog [9], Boris V. Skryabin[10], Timofey S. Rozhdestvensky [10], Albrecht Schwab[11], Jean Krutmann [5], Andrea Rossi [5], Thomas Budde [12], Stefan Bittner [3], Sven G. Meuth [1,14] & Tobias Ruck[1,13,14] ✉

K$_{2P}$2.1 (gene: *Kcnk2*), a two-pore-domain potassium channel, regulates leukocyte transmigration across the blood-brain barrier by a yet unknown mechanism. We demonstrate that *Kcnk2$^{-/-}$* mouse brain microvascular endothelial cells (MBMECs) exhibit an altered cytoskeletal structure and surface morphology with increased formation of membrane protrusions. Cell adhesion molecules cluster on those protrusions and facilitate leukocyte adhesion and migration in vitro and in vivo. We observe downregulation of K$_{2P}$2.1 and activation of actin modulating proteins (cofilin 1, Arp2/3) in inflamed wildtype MBMECs. In the mechanosensitive conformation, K$_{2P}$2.1 shields the phospholipid PI(4,5)P$_2$ from interaction with other actin regulatory proteins, especially cofilin 1. Consequently, after stimulus-related K$_{2P}$2.1 downregulation and dislocation from PI(4,5)P$_2$, actin rearrangements are induced. Thus, K$_{2P}$2.1-mediated regulatory processes are essential for actin dynamics, fast, reversible, and pharmacologically targetable.

The blood-brain barrier (BBB) forms a tightly regulated and highly selective interface between the blood stream and central nervous system (CNS) parenchyma. Its dysfunction and subsequent immune cell infiltration of CNS tissue is a hallmark of a wide variety of neurovascular, degenerative, infectious, and inflammatory disorders. Therefore, a deeper understanding of potential small molecule druggability of regulatory mechanisms at the BBB has profound implications for several CNS disorders. Previously, we demonstrated that K$_{2P}$2.1 (TWIK-related potassium channel-1, TREK1; encoded by the *Kcnk2* gene), a member of the two-pore domain potassium (K$_{2P}$) channel family, is critically

involved in the regulation of immune cell trafficking into the CNS[1]. K$_{2P}$2.1 is a mechanosensitive outwardly rectifying potassium channel[2,3] modulated by different stimuli such as membrane stretch, temperature, pH, lipids, polyunsaturated fatty acids and phosphorylation[4]. Interestingly, in experimental autoimmune encephalomyelitis (EAE), an animal model for multiple sclerosis (MS), K$_{2P}$2.1-deficient mice (*Kcnk2$^{-/-}$*) developed a worsened disease course with increased CNS immune cell infiltration compared to wild-type (WT) mice[1]. The observed phenotype was related to BBB endothelial cells. However, the underlying molecular mechanisms remained unclear[1].

BBB endothelial cells actively regulate immune cell extravasation by the expression of different molecules such as cell adhesion molecules (CAMs). Especially, the intercellular adhesion molecule 1 (ICAM1) and vascular cell adhesion molecule 1 (VCAM1) and their interaction with corresponding integrin ligands (lymphocyte function-associated antigen, LFA1 and very late antigen-4, VLA4, respectively) expressed on immune cells are crucial to initiate the transmigratory process[5]. To promote this interaction between immune and endothelial cells, "docking structures" formed by actin-supported membrane protrusions, presenting clusters of CAMs seem to be essential[6–9]. In addition, sensing of mechanical forces is a crucial step in the process of transendothelial immune cell migration[10–12]. Leukocytes sense the stiffness of endothelial cells by integrins on their surface and transmigrate according to the gradient of endothelial cell stiffness[10,13]. The main determinants for cellular mechanical properties including stiffness are cortical actin, defining the stiffness of the cell cortex, the intracellular actin cytoskeleton and actin stress fibers[14–17]. Therefore, cytoskeletal rearrangements especially of the actin cytoskeleton appear as a central regulatory element in transendothelial immune cell migration. For remodeling of f-actin and formation of stress fibers upon inflammation, activation of the RhoA- ROCK- actomyosin pathway is essential[18]. Actin depolymerization factors (ADF), such as cofilin 1 (Cfl1), are important for actin dynamics inside the cell and proteins of the actin-related proteins-2/3 (Arp2/3) complex are crucial for the formation of filopodia and protrusions[19–21]. Of note, previous studies suggest a potential role for $K_{2P}2.1$ in the regulation of actin cytoskeleton function and structure[22,23], yet precise mechanisms and functional downstream relevance remain superficially analyzed: the amino acid residues glutamate E306 and serine S333 of the C-terminal domain showed a regulatory function of actin dynamics[22]. Mutational studies revealed an inhibition of filopodia-like-structure formation by inhibiting stretch activation (E306A mutation) and mimicking the dephosphorylated form of $K_{2P}2.1$ (S333A)[2,3,22]. However, these studies were performed on immortalized cell lines using expression systems. In our study, we used primary mouse brain endothelial cells for characterizing the role of $K_{2P}2.1$ and its interplay with the actin cytoskeleton and actin regulatory proteins under physiological conditions. Mechanosensitivity of $K_{2P}2.1$ requires the interaction of positively charged amino acids in the C-terminal domain with phosphatidylinositol-(4,5)-bisphosphate $(PI(4,5)P_2)$[3]. In addition, interactions of the C-terminal domain with different cytoskeletal associated proteins, such as β-IV spectrin, microtubule associated protein 2 (MAP2) and filamin-A have been described[3,23–25].

In the present study, we elucidate morphological and cell mechanical changes of MBMECs mediated by $K_{2P}2.1$. We further identify key molecular mechanisms underlying $K_{2P}2.1$-mediated regulation of immune cell trafficking across the BBB. The importance of our study is supported by our finding of differential $K_{2P}2.1$. regulation under inflammatory conditions in vivo and in vitro. These findings highlight the functional relevance of $K_{2P}2.1$ for cytoskeletal stability under physiological conditions, with significant translational relevance for targeting BBB dysfunction in inflammatory neurological diseases.

## Results

### $K_{2P}2.1$ rapidly and reversibly modulates immune cell migration at the blood-brain barrier

$K_{2P}2.1$ was shown to be involved in the regulation of immune cell trafficking at the BBB[1]. However, exact mechanisms of immune cell transmigration in vivo and in vitro are not yet defined.

To elucidate whether $K_{2P}2.1$ regulation has an immediate effect on immune cell trafficking in vivo, we made use of a selective pharmacological $K_{2P}2.1$ inhibition by spadin (sortilin-derived peptide spadin; leading to $K_{2P}2.1$ internalization)[26–29] in an in vivo 2-photon microscopy approach. We imaged blood vessels in the brain stem, an area prone to inflammation in the EAE model, to follow T cell migration under pathophysiological conditions of neuroinflammation. Therefore,

MOG-specific activated CD4$^+$ T cells from B6.2D2-GFP mice were transferred to $Rag2^{-/-}cgn^{-/-}$ mice to induce passive EAE (Fig. 1a). Mice were investigated at an EAE disease score of 2. Injection of rhodamine-dextrane by a carotis catheter was used to counterstain vessel walls. T cell tracks were recorded 30 min for baseline recording, then followed by spadin injection and subsequent pharmacological ICAM1 blockade, each time for 30 min (Movie S1). The total number of migrating cells was counted and different phases of T cell migration behavior were investigated on single T cell tracks (Fig. 1b–d). We observed a significantly higher number of T cells extravasating the blood vessel directly after spadin injection (Fig. 1d, e). Injection of an ICAM1-blocking antibody reversed the spadin effect. To prove the impact of $K_{2P}2.1$ on endothelial cells of the BBB in this process, we generated a *Kcnk2* floxed mouse line (*Kcnk2$^{fl/fl}$*) and crossed these mice with endothelial cell-specific Tie2-Cre mice[30,31]. We confirmed the cell-specific knock-out by RNAscope, showing almost no residual expression of *Kcnk2* in CD31$^+$ endothelial cells and no impact on the expression on CD31$^-$ cells (Fig. S1a, b). Mice with an endothelial cell-specific knock-out of $K_{2P}2.1$ (*Kcnk2$^{fl/fl}$xTie2$^{cre+}$*) developed a worse EAE disease course compared to control mice (Fig. 1f, g). Consistent with the observations made on *Rag2$^{-/-}$* mice, spadin treatment of control mice (*Kcnk2$^{fl/fl}$xTie2$^{cre-}$*) also led to a worse EAE course. However, spadin demonstrated no additional effect on the disease progression in endothelial cell-specific knock-out mice indicating no major off-target effects. Therefore, EAE induction in *Kcnk2$^{fl/fl}$xTie2$^{cre+}$* and respective control mice with and without spadin treatment underlined the regulatory function of $K_{2P}2.1$ on brain endothelial cells.

### $K_{2P}2.1$ expression is regulated under inflammatory conditions and enhances immune cell adhesion in vitro

Next, we aimed to gain deeper insight into the regulatory mechanisms of $K_{2P}2.1$ on endothelial cells of the BBB. Previously, we have shown that $K_{2P}2.1$ is downregulated in mouse brain microvascular endothelial cells (MBMECs) under inflammatory conditions in vitro and in vivo[1]. However, the dynamics of $K_{2P}2.1$ regulation and the intracellular consequences are still unknown. Therefore, we analyzed mRNA and protein expression levels of $K_{2P}2.1$ in MBMECs after different time points of inflammation induced with tumor necrosis factor α (TNFα) and interferon γ (IFNγ). We observed a time-dependent downregulation of $K_{2P}2.1$ mRNA and protein levels (Fig. 2a, b). Lowest channel expression levels were detected at 24 h of inflammation. However, significant downregulation of *Kcnk2* mRNA was already observed after 30 min. Substitution of the medium of inflamed cells (24 h TNFα/IFNγ treatment) back to control conditions led to an upregulation of *Kcnk2*, fully restoring *Kcnk2* expression levels 6 h after wash-out of TNFα and IFNγ. To confirm that $K_{2P}2.1$ also has an impact on immune cell migration in vitro, we seeded primary MBMECs from WT and *Kcnk2$^{-/-}$* mice into flow chambers and treated the cells for 24 h with TNFα and IFNγ. We used stimulated WT CD4$^+$ T cells for all in vitro assays as $K_{2P}2.1$ is not expressed on T cells[1]. The analysis of immune cell migration was performed under low physiological flow conditions (0.25 dyn/cm$^2$) in the T cell – MBMEC co-culture (Fig. 2c). No difference in numbers of adhering T cells to untreated *Kcnk2$^{-/-}$* and WT MBMECs was observed after 30 min of recording. However, significantly increased numbers of T cells adhered to inflamed *Kcnk2$^{-/-}$* compared to WT MBMECs (Fig. 2d). This shows that, beside blocking of $K_{2P}2.1$ pharmacologically with spadin in vivo, also a genetic deletion of *Kcnk2* promotes the interaction of T cells with inflamed endothelial cells in vitro.

### Loss of $K_{2P}2.1$ leads to the formation of membrane protrusions with clusters of ICAM1

As shown before, blocking the interaction between immune cells and BBB endothelial cells by administering an ICAM1 blocking antibody reduces the number of transmigrating immune cells upon spadin

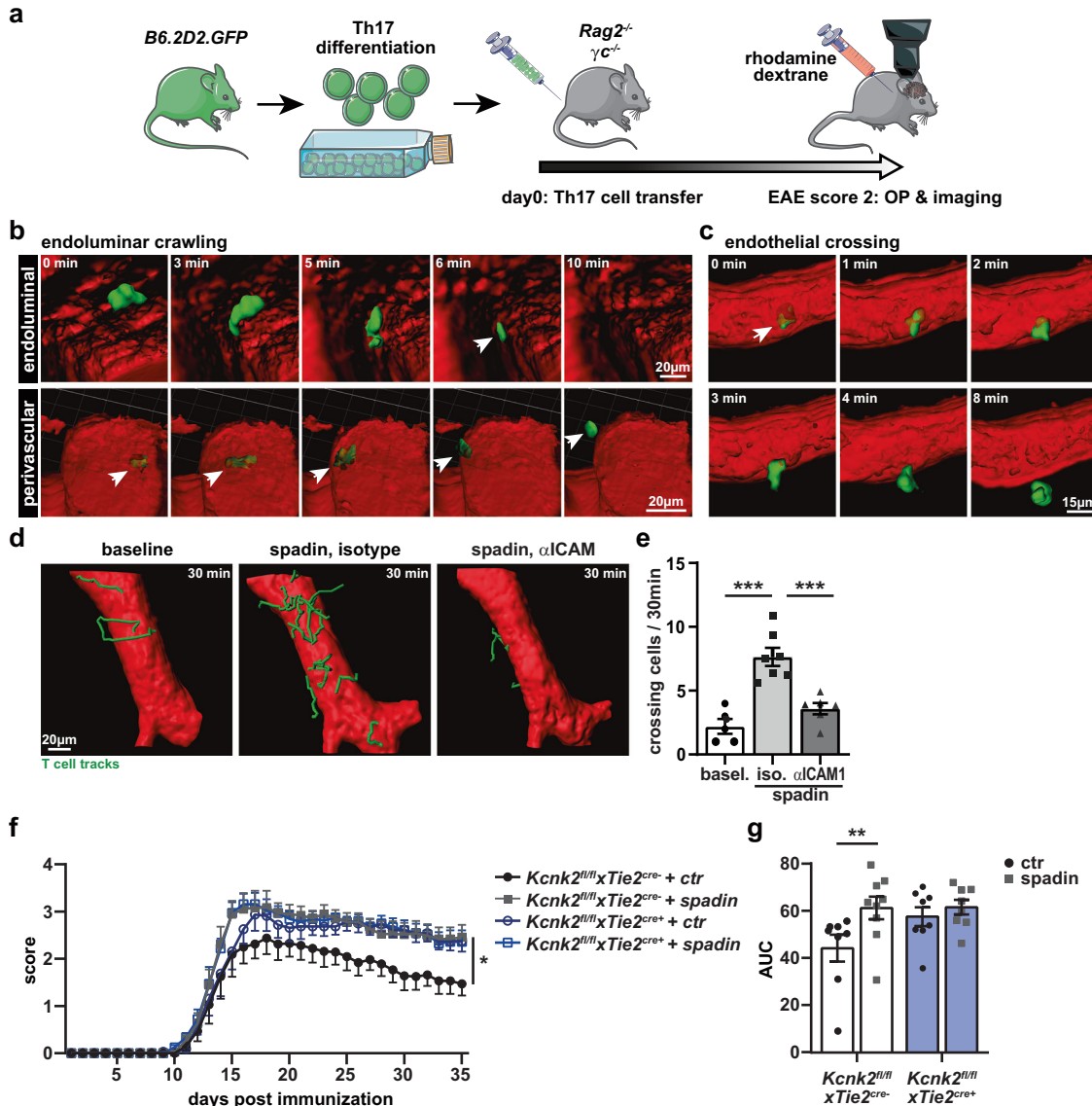

**Fig. 1 | K$_{2P}$2.1 modulation influences T cell migration in the context of neuroinflammation in vivo. a** Scheme of the experimental setup. Adoptive EAE was induced by injecting GFP$^+$ Th17 cells (green syringe) intravenously into *Rag2$^{-/-}$cgn$^{-/-}$* mice. At disease score of 2, mice were anesthetized, equipped with a carotid artery catheter and prepared for intravital two-photon microscope imaging of the brainstem. Blood brain vessels were visualized by rhodamine-labeled dextran (red syringe). K$_{2P}$2.1 inhibition was performed using spadin. **b** 3D-reconstructions with blood-vessels (red), CD4$^+$ T cells (green) were calculated using two-photon imaging derived XYZ-stacks. Representative endoluminal crawling sequence (10 min) depicted from the two perspectives of endoluminal (upper panel) and perivascular (lower panel), respectively. **c** Representative sequence showing a CD4$^+$ T cell (green) extravasating the blood vessel (lumen, red) shown from an extravascular sight. **d** The same blood vessel (red) was recorded for 30 min before (left), after

spadin treatment (center) and after spadin treatment and blocking ICAM1 (right) with tracks from CD4$^+$ T cells extravasating the vessel (green). **e** Mean number of CD4$^+$ T cells extravasating the blood vessel in baseline (basel., dots) recordings ($N = 5$) and after spadin and either isotype antibody (cubes, $N = 7$) or anti-ICAM1 injection ($N = 6$). **f** EAE disease course of endothelial cell-specific *Kcnk2$^{-/-}$* (*Kcnk2$^{fl/fl}$xTie2$^{cre+}$*, blue) and control (*Kcnk2$^{fl/fl}$xTie2$^{cre-}$*, white) mice with (gray cubes) or without (black dots) spadin treatment ($N = 8$). **g** Analysis of EAE disease courses; Area under the curve (AUC) is shown for the different groups ($N = 8$). Data are presented as mean ± SEM. N representing the number of individual mice. Statistical analysis using (**e**) 1-way ANOVA + Bonferroni correction and (**f, g**) 2-way ANOVA + Bonferroni correction with *$p < 0.05$ and ***$p < 0.001$. Exact $p$-values are listed in the Source Data file.

treatment (Fig. 1b-e). Along with an increased T cell adhesion under inflammatory conditions in vitro, we performed staining for ICAM1 in MBMECs, to identify the impact of ICAM1 upon knock-out or down-regulation of K$_{2P}$2.1 (Fig. 3a). We observed an increased number of actin-based protrusions with ICAM1 clusters on inflamed MBMECs (Fig. 3a, b). Of note, *Kcnk2$^{-/-}$* MBMECs showed these ICAM1 clusters already under untreated conditions (Fig. 3b). To characterize and quantify these protrusions, we analyzed the 3D surface topology of MBMECs by atomic force microscopy (AFM). Structures starting around 120 nm above the cell surface were considered as membrane

protrusions; they were identified using a combination of local geometry measures and automated classification with a trained artificial neural network as previously described[9]. We found a significant increase in membrane protrusion count and volume upon inflammation of WT and *Kcnk2$^{-/-}$* MBMECs, compared to untreated WT MBMECs (Fig. 3c, d). Interestingly, protrusion numbers and volume in untreated *Kcnk2$^{-/-}$* MBMECs were again higher than in WT MBMECs.

Strong cell-cell contacts are required for the initiation of transmigration. The observed protrusions, presenting ICAM1, could serve as "immune cell docking structures" and might be able to induce these

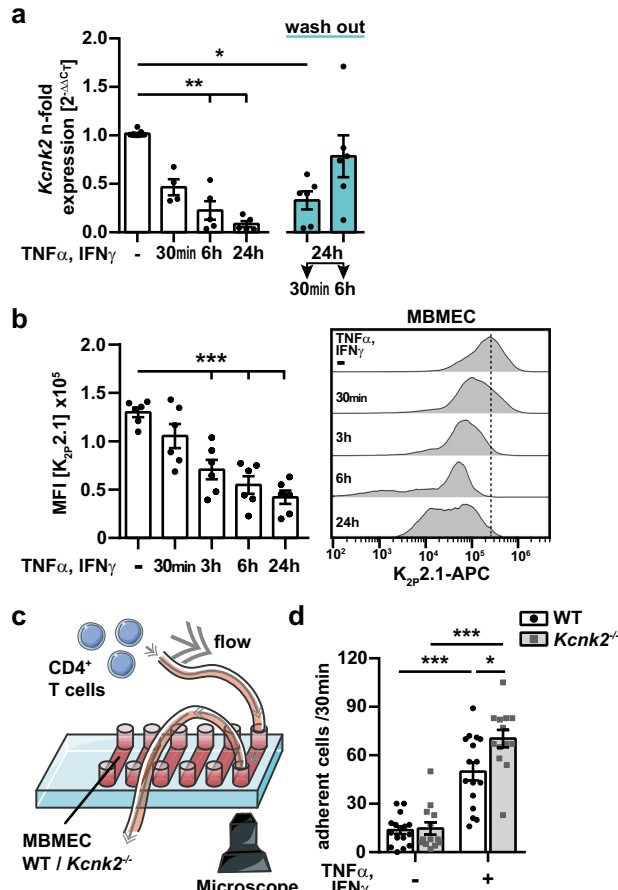

**Fig. 2 | K$_{2P}$2.1 expression pattern and cell adhesion in vitro. a** mRNA expression levels of *Kcnk2* in WT MBMECs with or without TNFα and IFNγ stimulation (500 U/ml, each, white) and after wash out of TNFα/IFNγ (blue). Expression levels were normalized to untreated cells. Statistics were calculated with ΔC$_T$ values (*N* = 4–6). **b** Mean fluorescence intensities (MFI, left) and representative histograms (right) of K$_{2P}$2.1 in WT MBMECs with or without TNFα/IFNγ stimulation (500 U/ml, each) determined by flow cytometry (*N* = 6). **c** Experimental setup for flow conditions. MBMECs were seeded into μ-slides (ibidi®); stimulated WT CD4$^+$ T cells were applied under low physiological flow (0.25 dyn/cm²). **d** Total number of T cells adhering to WT (white, black dots) and *Kcnk2*$^{-/-}$ (gray, gray cubes) endothelial cells within 30 min of acquisition (*N* = 12–16). N representing the number of individual MBMEC preparations. Exact N-numbers for each condition are listed in the Source Data file. Data are presented as mean ± SEM. Statistical analysis using (**a**, **b**) Kruskal-Wallis test + Dunn's multiple comparison correction and (**d**) 1-way ANOVA + Bonferroni correction with *$p$ < 0.05, **$p$ < 0.01 and ***$p$ < 0.001. Exact $p$-values are listed in the Source Data file.

intense immune cell-endothelial cell interactions[6–9]. Therefore, we investigated whether K$_{2P}$2.1 regulation influences the direct cell-cell adhesion of CD4$^+$ T cells to the MBMEC monolayer employing AFM-based single-cell force spectroscopy. A stimulated CD4$^+$ T cell was firmly attached to the tip of the AFM cantilever, then attached to (2 s contact time) and thereafter detached (1.5 nN force, 10 μm/s retraction speed) from the MBMEC monolayer. The force required to detach the immune cell was measured as adhesion force and used to calculate adhesion energy. Adhesion force and energy were significantly increased after inflammation of both, WT and *Kcnk2*$^{-/-}$ MBMECs. Inflamed *Kcnk2*$^{-/-}$ MBMECs demonstrated an even more pronounced increase in adhesion than inflamed WT MBMECs. Moreover, K$_{2P}$2.1 deletion led to significantly stronger cell-cell contacts already under untreated conditions (Fig. 3e). These results align with the effect observed after K$_{2P}$2.1 deletion, indicating that associated surface changes of MBMECs enhance T cell adhesion and migration in vitro.

ICAM1 is concentrated on finger-like membrane protrusions under inflammation and upon knock-out of K$_{2P}$2.1. Taken together, cellular protrusions formed by the actin cytoskeleton in inflamed MBMECs showed ICAM1 clusters on their surface representing ideal structures potentially facilitating immune cell transmigration into the CNS.

## Proteome profiling reveals changes in actin network proteins and cytoskeletal organization in *Kcnk2*$^{-/-}$ MBMECs

To elucidate the underlying molecular mechanisms upon knock-out of K$_{2P}$2.1, we analyzed the proteome of *Kcnk2*$^{-/-}$ and WT MBMECs. In total 273 differentially regulated proteins were detected (Fig. 4a). Network analysis revealed changes in different pathways related to cytoskeletal regulation and actin filament organization, including actin-based membrane projection, but also metabolic processes and organization of the extracellular matrix in *Kcnk2*$^{-/-}$ MBMECs (Fig. 4b). Analysis of interactions and networks of those differentially expressed proteins, revealed interconnections to the Rho/ROCK pathway and different actin depolymerizing factors. We also searched for proteins that are already linked to the function of ion channels. Some of the regulated proteins were dependent on ion channel functions, however, there was no clear connection to a specific channel or channel type (Fig. 4c). To validate these proteomic data hits, we first characterized the morphology of the actin cytoskeleton by AFM and immunofluorescence staining. We investigated and defined the intracellular structural and molecular consequences of K$_{2P}$2.1 knock-out and inflammation induced K$_{2P}$2.1 regulation on the actin cytoskeleton. We observed increased actin stress fiber formation in WT MBMECs upon inflammation, shown by an increased fibrosity index, calculated from AFM images (Fig. 4d, e). In *Kcnk2*$^{-/-}$ MBMECs stress fiber formation was already evident under untreated conditions and was not further increased upon inflammation. Extended characterization of the time-dependency of stress fiber formation revealed a significant increase in fiber thickness and fiber length in untreated *Kcnk2*$^{-/-}$ MBMECs in comparison to WT MBMECs, which was similar to inflamed WT MBMECs after 3 h and 6 h (Fig. S2a-c). A further gain in fiber thickness was observed in both, WT and *Kcnk2*$^{-/-}$ MBMECs after 24 h of inflammation. The length of actin stress fibers increased in both, WT and *Kcnk2*$^{-/-}$ cells over the time of inflammation; however, fibers were significantly longer in *Kcnk2*$^{-/-}$ compared to WT MBMECs after 6 h of TNFα / IFNγ treatment (Fig. S2c). As we found these changes in the intracellular f-actin cytoskeleton towards an increased stress fiber formation, we asked whether also the cortical actin rim, determining the cortical stiffness, would be altered[32,33]. Inflammation induced an increase in cortical stiffness in both WT and *Kcnk2*$^{-/-}$ MBMECs. However, *Kcnk2*$^{-/-}$ cells showed an increased stiffness of cortical actin already under untreated conditions (Fig. 4f). There was no further increase in cortical stiffness in inflamed *Kcnk2*$^{-/-}$ MBMECs in comparison to untreated *Kcnk2*$^{-/-}$ cells.

Thus, a decrease in K$_{2P}$2.1 expression is strongly associated with the regulation of actin cytoskeleton structure, cortical actin rim and intracellular stress fiber formation.

## K$_{2P}$2.1 deficiency is associated with altered expression of actin regulating factors

Considering the proteome profile and alterations in actin organization observed during inflammation or upon loss of K$_{2P}$2.1, we sought to identify potential actin regulators responsible for the above-mentioned changes. To investigate how K$_{2P}$2.1 downregulation is linked to changes in the structure and regulation of actin, we used a quantitative real-time PCR (qRT-PCR) array detecting transcripts of cytoskeletal regulators in untreated and inflamed WT and *Kcnk2*$^{-/-}$ MBMECs. Analysis of differentially regulated genes between untreated WT and *Kcnk2*$^{-/-}$ MBMECs revealed eleven (out of 84 tested, Table S1) significantly altered transcripts in untreated *Kcnk2*$^{-/-}$ cells. The actin depolymerizing factor cofilin 1 (Cfl1) was identified as promising

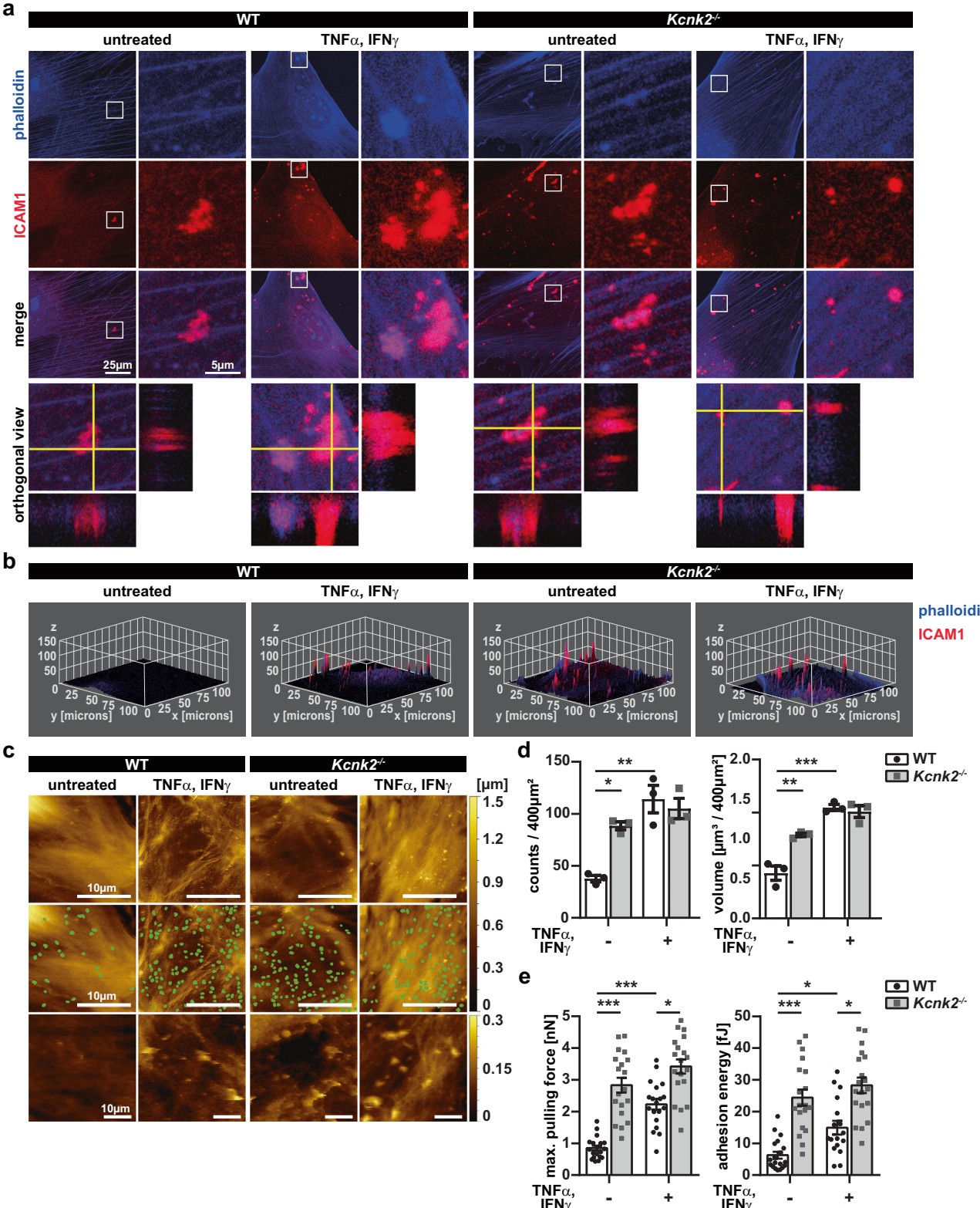

candidate, since it was significantly differentially regulated in untreated *Kcnk2*[-/-] MBMECs, as well as in inflamed conditions of both *Kcnk2*[-/-] and WT MBMECS (Fig. 5a). In WT MBMECs, *Cfl1* was significantly upregulated on transcriptional level under inflammatory conditions, whereas comparable expression levels were already found under untreated conditions in *Kcnk2*[-/-] cells (Fig. 5b). Of note, *Limk1* (LIM domain kinase 1; LIMK-1) and *Ssh1* (slingshot protein phosphatase

1, SSH-1), both regulators of Cfl1 activity, expression levels also demonstrated time-dependent fluctuations in both WT and *Kcnk2*[-/-] MBMECs (Fig. 5c, d). Cfl1 is a key regulator of actin dynamics at the leading edge of motile cells and thus of the formation of cellular protrusions[19,34]. The active form of Cfl1 severs new actin barbed ends that can either be depolymerized or used for further polymerization and growing of actin filaments[19,20,34,35]. Cfl1 acts synergistically with

**Fig. 3 | *Kcnk2$^{-/-}$* MBMECs show significantly altered morphology with pro-adhesive protrusions. a** Immunofluorescence staining of untreated and inflamed WT and *Kcnk2$^{-/-}$* MBMECs for f-actin (phalloidin, blue) and ICAM1 (red). White boxes in left panels represent the magnified area in the right panels. Orthogonal view is shown from Z-Stack images. Scale bars represent 25 μm and 5 μm. **b** 3D surface plot of images (ImageJ) shown in (**a**). Z-stacks (30 images) were used for analysis of the 3D profile of the cells, 5% signal intensity was set as minimum, 60% as maximum for projection. Smoothing was applied for better visualization. ICAM1 signals are shown in red, phalloidin in blue. **c** AFM images of WT (left two panels) and *Kcnk2$^{-/-}$* (right two panels) MBMECs. First row shows raw data images used for automated protrusion detection, which are highlighted as green spots in the second row. Z-profile indicated on the upper right panel ranges from 0 μm to 1.5 μm. Third row represents protrusions scaled to the cell surface. Z-profile ranges from 0 nm to 300 nm. Scale bars represent 10 μm. **d** Protrusion counts (left; $N = 3$)

and volume (right; $N = 3$) of AFM images shown in (**c**) of WT (white, black dots) and *Kcnk2$^{-/-}$* (gray, gray cubes) MBMECS. N representing the number of individual MBMEC preparations. **e** AFM-based single-cell force spectroscopy of stimulated CD4$^+$ T cell approached towards the indicated MBMEC monolayer. Contact time 2 s, constant force of 1.5 nN, retraction speed 10 μm/s. Maximum pulling force (left) and adhesion energy (right) in T cell – MBMEC (WT (white, black dots) and *Kcnk2$^{-/-}$* (gray, gray cubes)) adhesion measurements using CellHesion$^*$ ($N = 4-8$, $n = 17-19$; each data point represents the mean value of at least 20 force-distance curves measured with one T cell on different MBMECs.). N and n representing the number of individual MBMEC preparations and individual T cells, respectively. Exact N and n-numbers for each condition are listed in the Source Data file. All data are shown as mean +/- SEM. Statistical analysis using (**d**) 1-way ANOVA + Bonferroni correction and (**e**) Kruskal-Wallis test + Dunn's multiple comparison correction with *$p < 0.05$, **$p < 0.01$ and ***$p < 0.001$. Exact p-values are listed in the Source Data file.

Arp2/3 to amplify local actin polymerization via de novo nucleation of filaments upon cellular stimulation[34]. Interestingly, the expression of several Arp2/3 complex components (*Arpc1b*, *Arpc2*, *Arpc4* and *Arpc5*) was also significantly increased in *Kcnk2$^{-/-}$* MBMECs (Fig. S3a). As we could not detect significant alterations in the protein amount of Cfl1 in *Kcnk2$^{-/-}$* MBMECs compared to WT in the proteome analysis, we next assessed the phosphorylation state of Cfl1 that regulates its protein activity. The phosphorylated form of Cfl1 is inactive, whereas dephosphorylation by the slingshot protein phosphatase 1 (SSH-1) leads to its activation[36,37]. Therefore, we determined the ratio between dephosphorylated (active) and phosphorylated (inactive) Cfl1 by immunofluorescence staining (Fig. 5e, f). *Kcnk2$^{-/-}$* MBMECs showed significantly higher levels of active Cfl1 (higher Cfl1/pCfl1 ratio) compared to WT cells already under untreated conditions. Inflammatory stimuli further increased active Cfl1 levels in *Kcnk2$^{-/-}$* MBMECs reaching a maximum after 3 h. However, in WT cells the response was delayed and not observed prior to 6 h of inflammation (Fig. 5e, f). In vivo Cfl1 was elevated in the endothelial-specific K$_{2P}$2.1 knock-out brain slices in EAE (Fig. S3b). Taken together, the mRNA and Cfl1/pCfl1 profiles in *Kcnk2$^{-/-}$* MBMECS resemble those of inflamed MBMECs, supporting a regulatory role of K$_{2P}$2.1 for cellular actin networks in brain endothelial cells.

## K$_{2P}$2.1 regulates Cfl1 via PI(4,5)P$_2$

Both, K$_{2P}$2.1 and ADFs are capable of interacting with and are regulated by the phospholipid PI(4,5)P$_2$, thereby representing a potential link[2,38]. Consequently, we hypothesized that blockade of PI(4,5)P$_2$ activation could rescue the phenotype induced by loss of K$_{2P}$2.1. Therefore, we used phenylarsine-oxide (PAO) to reduce PI(4,5)P$_2$ activation by inhibition of PI4 kinase[39–41]. Indeed, inhibition of PI(4,5)P$_2$ activation by PAO in untreated *Kcnk2$^{-/-}$* MBMECs led to a restoration of physiological actin cytoskeletal organization and a loss of stress fibers (Fig. 6a). Consistent with this observation, stress fibers in inflamed WT and *Kcnk2$^{-/-}$* cells were disrupted by treatment with PAO. Untreated *Kcnk2$^{-/-}$* MBMECs as well as inflamed MBMECs from WT and *Kcnk2$^{-/-}$* cells demonstrated a decrease in actin fiber diameter and fiber length by inhibition of PI(4,5)P$_2$ (Fig. 6b, c). PI(4,5)P$_2$ is able to directly interact with Cfl1 thereby regulating actin depolymerization[42–44]. To further assess whether K$_{2P}$2.1 has a direct impact on this interaction, we performed a proximity ligation assay (PLA) by staining for PI(4,5)P$_2$ and Cfl1 in inflamed and uninflamed WT and *Kcnk2$^{-/-}$* MBMECs. PLA allows to investigate protein-protein interactions without directly coupling fluorescent proteins or dyes to the target protein[45]. Under inflammatory stimuli, we observed enhanced interactions of PI(4,5)P$_2$ and Cfl1 in WT MBMECs. In contrast, in *Kcnk2$^{-/-}$* MBMECs, we detected increased interactions of PI(4,5)P$_2$ and Cfl1 already under control conditions, which further increased over time (Fig. 6d, e). In line with these results, the interaction of PI(4,5)P$_2$ with K$_{2P}$2.1 in WT MBMECs during inflammation (Fig. 6f, g). These results show a competitive interaction of PI(4,5)P$_2$ with either K$_{2P}$2.1 or Cfl1.

Therefore, K$_{2P}$2.1 seems to be critical for the spatiotemporal regulation of membrane protrusion formation determined by fine-tuning of actin turnover.

## K$_{2P}$2.1 regulates immune cell adhesion on brain microvascular endothelial cells by alterations of actin rearrangements

Consequently, K$_{2P}$2.1 might regulate immune cell trafficking at the BBB via alteration of actin cytoskeleton dynamics resulting in the formation of "immune cell docking structures" facilitating leukocyte adhesion and migration.

To investigate the link between K$_{2P}$2.1 and actin regulation via Cfl1 and its consequences for T cell adhesion, we used siRNA to knock-down *Cfl1* in WT and *Kcnk2$^{-/-}$* MBEMCs. As knock-down strategy we used the *Cfl1* siRNA SMARTpool. Control cells were transfected with a non-target control (NTC) siRNA mix. We hypothesized that preventing the upregulation of Cfl1 by the knock-down or inflammation-mediated downregulation of K$_{2P}$2.1 should reduce the number of adherent T cells. We validated the knock-down efficiency and expression levels of Cfl1 by immunofluorescence staining and quantitative RT-PCR (Fig. 7a, b). As a functional readout of the K$_{2P}$2.1-dependent regulation of Cfl1, we performed cell adhesion assays under low physiological flow (0.25 dyn/cm$^2$). Similar to the previous results, an increased number of WT T cells were attached to inflamed MBMECs from *Kcnk2$^{-/-}$* mice, compared to WT (Fig. 7c). By the knock-down of *Cfl1* we were able to diminish the amount of adherent T cells to both inflamed MBMEC layers from WT and *Kcnk2$^{-/-}$* mice. These results demonstrate that Cfl1 is critically involved in the actin rearrangements, stress fiber formation and formation of membrane protrusions upon knock-out and inflammation-mediated downregulation of K$_{2P}$2.1 on endothelial cells of the BBB.

## Discussion

Endothelial cells (EC) integrate various signals from neighboring tissues and the blood stream. Detrimental processes such as tissue infection or neoplasia need a prompt response by transmigrating immune cells induced by ECs. Here, we show that K$_{2P}$2.1 regulates immune cell trafficking across the BBB by rapid and dynamical rearrangements of the endothelial f-actin cytoskeleton. Inflammatory stimuli lead to fast K$_{2P}$2.1 downregulation and detachment of K$_{2P}$2.1 from PI(4,5)P$_2$. These processes are associated with downstream activation of PI(4,5)P$_2$-dependent actin modulatory proteins (Cfl1 and Arp2/3) and alteration of connected pathways related to cytoskeletal regulation, cell membrane and extracellular matrix organization. A knock-out of K$_{2P}$2.1 is already sufficient to set ECs into an "inflammation-like" morphology. The expression pattern of cytoskeletal regulatory proteins, especially Cfl1, is similar to inflamed endothelial cells. Ultrastructurally, K$_{2P}$2.1 deficiency leads to fiber and membrane protrusion formation and stiffening of cortical actin. The protrusions contain clusters of ICAM1 forming specialized structures for leukocyte adhesion, facilitating transendothelial immune cell trafficking in vitro and in vivo. Thus, these

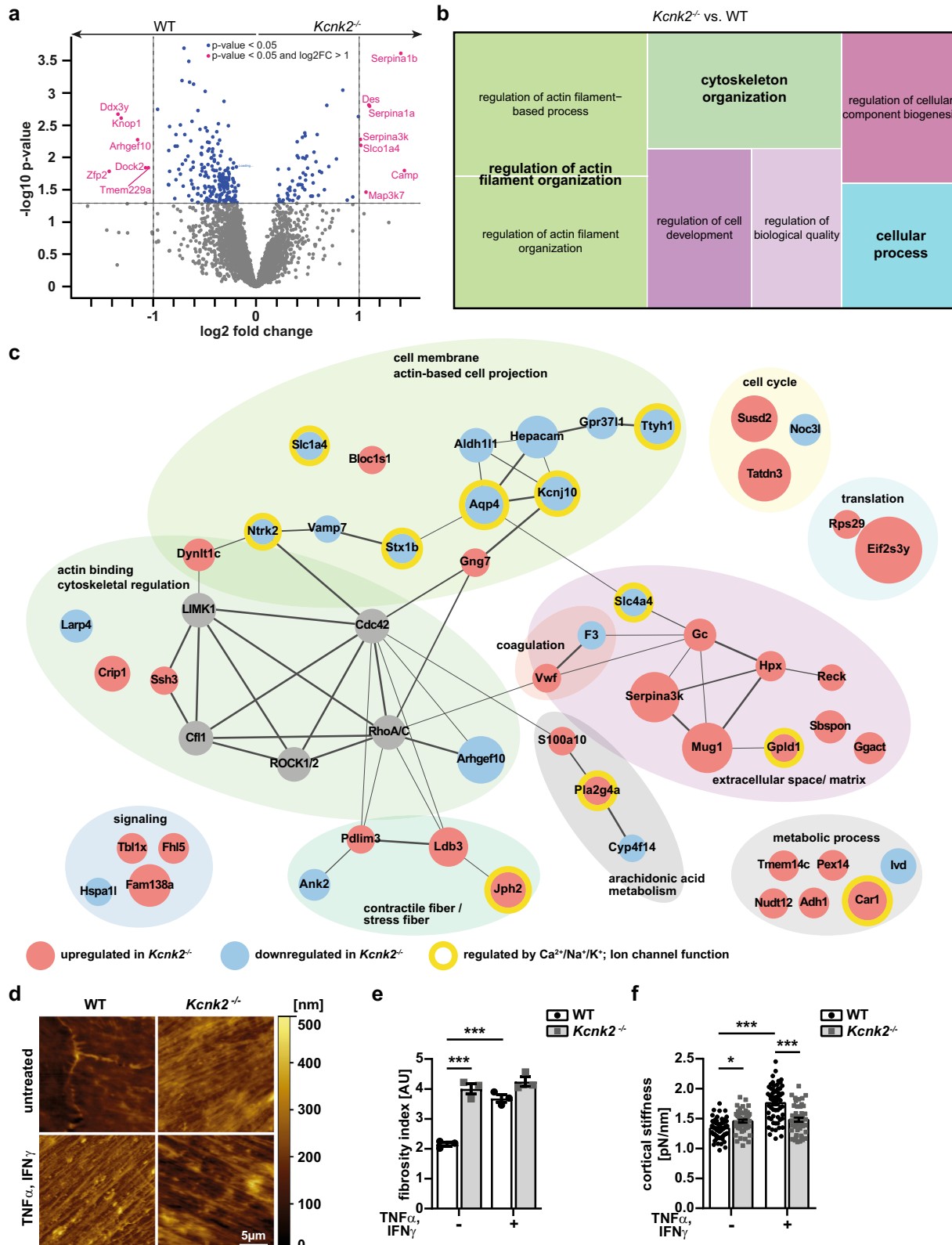

results demonstrate a central role of K$_{2P}$2.1 in cellular signaling pathways following an inflammatory stimulus.

Earlier studies have demonstrated a decisive role of the K$_{2P}$2.1 C-terminal domain for mechano-gating[3,22,46–48]. In these studies constitutively active K$_{2P}$2.1 led to membrane smoothing; however, the underlying molecular mechanisms remained unclear[22]. Correspondingly, loss of K$_{2P}$2.1 function by genetic deletion led to opposite effects in our

experiments. *Kcnk2*$^{-/-}$ MBMECs demonstrated most characteristics of inflamed MBMECs under untreated conditions, including an increase in stress fiber formation, pronounced membrane protrusion growth and stiffening of cortical actin.

Remodeling of cytoskeletal actin is regulated by complex mechanisms involving several regulatory proteins with specific spatiotemporal distribution. The dynamic nucleation zone at lamellipodia

**Fig. 4 | *Kcnk2^{-/-}* influences protein expression levels involved in cytoskeletal regulation and cell morphology. a** Volcano Plot of differentially regulated proteins of *Kcnk2^{-/-}* versus WT MBMECs, blue = $p < 0.05$, magenta = $p < 0.05$ and log2 fold change (FC) > 2 ($N = 5$). **b** Tree map of enriched GO-Terms of differentially expressed proteins in naïve *Kcnk2^{-/-}* MBMECs versus WT MBMECs. GO-Terms were summarized by REVIGO analysis[82]. **c** Network analysis of differentially regulated pathways, proteins and respective interaction partners in naïve *Kcnk2^{-/-}* MBMECs versus WT MBMECs. Red circles indicate upregulation, blue circles downregulation in *Kcnk2^{-/-}* MBMECs. Yellow borders indicate an ion-dependency of respective proteins. Interaction of proteins are shown according to STRING analysis[78]. Biological processes are annotated within the respective circles. **d** Representative AFM (atomic force microscopy) images of WT and *Kcnk2^{-/-}* MBMECs. Color code depicts

height of observed structures (0–500 nm), scale bars represent 5 μm. **e** Calculations of the fibrosity index from AFM images in (**d**) of WT (white, black dots) and *Kcnk2^{-/-}* (gray, gray cubes) MBMECs ($N = 3$). **f** AFM-based measurements of cortical stiffness of WT (white, black dots) and *Kcnk2^{-/-}* (gray, gray cubes) MBMECs ($N = 4$, $n = 45$–59; each data point represents the mean value of 6 force-distance curves of one MBMEC cell). N and n representing the number of individual MBMEC preparations and individual MBMEC cells, respectively. Exact N and n-numbers for each condition are listed in the Source Data file. All data are shown as mean +/- SEM. Statistical analysis using (**a**) two-tailed student's *t*-test + Benjamini-Hochberg correction for multiple testing and (**e**, **f**) 1-way ANOVA + Bonferroni correction with *$p < 0.05$ and ***$p < 0.001$. Exact *p*-values are listed in the Source Data file.

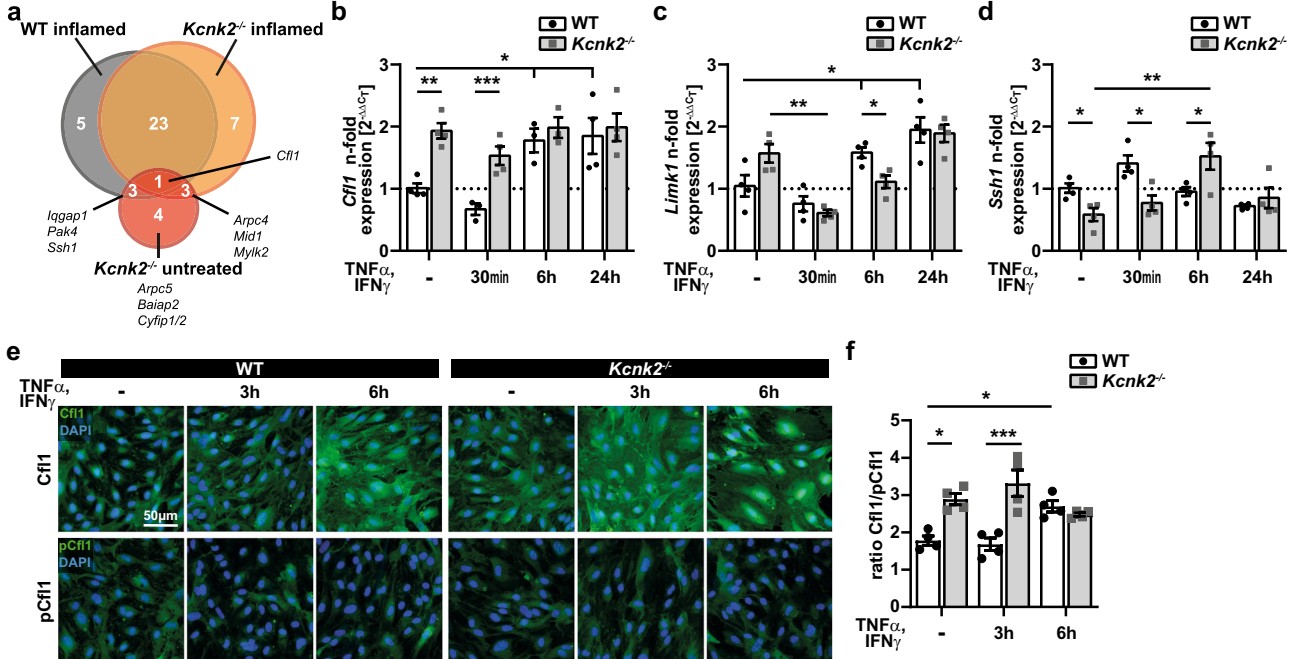

**Fig. 5 | K$_{2P}$2.1 depletion leads to altered expression of cytoskeleton regulators. a** Venn Diagram of significantly ($p < 0.05$) differentially regulated genes of inflamed WT (gray), untreated *Kcnk2^{-/-}* (red) and inflamed *Kcnk2^{-/-}* (orange) MBMEC (to untreated WT MBMECs). Numbers of overlapping transcripts are shown within the respective circles. The actin depolymerizing factor cofilin 1 (Cfl1) is differentially expressed in inflamed WT, as well as untreated and inflamed *Kcnk2^{-/-}* MBMECs ($N = 4$). **b–d** qRT-PCR data of (**b**) *Cofilin1* (*Cfl1*), (**c**) *LIM domain kinase 1* (*Limk1*) and (**d**) *Slingshot homolog 1* (*Ssh1*) expression in WT (white, black dots) and *Kcnk2^{-/-}* (gray, gray cubes) MBMECs upon inflammation (TNFα/IFNγ). Statistics were calculated with Δ$C_T$ values ($N = 3$–4). **e** Representative immunofluorescence staining

of untreated and inflamed (TNFa/IFNγ for 3 h and 6 h) WT and *Kcnk2^{-/-}* MBMECs for Cfl1 (green) or phosphorylated Cfl1 (pCfl1, green). DAPI is shown in blue. Scale bar represents 50 μm. **f** Quantification of Cfl1 and pCfl1 in untreated and TNFα/IFNγ treated WT and *Kcnk2^{-/-}* MBMECs by immunofluorescence staining shown in (**e**). Normalized data (to DAPI) was used to calculate ratios between Cfl1 and pCfl1 ($N = 4$). N representing the number of individual MBMEC preparations. Exact N-numbers for each condition are listed in the Source Data file. All data are shown as mean +/- SEM. Statistical analysis using (**a–d**, **f**) 1-way ANOVA + Bonferroni correction with *$p < 0.05$, **$p < 0.01$ and ***$p < 0.001$. Exact *p*-values are listed in the Source Data file.

and protrusions is enriched with Cfl1 and Arp2/3[19,20]. We demonstrate that downregulation upon inflammation or genetic deletion of *Kcnk2^{-/-}* is associated with a shift to active Cfl1 and Arp2/3 and protrusion formation at the plasma membrane as well as cortical actin rearrangements. This allows the establishment of functionally distinct actin compartments inside the cell with rapid generation of actin barbed ends in the Cfl1- and Arp2/3-rich compartment and almost no barbed end formation in the cell body[34].

An essential regulator of Cfl1- and Arp2/3-activity is the phospholipid PI(4,5)P$_2$[38], which is also crucial for the mechanosensitivity of K$_{2P}$2.1[3]. PI(4,5)P$_2$ activation via first PI4-kinase (PI4K) and second PI3-kinase (PI3K) is induced by various internal and external stimuli, e.g., cell stress, cell-cell interactions or inflammatory processes[49–51]. A high density of PI(4,5)P$_2$ was observed in close vicinity to membrane protrusions or sites of phagocytosis[49]. Of note, blockade of PI(4,5)P$_2$ formation by

PI4K inhibition reverted the *Kcnk2^{-/-}* and inflamed WT MBMEC phenotype, thereby demonstrating a central role of PI(4,5)P$_2$ in actin rearrangements induced by K$_{2P}$2.1 downregulation and detachment from PI(4,5)P$_2$. PI(4,5)P$_2$ then binds and thereby controls Cfl1 activation[42–44,52]. We demonstrate enhanced direct interactions of PI(4,5)P$_2$ and Cfl1 upon inflammation-mediated downregulation or knock-out of K$_{2P}$2.1 by proximity ligation assays. Binding of Cfl1 to PI(4,5)P$_2$ under these conditions leads to inactivation of Cfl1 and thereby stabilization of actin filaments with formation of actin stress fibers. Our proteomics data support a central role of K$_{2P}$2.1-dependent cytoskeleton regulation via Cfl1 as we found predominant alterations of directly and indirectly Cfl1-connected pathways related to cytoskeletal regulation, cell membrane and extracellular matrix organization. Of note, some of the regulated proteins are known to be modulated by altered intracellular ion concentrations, predominantly sodium and calcium rather than potassium

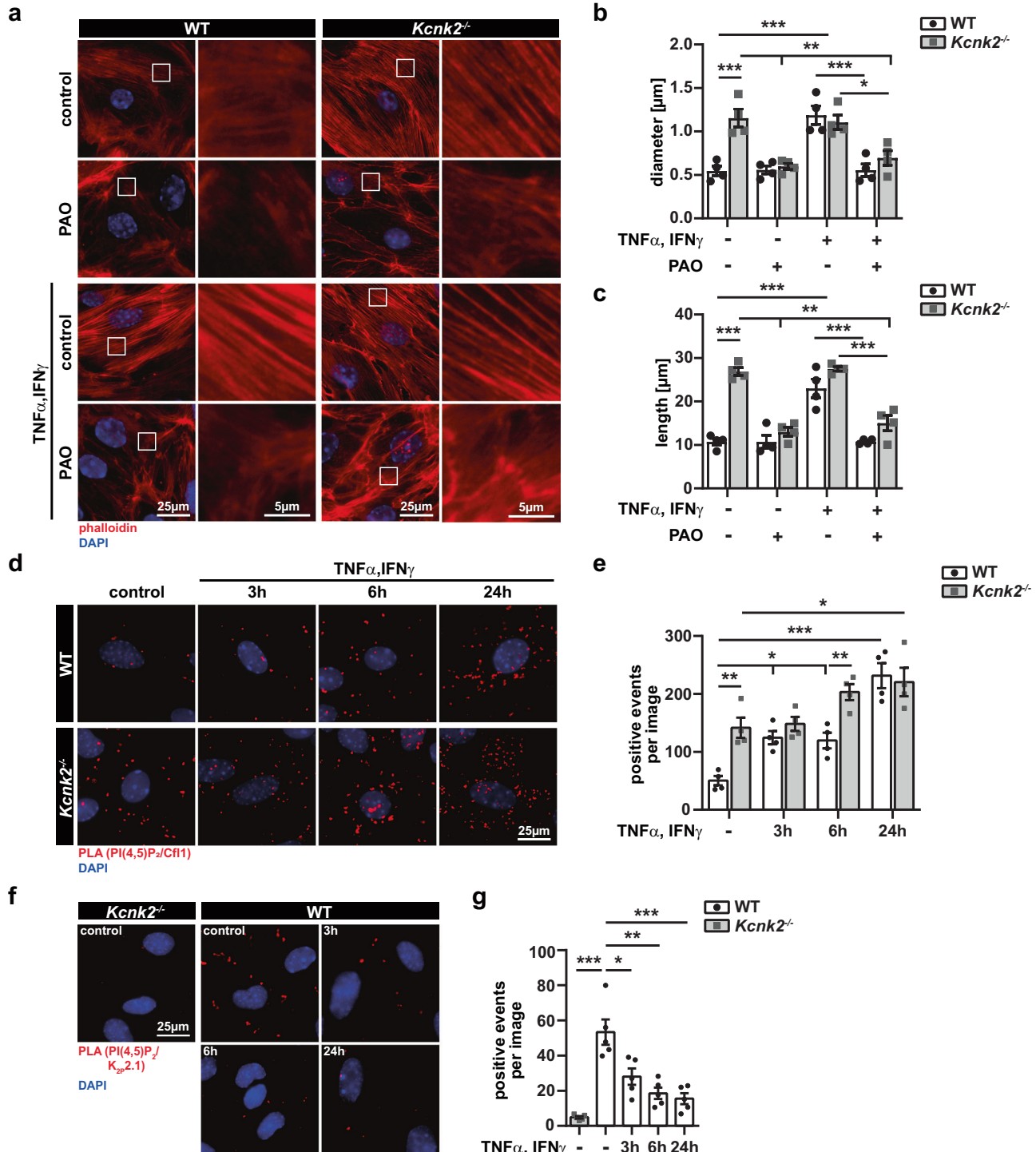

**Fig. 6 | Interaction of K$_{2P}$2.1 and Cfl1 with PI(4,5)P$_2$. a** Representative immunofluorescence images of untreated and TNFα/IFNγ treated (for 3 h, 6 h, 24 h) *Kcnk2$^{-/-}$* and WT MBMECs. Actin is stained in red using phalloidin, nucleus in blue (DAPI). White boxes indicate the magnified image depicted on the right side of each panel. Scale bars represent 25 μm and 5 μm, respectively. **b, c** Quantification of actin stress fiber diameter (**b**) and length (**c**) in images from (**a**) of WT (white, black dots) and *Kcnk2$^{-/-}$* (gray, gray cubes) MBMECs (*N* = 4). **d** Representative immunofluorescence staining of untreated and inflamed (TNFα/IFNγ for 3 h, 6 h and 24 h) WT and *Kcnk2$^{-/-}$* MBMECs using a proximity ligation assay (PLA). Red dots indicate interaction of Cfl1 with PI(4,5)P$_2$ in the PLA, DAPI is shown in blue. Scale bar represents 25 μm.

**e** Quantification of dots in the PLA shown in (**d**). Positive events per image are depicted (N = 4). **f** Representative immunofluorescence staining of untreated and inflamed (TNFα/IFNγ for 3 h, 6 h and 24 h) WT MBMECs using a PLA. Untreated *Kcnk2$^{-/-}$* MBMECs were used as control for the PLA. Red dots indicate interaction of K$_{2P}$2.1 with PI(4,5)P$_2$ in the PLA, DAPI is shown in blue. Scale bar represents 25 μm. **g** Quantification of dots in the PLA shown in (**f**). Positive events per image are depicted (N = 5). N representing the number of individual MBMEC preparations. All data are shown as mean +/- SEM. Statistical analysis using (**b**–**g**) 1-way ANOVA + Bonferroni correction with *$p < 0.05$, **$p < 0.01$ and ***$p < 0.001$. Exact *p*-values are listed in the Source Data file.

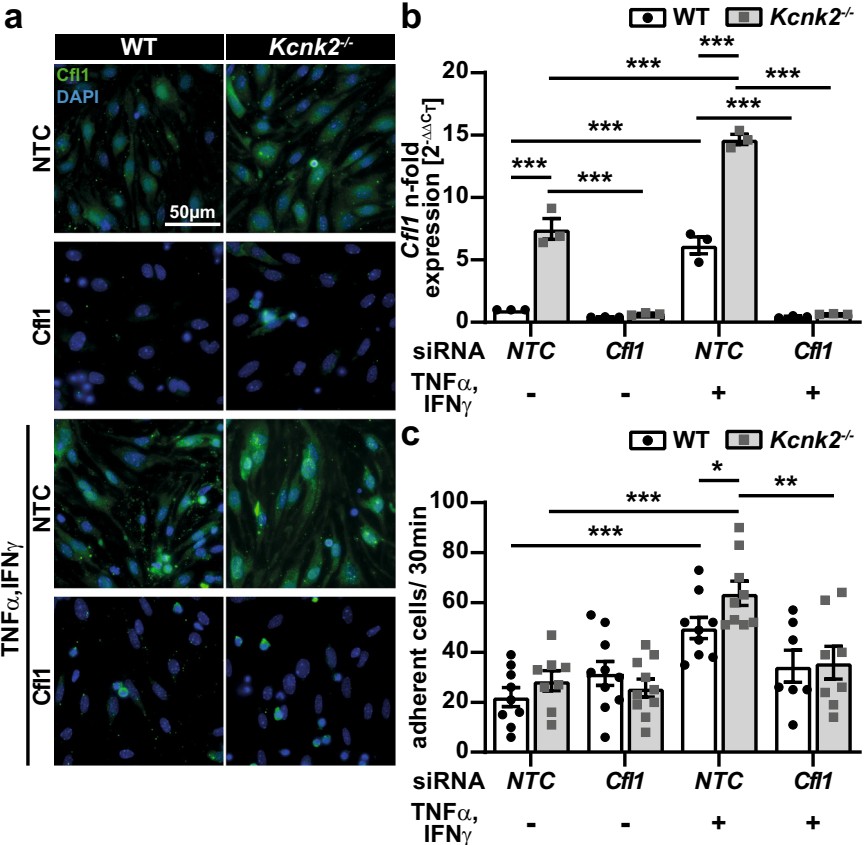

**Fig. 7 | Knock-down of Cfl1 decreases number of adherent T cells under inflammatory conditions. a** Representative immunofluorescence staining of untreated and inflamed (TNFα/IFNγ for 24 h) WT and *Kcnk2⁻/⁻* MBMECs, transfected with 240 nM of non-target control (NTC) or *Cfl1* SMARTpool. Cfl1 is shown in green, DAPI in blue, scale bar represents 50 μm. **b** mRNA expression levels of *Cfl1* in untreated and inflamed (TNFα/IFNγ for 24 h) WT (white, black dots) and *Kcnk2⁻/⁻* (gray, gray cubes) MBMECs, transfected with 240 nM of non-target control (NTC) or *Cfl1* SMARTpool assessed by qRT-PCR. Data are shown as n-fold change, normalized to the respective NTC WT MBMEC control condition. Statistics were

calculated with $\Delta C_T$ values ($N = 3$). **c** Untreated and inflamed (TNFα/IFNγ for 24 h) WT and *Kcnk2⁻/⁻* MBMECs, transfected with 240 nM of non-target control (NTC) or *Cfl1* SMARTpool MBMECs were seeded into μ-slides (ibidi®); stimulated WT CD4⁺ T cells were applied under low flow (0.25 dyn/cm²). Total number of T cells adhering to the endothelial cells within 30 min of acquisition ($N = 8$-10). N representing the number of individual MBMEC preparations. Exact N-numbers for each condition are listed in the Source Data file. Data are presented as mean ± SEM. Statistical analysis using (**b**, **c**) 2-way ANOVA + Bonferroni correction with *$p < 0.05$, **$p < 0.01$ and ***$p < 0.001$. Exact $p$-values are listed in the Source Data file.

ions[53–59]; nevertheless we cannot entirely rule out further effects of $K_{2P}2.1$ on the actin cytoskeleton via its potassium conduction function. Cfl1 knockdown restored T cell adhesion to baseline levels in inflamed MBMECs from both WT and *Kcnk2⁻/⁻* mice, highlighting the critical role of Cfl1 in pathways downstream of K2P2.1 regulation. Overall, these findings demonstrate that $K_{2P}2.1$ modulates the spatiotemporal interactions of actin regulatory factors and thereby actin dynamics. However, the observed regulation of the gene expression levels of *Cfl1* and both regulating proteins, *Limk1* and *Ssh1*, still needs to be investigated further. It would be important to define the transcription factors that are involved in the regulation of those genes and their dependence on $K_{2P}2.1$ expression levels. In this study, we decided to focus on the regulation of the protein function and localization.

Thus, we propose the following mechanism by which $K_{2P}2.1$ regulates actin network dynamics in brain endothelial cells: For membrane targeting of $K_{2P}2.1$ it was suggested that binding to actin is essential, which is mediated by amino acids E306 and S333, as well as to a hydrophobic region within the last 20 amino acids of the C-terminal domain (Fig. 8a)[2,3,22]. To reach the gated, mechanosensitive conformation, positively charged amino acids in the C-terminal domain of $K_{2P}2.1$ bind to PI(4,5)P₂ attaching the C-terminal domain to the cell membrane (Fig. 8b) forming specialized membrane

patches[2]. At those patches the attached C-terminal domain might prevent the access for actin regulating proteins and factors to bind to PI(4,5)P₂[38,49]. $K_{2P}2.1$ downregulation upon inflammation or genetic deletion of *Kcnk2⁻/⁻*, enables PI(4,5)P₂ to directly interact with actin regulating proteins at the plasma membrane. The disruption of the mechanosensitive patch induces actin turnover via Cfl1 and Arp2/3, thereby inducing protrusion formation and cortical stiffening at the plasma membrane (Fig. 8c). Moreover, these processes might induce stress fiber formation in the cell body by depletion of ADFs. Cfl1 can bind to accessible PI(4,5)P₂ at the plasma membrane, decreasing Cfl1 levels and actin depolymerization inside the cell body, thereby facilitating stress fiber formation. $K_{2P}2.1$ is associated with actin stress fibers after internalization in inflamed WT MBMECs. This association might allow rapid and dynamically regulated redistribution to the membrane by vesicle recycling mechanisms as demonstrated for other channels[27,60,61]. Thereby, $K_{2P}2.1$ conformation and localization translate various physiological and pathophysiological stimuli into specific structural changes of the actin cytoskeleton and thereby into a stimulus-related cellular response.

As a secondary effect of the actin cytoskeleton reorganization, ICAM1 may be relocated and anchored by proteins of the ERM complex, EBP50, α-actinin, filamin b and cortactin to those membrane

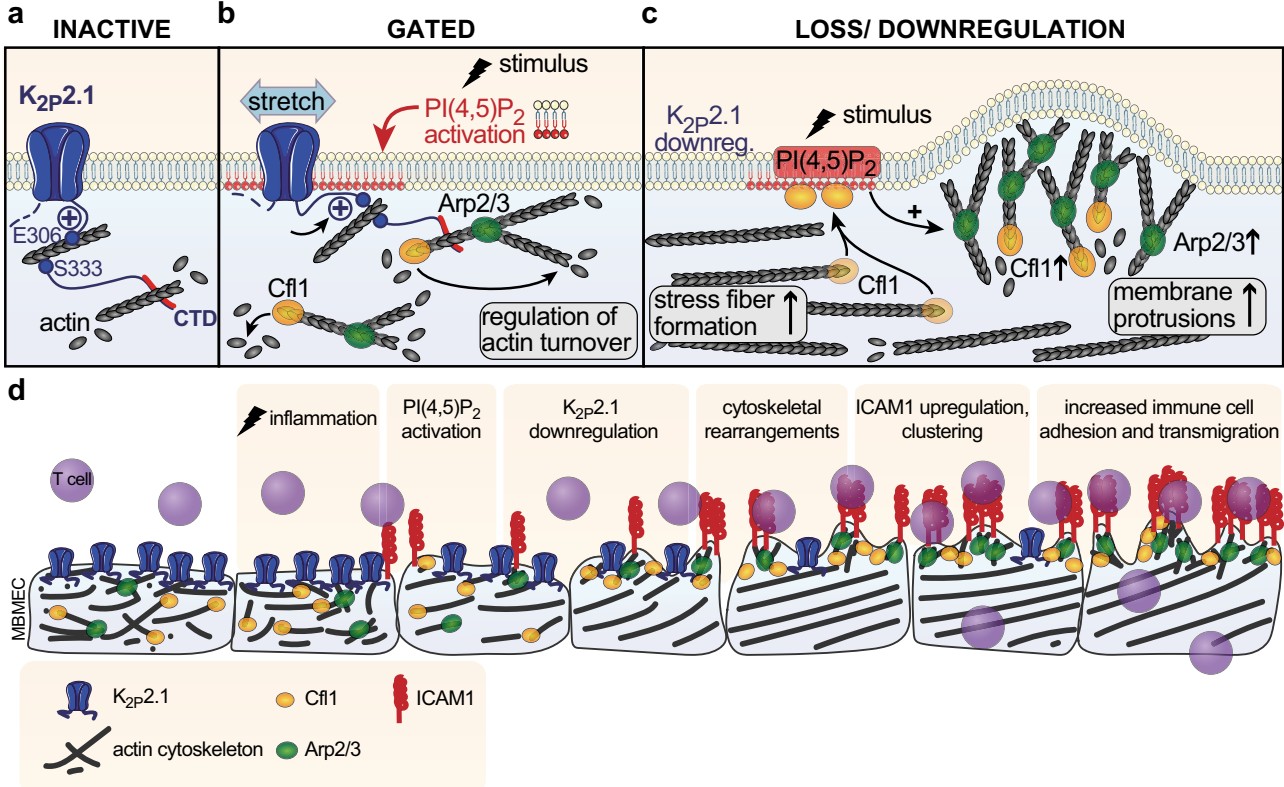

**Fig. 8 | Model of interaction mechanisms between K$_{2P}$2.1 and the actin cytoskeleton. a** K$_{2P}$2.1 (blue) is present as a homodimer in the plasma membrane. No interaction of the C-terminal domain of K$_{2P}$2.1 with the membrane in the closed state of the channel. Dephosphorylated E306 and S333 ensure membrane targeting of K$_{2P}$2.1 and provide potential interaction sites with the actin cytoskeleton (black lines). Additionally, binding to actin takes place by a hydrophobic region (red). **b** Gated state of K$_{2P}$2.1 is induced by binding of positively charged amino acids (+) in the C-terminal domain to PI(4,5)P$_2$. K$_{2P}$2.1 is thereby sensitive to stretch-activation. Actin turnover in close vicinity to this membrane patch is regulated in space and time by anchoring K$_{2P}$2.1 to PI(4,5)P$_2$. This binding prevents the interaction of PI(4,5)P$_2$ and Cfl1 which in turn leads to a dampened Arp2/3(green) and Cfl1 (yellow) activity. Upon mechanical, stretch-induced, activation of K$_{2P}$2.1, actin

depolymerization is activated. The actin cytoskeleton stays in a tightly regulated and controlled state. **c** Downregulation of K$_{2P}$2.1 after internalization or knock-out results in an altered regulation of actin turnover causing membrane protrusion induction at the cell periphery and formation of actin stress fibers inside the cell body. This effect is mediated by the disrupted interaction of K$_{2P}$2.1 with PI(4,5)P$_2$ (orange) and thereby the possible binding of Cfl1. Arp2/3 induces protrusion formation and exclusion of Cfl1 and Arp2/3 in the cell body stabilize actin stress fibers. K$_{2P}$2.1 can be found in close association to actin stress fibers. **d** Overview of K$_{2P}$2.1-mediated processes in MBMECs as described in (**a**–**c**) and the impact on ICAM1 expression, distribution and T cell (purple) adhesion and transmigration at the blood-brain barrier.

protrusions[62,63]. These alterations may be ultimately responsible for the regulation of immune cell migration via the BBB (Fig. 8d) as demonstrated by our 2-photon experiments, where K$_{2P}$2.1 blockade led to a rapid increase of transmigrated cells that was reversed by additional ICAM1 blockade. Moreover, the worsened disease course and lack of additional effect from spadin treatment in EAE of endothelial cell specific *Kcnk2*$^{−/−}$ mice strongly support the in vivo relevance of the endothelial expression of K$_{2P}$2.1.

Thus, K$_{2P}$2.1 is a key regulatory element in actin reorganization and thereby in membrane protrusion and stress fiber formation as well as cortical stiffening. K$_{2P}$2.1 regulated actin dynamics may influence various physiological and pathological processes (e.g. infectious, malignant, inflammatory or degenerative disorders) involving BBB function and integrity. Therefore, pharmacological modulation of K$_{2P}$2.1 provides a promising strategy to regulate the BBB.

## Methods
### Mice
The animal work within this study was approved by the institutional Animal Care Committee and appropriate state committees for animal welfare (AZ: A17.019; G14-1-038). All animal procedures were performed in accordance with the European Union normative for care and use of experimental animals and the German Animal Protection Law.

Mice were raised in an in-house animal facility or purchased from Charles River laboratories. Mice were maintained at 20–24 °C on a 12 h light/12 h dark cycle in individually ventilated cages and 50–60% humidity under Specific-Pathogen-Free conditions with food and water ad libitum. All mice were on a C57BL/6 background and used at the age of 8–15 weeks. Fluorescently labeled T cells were derived from B6.2D2.GFP mice[64,65]. Adoptive EAE was induced in *Rag2*$^{−/−}$*cgn*$^{−/−}$[66]. Active EAE was induced in *Kcnk2*$^{fl/fl}$*xTie2*$^{cre−}$ or *Kcnk2*$^{fl/fl}$*xTie2*$^{cre+}$ littermates. Tie2$^{cre}$ mice were purchased from the Jackson Laboratory (Strain No. 008863). The *Kcnk2*$^{fl/fl}$ mice were generated by Boris V. Skryabin and Timofey S. Rozhdestvensky (Department of Medicine, Core Facility Transgenic animal and genetic engineering Models (TRAM), University of Münster, Germany). The generation of the *Kcnk2*$^{fl/fl}$ mice is described in an extra section below. To avoid or minimize germline deletion of the floxed allele in Cre-lox experiments, Tie2$^{Cre}$ females were mated with floxed males. In addition, the genotyping protocol included the detection of germline deletion or recombination. *Kcnk2*$^{−/−}$[67] or C57BL/6 wildtype mice were used for MBMEC isolation. For EAE experiments, only female mice were used, in line with established protocols to ensure a robust and consistent disease course. For primary cell isolation, both male and female mice were used to reflect biological variability. Sex was not a primary variable of interest, and data from both sexes were pooled for analysis.

## MBMEC isolation and culture

MBMECs were isolated as described before[68,69]. In brief, ten 8–15 week-old mice were sacrificed and brains were isolated. Brain stem, cerebellum, thalamus and meninges were removed. Brains were dissociated and debris as well as red blood cells removed using the Adult Brain Dissociation Kit, mouse, and rat (Miltenyi Biotec, Bergisch Gladbach, NRW, Germany), following the manufacturer's instructions. The CD45⁻CD31⁺ endothelial cells were then purified by depletion of CD45⁺ cells and by positive selection of CD31⁺ cells using microbeads (Miltenyi Biotec, Bergisch Gladbach, NRW, Germany). MBMECs were cultured on coated plates (40 % collagen type IV from human placenta and 10% fibronectin from bovine plasma) for 2 days in MBMEC medium (20% PDS, 0.05% Heparin, 0.05% bFGF in DMEM) containing 0.1% puromycin and 4–5 days in puromycin-free MBMEC medium, in a humidified incubator with 5% $CO_2$ at 37 °C. For all experiments, MBMECs were allowed to grow as a confluent mono-layer. For each experiment a new batch of MBMECs was isolated from 10 pooled mice per genotype, if not stated otherwise.

## Flow cytometry

MBMECs from WT and $Kcnk2^{-/-}$ mice were isolated according to the protocol described above. Cells were directly used for flow cytometry analysis after isolation. Cells were stained for CD31 (CD31-BV605, Biolegend #102427, 1:150), CD45 (CD45-APC/Cy7, Biolegend #103116, 1:200) and $K_{2P}2.1$ (rabbit anit-$K_{2P}2.1$ (Sigma-Aldrich # T6448, 1:100), secondary antibody Allophycocyanin (APC) goat anti-rabbit IgG (Jackson-ImmunoResearch #111-136-144, 1:500)). CD31⁺CD45⁻ cells are defined as endothelial cells and were used for further analysis of expression levels of $K_{2P}2.1$. Data were acquired using the CytoFLEX S (Beckman Coulter) and CytExpert software (Beckman Coulter). Analysis was performed with Kaluza Analysis V2.1 (Beckman Coulter).

## Measurements of cortical stiffness

Mechanical stiffness of the endothelial cortex was determined using an atomic force microscope (MultiMode SPM, Bruker, Germany) as described elsewhere[70,71]. Briefly, a MultiMode 3 SPM AFM (Bruker, Germany) equipped with a feedback-controlled heating device (Nanoscope Heater Controller; Digital Instruments, Veeco, USA) was used to create force-distance-curves of the endothelial cells (MBMECs). To determine exclusively the stiffness of the endothelial cell cortex, soft triangular cantilevers (Novascan, USA) with a nominal spring constant of 0.03 N/m and a polystyrene sphere (10 μm) as a tip were used, with a ramp size of 2 μm and a trigger threshold of 100 nm were chosen. Measurements were performed at 37 °C in HEPES-buffered solution, containing 1% FCS to ensure an intact endothelial function. From each cell preparation the stiffness of ~20 endothelial cells was determined by recording and averaging 6 force distance curves for each cell. All obtained AFM data were collected with NanoScope software 5.31 and V8.10 (Bruker). Stiffness values were calculated from the force distance curves using the Protein Unfolding and Nano-Indentation Analysis Software PUNIAS 3D. We analyzed the slope of the force distance curves that gives direct information about the force (in pN), which is needed to indent the cell for a given distance (in nm). Here, the force is defined as cellular "stiffness": the stiffer the sample, the higher the deflection of the cantilever, i.e., the steeper the slope of the force distance curve. The contact point of the cantilever was defined by performing a baseline fit. Starting from this point, the first slope of the force distance curves has been analyzed, which could independently be shown to reflect the cell cortex. The analysis was performed using the Protein Unfolding and Nano-Indentation Analysis Software (PUNIAS) using the mode for "nanoindentation"[72].

## Single cell force spectroscopy

We employed a protocol modified from[73]. MBMECs, obtained from 4–8 individual preparations (10 mice pooled per preparation), were cultured in 2-well culture-inserts (ibidi®) on collagen/fibronectin coated plastic petri dishes for 48 h prior to inflammation using 500 U/ml TNFα and 500 U/ml IFNγ for 24 h. For measuring the cell-cell adhesion forces between MBMECs and a single T cell a JPK CellHesion 200 (JPK, Berlin, Germany) was used. Petri dishes were heated at 37 °C during measurements. An Arrow TL-1 tipless cantilever was pre-incubated for 30 min in Wheat Germ Agglutinin (WGA) to attach a CD4⁺ T cell to the cantilever. The WGA-coated and calibrated cantilever (spring constant 0.03 N/m) was brought into contact with stimulated CD4⁺ T cells for 2 s and a loading force of 1.5 nN T cells had been placed next to MBMECs. Adhesion force was measured by obtaining a sequence of minimum of 20 force-distance curves of T cell carrying cantilevers that were lowered on MBMEC cells with a constant force of 1.5 nN for 2 s contact time (constant height mode, 80 μm pulling length, approach/retraction speed 10 μm/s). Each CD4⁺ T cell was approached to 20–30 different endothelial cells as technical replicates. The measurements were repeated on 17–19 individual T cells per condition. Maximal adhesion force and adhesion energy were analyzed using JPK Data Processing software (Bruker). Automatic baseline subtraction was applied to force curves and minimum force value and area under the curve were used for analysis.

## Imaging and analysis of membrane surface and stress fibers by AFM

Arbitrarily chosen areas of 400 μm² were recorded from each sample. Surface object counting (nAnostic™ method) was performed using proprietary algorithms for AFM images (Serend-ip GmbH, Münster, Germany). The analysis was performed as previously described[9]. The analysis of the stress fibers was performed using ImageJ (Fiji)[74]. Noise was corrected applying a line wise median filter according to the scan directions. The image was smoothed with a bandpass filter of ImageJ suppressing horizontal signals and by a convolution with a Gaussian kernel. A sobel edge detector was applied and the image blurred again with a Gaussian kernel. The Fast Fourier Transformation was calculated after application of a Hann window v*(0.5−0.5 * cos(pi * (1-n/N))). After a lowpass filter applied to the Fourier transformed image the angular mean intensity was measured. Cardinal directions were ignored to avoid signals due to the scanning direction of the atomic force microscope. The standard deviation of the angular mean intensity is referred to as the fibrosity index.

## Immunofluorescence staining

MBMECs, cultured on coverslips, were fixed with 4 % paraformaldehyde at room temperature for 15 min. Blocking was performed with PBS containing 1 % goat serum, 1 % donkey serum 5 % BSA and 0.2 % Triton-X at room temperature for 30 min. Cells were stained with primary antibodies (rat anti-ICAM1 (Abcam #25375, 1:100) and phalloidin Cruz-Fluor 450 (Santa Cruz Biotechnology, 1:250)) in PBS, 5 % BSA at 4 °C overnight, following secondary antibody staining (Cy3 donkey-anti rat (Dianova # 712-166-153, 1:500) in PBS, 5 % BSA at room temperature for 1 h). Staining of Cfl1 and pCfl1, rabbit anti-Cofilin 1 (Abcam # ab42824, 1:400) and rabbit anti-p (S3) -Cofilin 1 (Abcam # ab12866, 1:400) were used with AF488 goat anti-rabbit secondary antibody (Invitrogen # A11034, 1:500). Images were acquired with a Leica SP-8 laser scanning confocal microscope or a Zeiss Axioscope fluorescence microscope. Images of 4 arbitrary chosen regions per coverslip were chosen for analysis. For analysis of stress fibers, MBMECs were stained with phalloidin Cruz-Fluor 590 (Santa Cruz Biotechnology, 1:500) and DAPI at room temperature for 1 h. Per coverslip images from 4 different randomly chosen areas were taken for further analysis. Stress fiber analysis was performed using ImageJ (Fiji)[74]. For measuring the diameter of actin fibers, the intensity profile of 20 fibers per image was plotted and base to base distance was measured. For measuring the length of actin fibers, also 20 different fiber length per image were measured.

## Proximity ligation assay

Inflamed (500 u/ml TNFα / IFNγ for 3 h, 6 h and 24 h) and untreated MBMECs were cultured on 12 mm coverslips and fixed with 4 % PFA, 0.2 % glutaraldehyde for 15 min at room temperature. Cells were washed with PBS, 50 mM NH₄Cl, followed by blocking in blocking buffer (pH 6.8), containing 20 mM PIPES, 137 mM NaCl, 2.7 mM KCl, 50 mM NH₄Cl, 0.5 % saponin, 5 % donkey serum for 1 h at room temperature. First antibody incubation with rabbit anti-Cofilin 1 (Abcam # ab42824, 1:400), or rabbit anit-K₂ₚ2.1 (Sigma-Aldrich # T6448, 1:100) and mouse anti-PIP2 (Santa Cruz Biotechnology # sc-53412, 1:80) was performed in buffer (pH 6.8), containing 20 mM PIPES, 137 mM NaCl, 2.7 mM KCl, 0.1 % saponin and 5 % donkey serum over night at 4 °C. Following first antibody incubation, reagents from the *Duolink*™ In Situ Red Starter Kit Mouse/Rabbit (Sigma-Aldrich # DUO92101) were used according to the manufacturer's protocols. In brief, cells were washed in buffer A and incubated with the PLA probe, containing the secondary antibodies, for 1 h at 37 °C. Cells were washed in buffer A and the *ligase* was added for 30 min at 37 °C. After a washing step, amplification was performed at 37 °C for 100 min. The last washing step was performed with buffer B and 0.01x buffer B for 1 min at room temperature. Cells were prepared for imaging with *Duolink*® mounting media with DAPI. Per coverslip 4 randomly chosen images were taken at 100x magnification using the Zeiss Axioscope fluorescence microscope. Image analysis was performed with ImageJ (Fiji). Therefore, background was subtracted, images were converted into binary images, followed by Gaussian blur and watershed correction. Positive signals were counted automatically by the finding maxima command in ImageJ. The mean of positive signals per coverslip was used for statistical analysis.

## Gene expression analysis

C57BL/6 and *Kcnk2⁻ᐟ⁻* MBMECs were cultured in MBMEC medium on collagen/fibronectin-coated 24 well-plates for 7 days prior to inflammation. Cells were treated with 500 U/ml TNFα and 500 U/ml IFNγ for 30 min, 6 h, 24 h and RNA was isolated using RNA isolation Kit (micro RNA isolation Kit, Zymo Research). Random hexamer primers were used for cDNA synthesis. Quantitative real-time-PCR (qRT-PCR) analysis was performed using FAM-labeled Taqman primer (Applied Biosystems) *Kcnk2* (Mm01323942_m1). To control for the Cfl1 knock-down efficiency, the using FAM-labeled Taqman primer (Applied Biosystems) *Cfl1* (Mm03057591_g1) was used. VIC-labeled 18S rRNA was used for internal control. For analyzing the transcriptome with RT² Profiler PCR array (Qiagen), RT² First Stand Kit (Qiagen) was used for cDNA synthesis. For analysis of transcripts related to cytoskeletal regulation, cDNA was prepared for quantitative real time PCR (qRT-PCR) using the RT² SYBR Green qPCR Mastermix (Qiagen) and loaded onto plates from the RT² Profiler PCR Array Mouse Cytoskeleton Regulators (Qiagen, PAMM-088Z). All kits were used according to the manufacturer's protocol. The qRT-PCR was performed in 2–3 technical replicates. Data analysis was performed using Qiagen GeneGlobe data analysis tool for RT²-qPCR Arrays. $\Delta C_T$ values were calculated, using the arithmetic mean of *Gapdh* and *Gusb* housekeeping genes. Fold changes were calculated using the $2^{(-\Delta\Delta CT)}$ method relative to the WT control group.

## Proteomics

MBMECs from naïve WT and *Kcnk2⁻ᐟ⁻* mice were isolated according to the protocol described above and directly snap frozen in liquid nitrogen after isolation. Samples were stored at −80 °C until further processing.

**Proteolytic digestion.** Samples were processed by single-pot solid-phase-enhanced sample preparation (SP3) as detailed before[75,76]. In brief, cell pellets were lysed using an SDS-containing buffer (1 % (w/v) SDS, 5 mM dithiothreitol (DTT), 1 x protease inhibitor cocktail

(COMPLETE protease inhibitor cocktail, Roche), 50 mM HEPES pH 8.0). To promote cell lysis, samples were heated for 5 min at 95 °C followed by sonication at 4 °C for 15 min using a Bioruptor (Diagenode, Liège, Belgium). After cell lysis, proteins were reduced and alkylated using DTT and iodoacetamide (IAA), respectively. Afterwards, 2 µL of carboxylate-modified paramagnetic beads (Sera-Mag SpeedBeads, GE Healthcare, 0.5 µg solids/µL in water as described by Hughes et al.[75]) were added to the samples. After adding acetonitrile to a final concentration of 70 % (v/v), samples were allowed to settle at room temperature for 20 min. Subsequently, beads were immobilized by incubation on a magnetic rack for 2 min and washed twice with 70 % (v/v) ethanol in water and once with acetonitrile. Beads were resuspended in 50 mM NH₄HCO₃ supplemented with trypsin (Mass Spectrometry Grade, Promega) at an enzyme-to-protein ratio of 1:25 (w/w) and incubated overnight at 37 °C. After overnight digestion, acetonitrile was added to the samples to reach a final concentration of 95 % (v/v). Subsequently, samples were incubated for 20 min at room temperature. To increase the yield, supernatants derived from this initial peptide-binding step were additionally subjected to the SP3 peptide purification procedure as described before[76]. Each sample was washed with acetonitrile. To recover bound peptides, paramagnetic beads from the original sample and corresponding supernatants were pooled in 2 % (v/v) dimethyl sulfoxide (DMSO) in water and sonicated for 1 min. After 2 min of centrifugation at 15,000 g and 4 °C, supernatants containing tryptic peptides were transferred into a glass vial for MS analysis and acidified with 0.1 % (v/v) formic acid.

**Liquid chromatography-mass spectrometry (LC-MS) analysis.** Tryptic peptides were separated using an Ultimate 3000 RSLCnano LC system (Thermo Fisher Scientific) equipped with a PEPMAP100 C18 5 µm 0.3 × 5 mm trap (Thermo Fisher Scientific) and an HSS-T3 C18 1.8 µm, 75 µm x 250 mm analytical reversed-phase column (Waters Corporation). Mobile phase A was water containing 0.1 % (v/v) formic acid and 3 % (v/v) DMSO. Peptides were separated running a gradient of 2–35 % mobile phase B (0.1 % (v/v) formic acid, 3 % (v/v) DMSO in ACN) over 40 min at a flow rate of 300 nL/min. Total analysis time was 60 min including wash and column re-equilibration steps. Column temperature was set to 55 °C. Mass spectrometric analysis of eluting peptides was conducted on an Orbitrap Exploris 480 (Thermo Fisher Scientific) instrument platform. Spray voltage was set to 1.8 kV, the funnel RF level to 40, and heated capillary temperature was at 275 °C. Data were acquired in data-independent acquisition (DIA) mode. Full MS resolution was set to 120,000 at *m/z* 200 and full MS automated gain control (AGC) target to 300 % with a maximum injection time (IT) of 20 ms. Mass range was set to *m/z* 345 – 1250. Fragment ion spectra were acquired with an AGC target value of 1000 %. In total, 25 windows with varying sizes (adjusted to precursor density) were used with an overlap of 0.5 Da. Resolution was set to 30,000 and IT was determined automatically ("auto mode"). Normalized collision energy was fixed at 27 %. All data were acquired in profile mode using positive polarity. Samples were analyzed in three technical replicates.

**Data analysis and label-free quantification.** DIA raw data acquired with the Exploris 480 were processed using DIA-NN (version 1.7.15)[77] applying the default parameters for library-free database search. Data were searched against a custom compiled database containing UniprotKB/Swissprot entries of the mouse reference proteome (UniProtKB release 2020_03, 17,033 entries) and a list of common contaminants. For peptide identification and in-silico library generation, trypsin was set as the protease allowing one missed cleavage. Carbamidomethylation was set as the fixed modification and the maximum number of variable modifications was set to zero. The peptide length ranged between 7–30 amino acids. The precursor *m/z* range was set to 300–1,800, and the product ion *m/z* range to 200–1,800. As the quantification strategy we applied the "any LC (high

accuracy)" mode with RT-dependent median-based cross-run normalization enabled. We used the in-build algorithm of DIA-NN to automatically optimize MS2 and MS1 mass accuracies and scan window size. Peptide precursor FDRs were controlled below 1 %. In the final dataset, proteins had to be identified by at least two peptides. Statistical analysis of the data was conducted using Student's $t$-test, which was corrected by the Benjamini–Hochberg (BH) method for multiple hypothesis testing (FDR of 0.01). In addition, differentially expressed proteins had to show a fold change of at least 1.4 (i.e., $\log2 > 0.5$ or $\log2 < -0.5$). Network analysis was conducted using the STRING database (version 11.0)[78] through its web interface as well as the stringApp[79] in Cytoscape (version 3.8.2)[80]. Protein networks in STRING were generated using default settings. GO Terms were identified by Panther analysis[81] and subsequently reduced using REVIGO[82]. Mass spectrometry-based proteomic data have been deposited in the ProteomeXchange Consortium via the PRIDE partner repository[83] with the dataset identifier PXD031051.

### T cell isolation and culture
Murine T cells were isolated from spleens of C57Bl/6 mice after homogenization by a cell strainer (pore size 40 µm). CD4$^+$ cells were isolated using CD4$^+$ T cell isolation kit from Miltenyi Biotec (Germany). Isolated CD4$^+$ T cells were stimulated for 48 h with 2 µg/ml plate-bound purified anti-mouse CD3 antibody (clone 145-2C11, Biolegend) and 2 µg/ml soluble purified anti-mouse CD28 antibody (clone 37.51, Biolegend) in splenocyte complete medium (5% FCS, 0.1% β-mercaptoethanol, 25 µg/ml Gentamycin, 1% NEAA,10 mM HEPES in DMEM) in a humidified incubator with 5% $CO_2$ at 37 °C.

### Migration Assay
After 5 days of culture MBMECs were seeded on fibronectin/collagen coated ibidi® µ-slides IV 0.4 (20,000 cells per chamber) for flow conditions to allow the cells growing to reach confluency. Thereafter, MBMECs were inflamed (500 U/ml TNFα / IFNγ) for 24 h prior to video recordings. Stimulated CD4$^+$ T cells were added to the MBMEC cultures for image acquisition Zeiss Axiovert A.1 microscope was used to record time series bright-field images (flow: 1 image per 10 s for 30 min). Cells were imaged in a heating chamber at 37 °C. Flow conditions were set up using a pump system with CD4$^+$ T cells in a 10 ml syringe (10$^6$ cells/ml) at a flow rate of 0.25 dyn/cm$^2$. Migration of T cells was analyzed by TrackMate plugin from ImageJ (Fiji)[74].

### si-RNA knock down
To achieve knockdown of Cofilin (Cfl1) in MBMECs, either a SMART pool of Cfl1-specific siRNA (Horizon Discovery, L-058638-01-0005) or a mix of four siRNAs (LQ-058638-01-0002, Horizon Discovery) was used. For control conditions, cells were transfected with a non-targeting siRNA pool (D-001810-10-05, Horizon Discovery). Lipofectamine RNAiMAX (Thermo Scientific, #13778100) served as the transfection reagent.

A siRNA-lipid complex transfection mix was prepared to a final concentration of 240 nM siRNA and 1.1% (v/v) Lipofectamine in Opti-MEM™ I medium (serum-reduced and without phenol red; Gibco, #11058021) and incubated for 20 min to allow complex formation. Following incubation, the transfection mix was applied to cells maintained in OptiMEM™ I medium. The transfection was conducted overnight for 8–12 h, after which the medium was exchanged to MBMEC control medium to proceed with subsequent experiments.

### Experimental Autoimmune Encephalomyelitis (EAE)
**Adoptive transfer EAE.** For adoptive transfer EAE, spleens and lymph nodes from B6.2D2.GFP and B6.2D2 mice were dissected after cervical dislocation. After magnetic bead-based cell sorting (MACS, Miltenyi Biotec), performed in accordance with the manufacturer's protocol, fluorescent CD4$^+$CD62L$^+$ cells were co-cultured with irradiated CD90$^+$-

depleted B6.2d2 antigen presenting spleen cells (APC, 1:10) in the presence of 2 µg/ml αCD3e (BD, USA), 3 ng/ml TGFβ, 20 ng/ml IL-23, and 20 ng/ml IL-6 (all R&D Systems, Inc., USA) in mouse medium (10% FCS, 1% P/S, 1% L-glutamine, 0.1% β-mercaptoethanol, 1% HEPES in RPMI). T cell activation was further driven by splitting the cells on day 3 and 5 of culture using mouse medium supplemented with 50 U/ml IL-2 (25 U/ml on day 5) and 10 ng/ml IL-23. On day 7, cells were harvested, counted and subsequently seeded again in the presence of freshly isolated irradiated CD90$^+$ depleted APC (1:5) in mouse medium supplemented with 2 µg/ml αCD3e, 0.75 ng/ml TGFβ, 20 ng/ml IL-23 and 10 ng/ml IL-6 for restimulation. Three days after the second restimulation, cultured cells were checked for adequate cytokine release by flow cytometry (15-30% IL-17$^+$) and subsequently $10 \times 10^6$ 2d2.GFP CD4$^+$ T cells were intravenously injected (i.v.) into $Rag2^{-/-}cgn^{-/-}$ mice. The severity of atypical (transfer) EAE was monitored daily using the following scoring system: 0 = no detectable signs of EAE; 1 = complete tail paralysis; 2 = partial hind limb paralysis; 3 = complete hind limb paralysis; 4 = tetraparesis; 5 = death. Mice reaching a score of 2 were selected for intravital two-photon laser microscopy.

**Active EAE.** Induction of experimental autoimmune encephalomyelitis (EAE) was performed in 8–12 week-old female mice as previously described[1,84]. Briefly, MOG35-55 peptide was dissolved in PBS (2 mg/ml) and homogenized with complete Freund's Adjuvant (CFA, 2 mg/ml) in a 1:1 ratio and stored for 30 min at 4 °C. 100 µl emulsified MOG was injected in each flank of anesthetized mice (isoflurane). Injection of pertussis toxin (PTX, 1 µg/µl) was performed on day 0 and day 2 after MOG immunization intraperitoneally (i.p., 200 µl per dose). Health status (weight, disease score, general appearance and performance) of mice was monitored on a daily basis. The severity of EAE was monitored daily using the following scoring system: 0 = no detectable signs of EAE; 1 = complete tail paralysis; 2 = partial hind limb paralysis; 3 = complete hind limb paralysis; 4 = tetraparesis; 5 = death.

### Intravital microscopy
The mice were provided with a carotid artery catheter prior to exposure of the brainstem. Mice preparation and intravital microscopy using two-photon laser scanning technique were previously published[85–87] and are described in more detail in the supplemental experimental procedures. Body temperature was strictly held at 35-37 °C during the operation and microscopy. Vital parameters were continuously monitored using a MouseOx Plus (Starr Life Sciences, USA) and Lab Chart Pro Software (AD Instruments, Australia).

### Statistical analysis
All experiments were randomized and blinded in acquisition and analyses. D'Agostino-Pearson or Shapiro-Wilk test was used for evaluation of normal distribution. For analysis of two groups, two-tailed $T$-test (unpaired/paired) or Kruskal-Wallis test with Bonferroni or Dunn's correction for multiple comparisons was used. Analysis of more than two groups was performed with one-way ANOVA or two-way ANOVA, with Bonferroni correction for multiple comparison. "N" indicates the number of biological replicates on which the statistical evaluation is based. Extended information about statical analysis is summarized in Table S2. Statistical analysis of normalized data was performed on the respective raw data (qRT-PCR data and cortical stiffness measurements). Sample sizes were calculated based on an expected Cohen's $d > 0.5$ (estimated using data from previous comparable studies) to detect medium to large effects with a type I error of $\alpha = 0.05$ and a power of 0.85 (two-tailed, unpaired $t$-test). Effect sizes were not systematically reported in the results. $P$-values $> 0.05$ were classified as not significant, $p < 0.05$ (*) as significant, $p < 0.01$ (**) and $p < 0.001$ (***) as highly significant. Exact $p$-values, N and n-numbers for each experiment are listed in the Source Data file.

**Reporting summary**

Further information on research design is available in the Nature Portfolio Reporting Summary linked to this article.

## Data availability

Mass spectrometry-based proteomic data have been deposited in the ProteomeXchange Consortium via the PRIDE partner repository[83] with the dataset identifier PXD031051. Source data are provided with this paper.

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

## Acknowledgements

This work was funded by the "Else Kröner-Fresenius-Stiftung" (2018_A03 to T.R.), by the "Innovative Medizinische Forschung (IMF)" Münster (I-RU211811 to T.R.), by the Deutsche Forschungsgemeinschaft (DFG, to T.R.: 549557400/RU 2169/5-1, 417677437/GRK2578, SFB CRC TR128 to S.Bi., CRC TR128 project B06 to T.B. and S.G.M., SCHW407/17-1 to A.S., KU 1496/7-1, KU 1496/7-3 and INST 392/141-1 FUGG to K.K.V), by Chembion to A.S. and the Hertie Foundation (mylab to S.Bi.). This work was supported by a fellowship of the Graduate School of the Cells-in-Motion Cluster of excellence (EXC 1003 – CiM), University of Münster, Germany to S.L. We thank Jeannette Budde and Mary Bayer for their excellent technical assistance. Images were adapted from Servier Medical Art (https://smart.servier.com/), licensed under CC BY 4.0 (https://creativecommons.org/licenses/by/4.0/).

## Author contributions

S.L., L.V. S.B., R.T., and S.G.M. conceived and planned the experiments. S.L., L.V., F.S., I.P., N.H., V.D., J.G., C.B.S., H.R., B.W., D.B., C.N., J.F., S.T., U.D., C.F.H., K.K.V., B.S., and T.S.R. performed and analyzed the experiments. S.L. and T.R. wrote the original draft; all authors revised and edited the manuscript.

## Funding

## Competing interests

The authors declare no competing interests.

## Additional information

[1]Department of Neurology, Medical Faculty and University Hospital Düsseldorf, Heinrich Heine University Düsseldorf, Düsseldorf, Germany. [2]Core Facility Flow Cytometry, Medical Faculty and University Hospital Düsseldorf, Heinrich Heine University Düsseldorf, Düsseldorf, Germany. [3]Department of Neurology, University Medical Center of the Johannes Gutenberg-University Mainz, Mainz, Germany. [4]Department of Neurology with Institute of Translational Neurology, University of Muenster, Münster, Germany. [5]IUF-Leibniz Research Institute for Environmental Medicine, Core Unit Model Development Düsseldorf, Düsseldorf, Germany. [6]Institute of Neuropathology, University of Göttingen, Göttingen, Germany. [7]nAnostic Institute, Centre for Nanotechnology, University of Muenster, Münster, Germany. [8]Institute for Immunology, University Medical Center of the Johannes-Gutenberg University Mainz, Mainz, Germany. [9]Institute of Physiology, University of Lübeck, Germany and DZHK (German Research Centre for Cardiovascular Research), Partner Site Hamburg/Lübeck/Kiel, Lübeck, Germany. [10]Core Facility Transgenic Animal and Genetic Engineering Models (TRAM), Medical Faculty, University of Münster, Münster, Germany. [11]Institute of Physiology II, University of Münster, Münster, Germany. [12]Institute of Physiology I, University of Münster, Münster, Germany. [13]Department of Neurology, Ruhr University Bochum, BG University Hospital Bergmannsheil, Bochum, Germany. [14]These authors contributed equally: Stefanie Lichtenberg, Laura Vinnenberg, Sven G. Meuth, Tobias Ruck. ✉e-mail: tobias.ruck@bergmannsheil.de

