## [Transparent Peer Review File · Nature Communications]

The potassium channel K2P2.1 shapes the morphology and function of brain endothelial cells via actin network remodeling

Corresponding Author: Professor Tobias Ruck

Version 0:

Reviewer comments:

Reviewer #1

(Remarks to the Author)

The manuscript # NCOMMS-21-01217-T by Bock et al., is an extension of a previous study that demonstrated a critical role for the K2p 2.1 potassium channel in EAE. Previously, it has been shown that in *Kcnk2*^{-/-} mice, brain endothelial cells upregulate cellular adhesion molecules ICAM1, VCAM1 and PECAM1 and facilitated leukocyte trafficking into the CNS. Following the induction of experimental autoimmune encephalomyelitis (EAE) by immunization with a myelin oligodendrocyte protein (MOG)_{35–55} peptide, *Kcnk2*^{-/-} mice showed higher EAE severity scores that were accompanied by increased cellular infiltrates in the central nervous system (CNS).

In this study, the authors use cell biological approaches in purified mouse BMECs (mBMECs) from wild-type or *Kcnk2*^{-/-} to analyze the cell distribution of the protein under healthy or inflammatory conditions and find that the mutant cells have active stress fibers. The authors perform a differential screen between the wild-type and *Kcnk2*^{-/-} cells and identify several actin regulatory proteins (e.g. Cfl1) that change in mutant cells. Moreover, they use an in vitro system to demonstrate that T cells have a higher interaction force in mutant versus wild-type cells and under inflammatory conditions. Finally, the authors use an adoptive transfer approach to look at T cell trafficking across the CNS blood vessels under treatment with Spadin, an inhibitor of K2p 2.1 potassium channel and ICAM-1 inhibition and show that Spadin treatment enhances T cell trafficking into the CNS and ICAM antibody reverses the effect.

The study is performed carefully and provides some additional insights into the cell biology of the K2p 2.1 potassium channel in endothelial cells and how it may affect interactions with the immune cells during neuro-inflammation. However, the study seems like an extension of the previous work published on this receptor and the work is primarily focused on in vitro studies in mBMECs. Therefore it does not provide a major conceptual breakthrough regarding the role of this receptor in neuroinflammation but rather an incremental information. Moreover, the downstream effectors that link K2p 2.1 potassium channel with the actin cytoskeleton are not explored in detail. Following are some additional concerns as follows:

- 1) In Figure 3, the authors identify some of the downstream targets for K2p 2.1 potassium channel in mBMECs. The authors need to show images for quantification performed in Panel d and perform Western blotting to determine the ration of phosphorylated versus non-phosphorylated forms of Cfl1 in endothelial cells.
- 2) The authors need to perform also in vivo validation with these targets to show that these targets are upregulated under inflammation (EAE) or in *Kcnk2*^{-/-} cells. The validation of the screen between the wild-type and mutant cells in vivo is critical since the mBMECs are very different to in vivo brain endothelial cells.
- 3) The in vitro studies of T cell interactions with the wild-type and *Kcnk2*^{-/-} cells also need to be performed with cells lacking some of the downstream intermediates such as Cfl1 to demonstrate a link between the K2p 2.1 potassium channel and the actin regulation and provide a clear cell biological mechanisms for the activity of K2p 2.1 potassium channel.
- 4) The resolution of images in Figures 1, 2 needs to be higher. The author needs to show the location of insets and provide confocal images with all the profile views in all three dimensions.

5) The authors need to show real images and movies in Figure 6 rather than animated movies.

6) Figure 6d needs to show also the ICAM inhibition.

Reviewer #2

(Remarks to the Author)

The manuscript by Bock et al studies the role of the *kcnk2* potassium channel in brain endothelial cell morphology and leukocyte transmigration. The study uses a combination of in vitro and in vivo approaches to determine the role of *kcnk2* in endothelial cell responses to proinflammatory stimuli and T cell migration. The study of potassium channels in the regulation of BBB functions is an important area of research. The manuscript provides several mechanistic insights using standard cell biology methods regarding changes in subcellular localization of *kcnk2* upon TNF stimulation, redistribution of ICAM-1, and phosphorylation rates of cofilin. The major concern with the manuscript is the very limited in vivo validation of these findings that is preliminary with numerous technical limitations. As a result, whether *kcnk2* plays an in vivo role in the endothelial cells remains largely unknown. Specific comments are listed below.

1. Analysis of EAE by in vivo two photon microscopy focuses on the brainstem, which is not the main site of T cell transmigration. Instead two photon imaging should have been performed in the spinal cord and the migratory patterns of multiple T cells need to be imaged and analyzed.
2. Labeling of blood vessel lumen using dextran cannot be used to document "endothelial crossing". Instead, endothelial cells need to be labeled with fluorescent reporters to document "endothelial crossing".
3. Spadin is administered systemically, therefore it is not possible to determine if the observed effects are due to endothelial expression of *Kcnk2*. Can *Kcnk2* be blocked specifically in endothelial cells?
4. The analysis of BBB needs to include tight junction and plasma proteins. Is there increase only in crossing of cells or increased leaks of the BBB?
5. Spadin is administered acutely. What are the effects on clinical signs and histopathology in the spinal cord in the EAE model after chronic administration of spadin or other pharmacologic or genetic approaches to deplete *kcnk2* in endothelial cells?
6. The EAE experiment is missing controls using Ova-T cells. As only T cells from 2D2 mice are injected, it is unknown whether these responses are specific to encephalitogenic T cells from 2D2 mice or are T cell not recognizing CNS antigens. An Ova T cell control needs to be included. The choice of imaging a single T cell in the brainstem, a region with low T cell infiltration, further impedes addressing these questions.
6. ICAM-1 deficiency has relatively mild protective effects in EAE and it is unclear whether regulating ICAM-1 alone would be sufficient to modulate trafficking. Are there other molecules regulated by *kcnk2*?
7. The number of biological replicates used in the study is not stated. For example in Figure 6, it is mentioned that data are from "n=3-6 EAEs". Are these mice, vessels, or cells? If they are mice, how many vessels have been sampled per mouse? As the variability in Fig. 6g is very high, are the reported differences statistically significant? Have power calculations been performed for Fig. 6e? It appears that n=3 is not sufficient to draw any conclusions given the large variability. If n represents the number of cells from the same mouse, then the study needs to be repeated in at least n=8 EAE mice and the number of mice needs to be used for statistical analysis as an independent biological replicate.

Reviewer #3

(Remarks to the Author)

Reviewer report on "K2P2.1 shapes morphology and function of brain endothelial cells via actin network remodeling" by S. Bock et al. (2021)

In this manuscript the authors report a novel role of K2P2.1, a two-pore domain mechanosensitive potassium channel which express in brain microvascular endothelial cells and is an important regulator of immune cell trafficking/transmigration across the blood-brain barrier (BBB). The manuscript explores the role of K2P2.1 in modulating endothelial cells morphology and filamentous actin cytoskeleton remodeling. The authors show that reduction in K2P2.1 expression in microvascular endothelial cells results in altered morphology, increased in membrane protrusions, and increased formation in stress fibers with K2P2.1 channels colocalizing with stress fibers. In addition, the authors showed that absence of K2P2.1 results in remodeling of the actin cytoskeleton with increase in stress fibers formation and enhanced cellular cortical stiffness. Furthermore, the authors observe that activation of key filamentous actin regulating proteins Cofilin1 and Arp2/3 complex. Then, the authors show in vitro enhanced CD4+ T cell adhesion and morphological changes on *Kcnk2* depleted microvascular endothelial cells. Finally, they perform in vivo experiments and observe increased immune CD4+ T cell adhesion and transmigration in mouse brain stem blood vessels when adding Spadin a pharmacological drug that causes inhibition of K2P2.1. The experiments seem to be performed carefully, and in my view the manuscript shows novel and relevant results. However, there are some aspects as data quantification and some clarifications that the authors need to address.

Broadly speaking, I see a clear contribution of this paper towards understanding mechanosensitive regulators of immune cell trafficking across the blood-brain barrier essential for central nervous system immune and pharmacological therapeutics. The presented research work is interesting, and I fell is of broad interest to the Nature Communications readership and I may recommend its publication after the authors address the following comments:

- 1) Lines 98 to 100; the sentence "The main determinants for cellular mechanical properties..." will be significantly improved

if references are provided/added to this sentence supporting that the actin cytoskeleton and stress fibers are the main determinant of cellular mechanical properties.

2) Lines 159 to 164 and Figure 1e; the cortical stiffness is determined, however the authors report a normalized cortical stiffness instead of the true estimated value. In order to provide the reader a real comparative sense of the stiffness behavior of the endothelial cells it would be great if the authors provide the real stiffness values.

3) Furthermore, following up my previous comment; How the authors extract the stiffness values from the acquired AFM force-distance curves? Do the authors perform a linear regression to fit the force distance curves thus the obtained slope is the measured apparent stiffness? Is the estimated stiffness in N/m units? The authors fail to properly describe in the Materials and Methods how these calculations were made (in lines 403-405 the authors state the use a nano-indentation analysis software named Punias 3D).

4) Lines 215 to 217; I suggest the authors include the ARP2/3 complex data in the main text Figure 3 since this observation is one of the main findings and an important component of the proposed model in Figure 7.

5) Figure 6; statistical analysis and significance labels are missing on panels Figures 6e and 6g. The authors should perform the statistical analysis and add the significance labels.

6) Supplementary Figure 2; statistical analysis and significance labels seems to be missing in presented data for Arpc3 and Arpc4 in Supplementary Fig. 2C.

Reviewer #4

(Remarks to the Author)

Comments:

The manuscript by Bock et al. explores the role of the two-pore domain potassium channel K2P2.1 (TREK1) in the immune cell trafficking into the CNS through a number of cell biology and imaging experiments. The key conclusion by the authors is that K2P2.1 mediates the immune cell transmigration across the blood-brain barrier by influencing the actin cytoskeleton structures, completing and deepening the study published by Bittner et al, Nature Medicine 2013.

Originality and significance

The originality of this work lies on the completing the model described in the Nat Med 2013 in which TREK1 activity regulate the cell transmigration through the BBB. This involving regulation of cytoskeleton and actin dynamics through the mechanosensitive K2P2.1 channel.

Although the observations made by the authors are potentially interesting, the analyses provided appear incomplete and more rigorous analyses would be critical to a better understanding of the role of K2P2.1 in BBB and a better emphasizing of the novelty of this work.

Major concerns/points for improvements are as follows:

The authors used an appropriate battery of tools for demonstrate their data, including immunofluorescence staining, atomic force microscope, single cell force spectroscopy proximity ligand binding assays and in vivo two-photon scanning and the utilization of specific knock-out mice.

1. Figure 1b and 2b. K2P2.1 shares high homology with other members of the same family, making the antibody unspecific. Moreover, some of this members are also mechanosensitive. I suggest to the authors adding a negative control using cells from KO animals for Fig 1B and Fig 2B. what is the labelling of the cells in KO mice. This would give an information regarding AB specificity. Moreover, it would interesting to make it with KO cells to see if clusters arranges similarly to WT cells upon inflammation.

2. qRT-PCR experiments indicate that K2P2.1 mRNA is rapidly decreased (30 minutes) under inflammatory stimulus driven by IFN γ and TNF α , with a minimal expression level reached after 24 hours of treatment. Despite this striking decrease at 24h, the protein level of expression seems not to be reduced even this point is difficult to evaluate on the immunofluorescence staining. The authors should evaluate this discrepancy and quantify K2P2.1 protein expression and turn-over in the different experimental conditions. Moreover, the authors mentioned that K2P2.1 is expressed at the membrane, in the cytosol and close to the nucleus of the MBMECs and they conclude that K2P2.1 protein distribution is altered by the IFN γ /TNF α treatment. To better characterize this channel redistribution, the authors should provide images with a better resolution.

3. 2. AFM experiments highlight the presence of protrusions both in wild-type MBMECs treated with IFN γ /TNF α and in Kcnk2 mutant cells. Other markers should be used to confirm the presence of these structures. Are these protrusions ezrin-rich? The conclusion that loss of K2P2.1 leads to the formation of membrane protrusions with clusters of ICAM-1 is based on co-immunofluorescence analyses with ICAM1 and K2P2.1. Providing images with a better resolution will allow to evaluate this point. The authors should also deepen the analyze of the docking structures and investigate their different components such as ERM proteins, EBP50, filamin b, etc.

4. To reinforce the main message of the manuscript and its novelty, the authors should provide a more detailed analysis of the molecular pathways whereby K2P2.1 modulates BBB permeability and ion flux-dependent or independent functions of this ion channel.

5. Cofilin-1 was identified as the main cytoskeletal regulator involved here. The authors should provide the

immunofluorescence images based on which the Cfl1/P-Cfl1 quantifications have been done (Figure 3c,d).

6. Moreover, the involvement of the different components of ARP2/3 complex is shown here by measurement of their mRNA levels of expression. As ARP2/3 activity is regulated by phosphorylation, to better address ARP2/3 participation, the authors should investigate the phosphorylation status and activity of the complex in the different experimental conditions.

7. RHOA-ROCK signaling should also be investigated to draw a complete picture of the signaling pathways involved in cytoskeletal rearrangements in the different experimental conditions.

8. 4. The authors used a Proximity Ligation Assay to assess whether Cofilin1 and PI(4,5)P2 interact in presence of in absence of Kcnk2. Can the authors provide the reverse experiments in which they look at TREK1 colocalization with PIP2?

9. authors conclude that K2P2.1 has a direct impact on this interaction. What could be this "impact" at a molecular level?

Clarity and context

The way the text is written is clear and easy to understand for general audience. I only found two minor corrections concerning to grammars:

- ICAM1 is intercellular adhesion molecule and not intracellular (page 3)

- Weather for whether (page 11)

Figure 7 is hard to understand and does not show permeation of cells, which is supposed to be the main finding.

Version 1:

Reviewer comments:

Reviewer #1

(Remarks to the Author)

The revised manuscript # NCOMMS-21-01217A addresses some of the concerns raised by several reviewers. The authors have done a tremendous job to elucidate some of the cell biological mechanisms that link K2p 2.1 potassium channel to the actin cytoskeleton in endothelial cells that plays an important role in membrane ruffling and recruitment of immune cells.

Despite these major efforts, there are still several concerns for the revised study.

1. The major concern is that the study still remains largely an in vitro study where the authors have used bEND3.0 cells, a endothelioma cell line that does not resemble at all brain endothelial cells. Therefore, the relevance to the in vivo remains limited. The in vivo validation of these targets to show that they are upregulated under inflammation (EAE) or in Kcnk2^{-/-} cells in mice is not performed. The validation of the screen between the wild-type and mutant cells in vivo is critical since the mBMECs are very different to in vivo brain endothelial cells. Therefore, the study remains limited in scope.

2. In Figure 6, the authors perform in vitro studies of T cell interactions with the wild-type, Kcnk2^{-/-} cells, Kcn2^{-/-} Cfl1^{+/-} cells and Csf1^{+/-} bEnd3.0 cells. Why was knockdown performed in bEnd3.0 cells and not primary brain endothelial cells? As mentioned above bEND3.0 cells are an endothelioma cell line that does not resemble at all brain endothelial cells. Therefore, the value of this experiments is very limited.

3. Figure 4 - The assessment of phosphorylated versus non-phosphorylated Cfl1 data are not convincing.

Reviewer #2

(Remarks to the Author)

The revised manuscript by Bock et al has improved in response to the Reviewers' comments. A major improvement in the manuscript is in vivo validation in newly generated endothelial-specific knock out mice for kcnk2. Although the generation of the Kcnk2^{f/fl} mice is described in the Methods, Figures are not included to demonstrate the targeting strategy, DNA southern blot analysis and immunohistochemical validation of the cell-specific targeting in endothelial cells. Another major concern is the overall rigor of the study and methods of statistical analysis used to derive the conclusions. Most panels are described by double N numbers, such as Fig. 2b (N=3, n=27-34), Fig. 4d (N=4, n=8), etc. The statistics section indicates that "N represents the number of MBMEC preparations; n the number of analyzed single cells or independent coverslips. Statistics was calculated on single cell data, if n is indicated in the figure legend or on mean values, if only N is given". The majority of the study is based on statistical analysis on technical replicates or individual cells. No criterion is provided why in some few cases the N number of biological replicates is used, while in the majority of the study statistics are performed with individual cells. Furthermore, the Figure Legends do not include sufficient information for Reviewers to evaluate the rigor and methods of statistical analysis used in the study. The Figure legends need to be rewritten to clearly define the N used for statistical analysis and eliminate the double "N, n". Source Data can be included to report all the collected data. All statistics performed with individual cells and technical replicates, will need to be repeated with biological replicates (either number of mice or number of independent biological experiments). The exact method of statistical analysis including the post-hoc correction test also needs to be included for each panel. Some examples are given below, but statistics need to be repeated and legends need to be corrected throughout the manuscript.

1. The newly generated Kcnk2^{f/fl} mice are only described in the text in Supplemental Methods with no data shown. Figures need to be included to fully disclose the data on the generation of the Kcnk2^{f/fl} mice including targeting strategy, DNA southern blot analysis, etc. Importantly, immunohistochemistry data need to be included to characterize the cell-specific

depletion of Kcnk2 in endothelial cells. What is the percentage of endothelial cells that do not express Kcnk2? The IHC analysis needs to be performed before and after EAE induction.

2. The breeding strategy of the Kcnk2^{fl/fl} mice for EAE experiments need to be included in the Methods. Were littermate controls used for the EAE study?
3. Figure 1. What does the N refer to? Number of mice, biological replicates, etc? What is the method of statistical analysis used for each panel? In Figure 1f, if statistics have been performed with technical replicates (n=45-49), they need to be repeated with biological replicates.
4. Figure 2: Have the statistics been performed with N=3 or n=27-34? What was the method of statistical analysis used?
5. Figure 5b, c: "N=4, n=8". The bar graphs show 8 data points, assuming that n=8 was used for statistical analysis. In contrast, in Figure 5g "N=5, n=20", the bar graph shows 5 data points. How do the authors choose the N for statistical analysis?
6. Figure 6: The majority of the Figure includes n=17-19 data points. Are these numbers of individual cells? Are results statistically significant if N=biological replicates and/or does the P value change if statistics are performed with biological replicates?
7. Other than the intravital imaging, were there any other experiments or quantification methods blinded in the study? Were the EAE experiments scored in a blinded manner? Were the drugs administered in a blinded manner? Blinding and randomization for the preclinical pharmacology studies need to be described.
8. Figure 7h "(h) EAE disease course of endothelial cell-specific Kcnk2^{-/-} and control mice with or without spadin treatment (N=8). Data from 7 mice from 3 different EAEs (baseline: n=5, spadin and isotype: n= 7, spadin and anti-ICAM1: n=6)". What does the N=8 refer to? If N=8 mice, how is this possible from "7 mice from 3 different EAEs", which should be 7X3=21 mice? Does baseline refer to the Kcnk2^{fl/fl}xTie2^{cre} + ctr group? The Figure legend does not correspond to the graph shown in Fig. 7h and the number of the conditional KO mice used in the study is not disclosed.
9. Which method of statistical analysis was used for the EAE experiment in Fig 7h? P values need to be indicated for every data point that is statistically significant.
10. Supplementary Figure 2c: The Western blot image is of very poor quality and needs to be replaced with a high quality image. Furthermore, the western blot image is not representative of the quantification shown.

Reviewer #3

(Remarks to the Author)

The authors have satisfactorily addressed all my comments. This is an interesting and relevant scientific manuscript performed in a careful and meticulous manner. Therefore, I recommend the revised manuscript version for publication in Nature Communications.

Reviewer #4

(Remarks to the Author)

The manuscript NCOMMS-21-01217A by Bock et al., entitled « K2P2.1 shapes morphology and function of brain endothelial 1 cells via actin network remodeling », deepens the conclusions that these authors have made in Bittner et al., Nature Medicine 2013. Blood Brain Barrier (BBB) dysfunction is a common feature in several neurological diseases. Whereas previously the authors demonstrated that TREK1 potassium channel (K2P2.1, encoded by Kcnk2 gene) is expressed in mouse brain microvascular endothelial cells (MBMECs) and that either the genetic deletion of Kcnk2 in mice or the pharmacological inhibition of K2P2.1 by Spadin facilitates lymphocyte migration into the CNS by upregulating cellular adhesion molecules ICAM1 and VCAM1, revealing a role for TREK1 in the Blood Brain Barrier (BBB) function, the present paper describes the molecular events leading to it.

Notably the authors described the detailed molecular pathways whereby TREK1 modulates BBB permeability.

The authors made a good work in response to the comments

The new version of the paper is clear and easy to understand making it accessible for general audience, the experiments are well designed and the quality of the results is now really high.

Because of the pathologic states induced by dysfunction in BBB, this study is of general interest by characterizing TREK1 function on BBB maintenance. Together the results support TREK1 as a promising drugable target to treat such diseases.

The authors responded well to the concerns rose during the first round of reviewing, notably:

In the first version of their manuscript, the authors made use of different imaging techniques to investigate TREK1 localization and actin cytoskeleton characterization but some images for quantification were missing or not of optimal quality. They now fully completed their data by providing pictures with a better resolution and by adding quantifications. They also improved the clarity of the procedures and reinforced their conclusions by including new controls in their analyses. In addition, they performed new experiments (proteomics, generation and analysis of mutant MBMEC lines, use of conditional KO mice for Kcnk2) that further document the role of TREK1 in cytoskeleton regulation.

Specific minor comments are listed below :

- Results section, Title 1st paragraph : « down-regulation of Kcnk2 gene expression... » instead of « modulation of K2P2.1 function » would be more appropriate.

Reviewer #5

(Remarks to the Author)

The manuscript presented by Bock et al., entitled " K2P2.1 shapes morphology and function of brain endothelial cells via actin network remodeling", describes a study which includes proteomics analyses by data-independent acquisition (DIA) mass spectrometry (MS) using an Orbitrap Exploris 480 instrument and variable size precursor isolation windows. The proteomics experiments and results are well described. Please be sure to release the PRIDE repository with identifier PXD031051, as it is not yet public.

Reviewer #6

(Remarks to the Author)

K2P2.1 shapes morphology and function of brain endothelial 1 cells via actin network remodeling.

Bock et al.

The manuscript NCOMMS-21-01217A by Bock et al., entitled « K2P2.1 shapes morphology and function of brain endothelial 1 cells via actin network remodeling », deepens the conclusions that these authors have made in Bittner et al., Nature Medicine 2013. Blood Brain Barrier (BBB) dysfunction is a common feature in several neurological diseases. Whereas previously the authors demonstrated that TREK1 potassium channel (K2P2.1, encoded by Kcnk2 gene) is expressed in mouse brain microvascular endothelial cells (MBMECs) and that either the genetic deletion of Kcnk2 in mice or the pharmacological inhibition of K2P2.1 by Spadin facilitates lymphocyte migration into the CNS by upregulating cellular adhesion molecules ICAM1 and VCAM1, revealing a role for TREK1 in the Blood Brain Barrier (BBB) function, the present paper describes the molecular events leading to it.

Notably the authors described the detailed molecular pathways whereby TREK1 modulates BBB permeability.

The authors made a good work in response to the comments

The new version of the paper is clear and easy to understand making it accessible for general audience, the experiments are well designed and the quality of the results is now really high.

Because of the pathologic states induced by dysfunction in BBB, this study is of general interest by characterizing TREK1 function on BBB maintenance. Together the results support TREK1 as a promising drugable target to treat such diseases.

The authors responded well to the concerns rose during the first round of reviewing, notably:

In the first version of their manuscript, the authors made use of different imaging techniques to investigate TREK1 localization and actin cytoskeleton characterization but some images for quantification were missing or not of optimal quality. They now fully completed their data by providing pictures with a better resolution and by adding quantifications. They also improved the clarity of the procedures and reinforced their conclusions by including new controls in their analyses. In addition, they performed new experiments (proteomics, generation and analysis of mutant MBMEC lines, use of conditional KO mice for Kcnk2) that further document the role of TREK1 in cytoskeleton regulation.

Specific minor comments are listed below :

- Results section, Title 1st paragraph : « down-regulation of Kcnk2 gene expression... » instead of « modulation of K2P2.1 function » would be more appropriate.

Version 2:

Reviewer comments:

Reviewer #1

(Remarks to the Author)

I appreciate the efforts that the authors have put in the revised manuscript (# NCOMMS-21-01217B-Z) to address the concerns raised by the reviewers.

1. It is great to see that the authors have addressed the concern from Reviewer # 1 and have used the siRNA knockdown of Cfl1 in primary mouse brain endothelial cells for their experiments in Figure 7 which have strengthened the conclusions of the study since they have moved away from the bEnd3.0 endothelioma cell line.

2. In addition, the authors have done an excellent job validating some of the actin regulatory factors in control and EAE WT and *Kcn2fl/fl Tie2-Cre* mice.

3. The concern raised by Reviewer # 2 about the "meaning" of N versus n in many figure legends still remains unresolved. The authors should make the effort to include a couple of additional sentences in each figure legends to explain what N or n means rather than making the reader guess. In addition the statistical method used for the analysis should be put in the respective figure legends. This will improve the understanding of the data described in the respective figures.

4. Statistics concerns raised by Reviewer # 2 still remains an issue. I feel that the statistics are overinflated if they are performed on the n, rather than N (biological replicates). This seems to be the case for Figs 3d, 4e, 4f, 5f, 6b, 6c, 6e, S2b, S2c. The authors should perform the statistical analysis on biological, not technical, replicates. The technical replicates measure the same sample multiple times, assessing experimental variability, while biological replicates measure different samples, capturing biological variation which is used for statistical analysis.

Reviewer #3

(Remarks to the Author)

The revised manuscript addresses the concerns raised by several reviewers. The amount of data is impressive, experiments are well performed and controlled, imaging data is of good quality, and results convincingly support the main authors conclusions. Issues raised by Reviewer #2 regarding proper statistics and reproducibility has been satisfactorily addressed. The authors provided a comprehensive table detailing the manuscript statistics used in every figure. Altogether, the results presented in this study are of high-quality and convincing, thus I recommend acceptance for publication of this revised manuscript version.

Reviewer #4

(Remarks to the Author)

Regarding the number of replicates and statistical analyses:

The authors have provided a detailed response to the statistical concerns, with appropriate clarification regarding the distinction between biological (N) and technical (n) replicates.

They state that statistical analyses have been recalculated based solely on biological replicates, which aligns with standard practice for ensuring valid inference. Sample sizes are explicitly reported, and the statistical methods used are now documented in the manuscript and supplementary materials.

In experiments involving single-cell measurements, the authors justify their approach by referencing established literature employing similar methodologies. While this aligns with common practices in the field, it is worth noting that in some cases, a large number of measurements were obtained from a relatively small number of biological preparations. This raises potential concerns about pseudoreplication, particularly if statistical independence was assumed at the level of individual cells rather than the biological preparation. Although the authors acknowledge this issue and cite relevant precedent, it would be helpful to provide further clarification on how the statistical analyses accounted for this nested data structure. That said, their engagement with this concern reflects a thoughtful and responsible approach to data interpretation.

Additionally, in some experiments where the number of biological replicates is low (e.g., N=4–7), a brief discussion on the statistical power to detect meaningful effects may help contextualize the reliability of the findings.

Overall, the revisions thoughtfully address the main statistical concerns. While a few minor uncertainties remain, especially regarding the treatment of nested data and the robustness of inferences drawn from small sample sizes, the statistical reasoning is generally sound. The response is well considered and, with a few additional clarifications, could be further strengthened. To conclude, the authors have clearly made thoughtful efforts that have significantly enhanced the rigor of their work.

Point to point reply: NCOMMS-21-01217A (“K_{2p}2.1 shapes morphology and function of brain endothelial cells via actin network remodeling” by Bock et al.)

REVIEWER COMMENTS

Reviewer #1 (Remarks to the Author):

The manuscript # NCOMMS-21-01217-T by Bock et al., is an extension of a previous study that demonstrated a critical role for the K_{2p} 2.1 potassium channel in EAE. Previously, it has been shown that in *Kcnk2*^{-/-} mice, brain endothelial cells upregulate cellular adhesion molecules ICAM1, VCAM1 and PECAM1 and facilitated leukocyte trafficking into the CNS. Following the induction of experimental autoimmune encephalomyelitis (EAE) by immunization with a myelin oligodendrocyte protein (MOG)₃₅₋₅₅ peptide, *Kcnk2*^{-/-} mice showed higher EAE severity scores that were accompanied by increased cellular infiltrates in the central nervous system (CNS).

In this study, the authors use cell biological approaches in purified mouse BMECs (mBMECs) from wild-type or *Kcnk2*^{-/-} to analyze the cell distribution of the protein under healthy or inflammatory conditions and find that the mutant cells have active stress fibers. The authors perform a differential screen between the wild-type and *Kcnk2*^{-/-} cells and identify several actin regulatory proteins (e.g. Cfl1) that change in mutant cells. Moreover, they use an in vitro system to demonstrate that T cells have a higher interaction force in mutant versus wild-type cells and under inflammatory conditions. Finally, the authors use an adoptive transfer approach to look at T cell trafficking across the CNS blood vessels under treatment with Spadin, an inhibitor of K_{2p} 2.1 potassium channel and ICAM-1 inhibition and show that Spadin treatment enhances T cell trafficking into the CNS and ICAM antibody reverses the effect.

The study is performed carefully and provides some additional insights into the cell biology of the K_{2p} 2.1 potassium channel in endothelial cells and how it may affect interactions with the immune cells during neuro-inflammation. However, the study seems like an extension of the previous work published on this receptor and the work is primarily focused on in vitro studies in mBMECs. Therefore it does not provide a major conceptual breakthrough regarding the role of this receptor in neuroinflammation but rather an incremental information. Moreover, the downstream effectors that link K_{2p} 2.1 potassium channel with the actin cytoskeleton are not explored in detail. Following are some additional concerns as follows:

1) In Figure 3, the authors identify some of the downstream targets for K_{2p} 2.1 potassium channel in mBMECs. The authors need to show images for quantification performed in Panel d and perform Western blotting to determine the ratio of phosphorylated versus non-phosphorylated forms of Cfl1 in endothelial cells.

We apologize for not including the images in the manuscript. We updated the figure accordingly (**Figure 4c, d**) and performed additional Western blotting to quantify Cfl1 and pCfl1 in untreated and inflamed MBMECs (**Supplementary Figure 4c**). The quantification of the Western blot also shows an increase in Cfl1 in *Kcnk2*^{-/-} MBMECs compared to WT MBMECs. However, the increase in Cfl1 under inflammatory conditions in WT MBMECs is more prominent in immunofluorescence staining, compared to Western blot.

2) The authors need to perform also in vivo validation with these targets to show that these targets are upregulated under inflammation (EAE) or in *Kcnk2*^{-/-} cells. The validation of the screen between the wild-type and mutant cells in vivo is critical since the mBMECs are very different to in vivo brain endothelial cells.

We thank the reviewer for the excellent comment. Indeed, in vivo brain endothelial cells differ from cultured MBMECs. To address this concern, we performed immediate ex vivo analyses with isolated MBMECs.

We induced EAE in WT and *Kcnk2*^{-/-} mice to validate the targets. Consistent with previous results¹, *Kcnk2*^{-/-} mice developed worse EAE symptoms than WT mice (**Supplementary Figure 1a**). We isolated MBMECs from naïve mice and EAE mice at the peak of disease (day 15) and analyzed the proteome of these cells (**Figure 4a, b; Supplementary Figure 3, Figure for the reviewer 1**). Under inflammatory conditions, differentially expressed proteins were mostly related to cellular and metabolic processes due to inflammation. There was a large overlap between WT and *Kcnk2*^{-/-} and only a small group of proteins was related to cytoskeletal rearrangements (**Supplementary Figure 3**). However, comparing WT and *Kcnk2*^{-/-} brain endothelial cells from naïve mice revealed major changes in pathways regulating the cytoskeletal organization (**Figure 4a**). To eliminate the bias from inflammation, we focused further evaluations on naïve conditions. A deeper analysis of differentially regulated proteins demonstrated regulation of proteins involved in actin binding, stress fiber organization, cell membrane organization and actin-based cell projections (**Figure 4b**). Additionally, proteins involved in extracellular matrix organization and metabolic processes were also altered in *Kcnk2*^{-/-} endothelial cells.

Other than at the transcriptomic level, the total amount of Cfl1 was not altered. However, analysis of levels of phosphorylated and dephosphorylated Cfl1 by immunofluorescence and Western blot showed a shift towards more dephosphorylated Cfl1 in *Kcnk2*^{-/-} MBMECs. As discussed in our statement for the editor, we aimed to perform phospho-proteomics in MBMECs from naïve and EAE mice, but we were not able to acquire a sufficient amount of protein. We pooled three mice for each replicate in the proteome analysis to get 5 µg of protein. In our setup, a minimum of 25 µg protein would have been required to generate robust data on protein phosphorylation. Overall, combining analyses of transcriptome, proteome and Cfl1 phosphorylation by immunofluorescence and Western blot analyses supported a clear link between K₂P_{2.1} and the actin cytoskeleton in brain microvascular endothelial cells and the critical role of actin depolymerization factors in this interaction.

Figure for the reviewer 1. Venn diagram of differentially expressed proteins in mouse brain microvascular endothelial cells from WT and *Kcnk2*^{-/-} mice with EAE at peak of disease and naïve *Kcnk2*^{-/-} mice plotted against naïve WT mice. The numbers of proteins are indicated within the respective circles. (N=6, n=18)

3) The in vitro studies of T cell interactions with the wild-type and *Kcnk2*^{-/-} cells also need to be performed with cells lacking some of the downstream intermediates such as *Cfl1* to demonstrate a link between the K2p 2.1 potassium channel and the actin regulation and provide a clear cell biological mechanisms for the activity of K2p 2.1 potassium channel.

We greatly appreciate this valuable point. Indeed, it would be highly interesting to investigate the effects of knocking out downstream targets and especially *Cfl1*. Unfortunately, *Cfl1* knock-out mice are not viable^{2,3}. To address this concern, we decided to use the bEnd.3 endothelial cell line⁴ and generated stable *Kcnk2* and *Cfl1* single- and double-knock-out cell lines (**Supplementary Figure 5**) by CRISPR/Cas9 technology. However, *Cfl1* full-knock-out cells were also not viable. Therefore, we used heterozygous *Cfl1*^{+/-} cells for studying the effect of *Cfl1* and *Kcnk2* single- and double-knock-out on brain endothelial cells. The *Kcnk2*^{-/-}, *Cfl1*^{+/-} and *Cfl1*^{+/-}*Kcnk2*^{-/-} bEnd.3 cells showed morphological alterations after stably inducing the knockout (**Supplementary Figure 6**). They were larger than bEnd.3-WT cells and showed an enhanced formation of protrusions and filopodia.

Coculture experiments with stimulated CD4⁺ T cells from WT mice under flow conditions (0.25 dyn/cm²) revealed an increase in T cell adhesion to *Kcnk2*^{-/-}, *Cfl1*^{+/-} and *Cfl1*^{+/-}*Kcnk2*^{-/-} bEnd.3 cells compared to bEnd.3-WT cells already under basal conditions (**Figure 6f**). These data indicate that knock-out of K2p2.1 has a similar effect on T cell adhesion as knock-out of *Cfl1*. Knock-out of both, *Cfl1* and K2p2.1, had no additional effect on T cell adhesion. These data provide clear evidence for an involvement of *Cfl1* in the downstream effects of the knock-out or downregulation of K2p2.1. Corresponding paragraphs were added to the results and discussion section (see p 12, l 281-291; p 16, l 381-391).

4) The resolution of images in Figures 1, 2 needs to be higher. The author needs to show the location of insets and provide confocal images with all the profile views in all three dimensions.

As suggested, we improved the resolution and presentation of the images in Figure 1 and 2 (**Figure 1c, Figure 2d**). Therefore, we repeated all the staining and acquired new confocal images. For better visualization of the membrane protrusions and localization of ICAM1, we included the orthogonal views and a 3D profile view of the staining for ICAM1, K_{2P}2.1 and actin (**Figure 2d, e**). The new images and analyses provide additional support for the presence of membrane protrusions in the absence of K_{2P}2.1 and after inflammation of WT cells.

5) The authors need to show real images and movies in Figure 6 rather than animated movies.

A movie exemplifying 2-photon experiments can now be found in the supplemental materials (**supplementary video 1**).

6) Figure 6d needs to show also the ICAM inhibition.

We thank the reviewer for this remark. We added the image showing T cells tracks during spadin-treatment and ICAM1 inhibition to the figure (**Figure 7d**).

Overall, we thank the reviewer for his/her critical revision and highly valuable comments that significantly improved our manuscript.

Reviewer #2 (Remarks to the Author):

The manuscript by Bock et al studies the role of the kcnk2 potassium channel in brain endothelial cell morphology and leukocyte transmigration. The study uses a combination of in vitro and in vivo approaches to determine the role of kcnk2 in endothelial cell responses to proinflammatory stimuli and T cell migration. The study of potassium channels in the regulation of BBB functions is an important area of research. The manuscript provides several mechanistic insights using standard cell biology methods regarding changes in subcellular localization of kcnk2 upon TNF stimulation, redistribution of ICAM-1, and phosphorylation rates of cofilin. The major concern with the manuscript is the very limited in vivo validation of these findings that is preliminary with numerous technical limitations. As a result, whether kcnk2 plays an in vivo role in the endothelial cells remains largely unknown. Specific comments are listed below.

1. Analysis of EAE by in vivo two photon microscopy focuses on the brainstem, which is not the main site of T cell transmigration. Instead two photon imaging should have been performed in the spinal cord and the migratory patterns of multiple T cells need to be imaged and analyzed.

We thank the reviewer for this important remark. However, we only partly agree with this assessment as outlined below. First, to better illustrate the experimental procedures and the setup of intravital two-photon imaging (2PM), we refer to **Figure for the reviewer 2**. A total of 7 mice from 3 different EAE experiments were used for 2PM. As carotid artery catheter operation as well as imaging window preparation are associated with various difficulties, and to exclude temporal effects on extravasation rate, the sequence *baseline recording without*

spadin -> administration of *spadin* with isotype-AB -> administration of α ICAM-AB was not performed in all mice, but some mice started with *spadin* directly. Nevertheless, each sequence was analysed in the same ROI to exclude spatial bias.

Figure for the reviewer 2: A flowchart illustrating the experimental procedures and the setup of intravital two-photon imaging. Abbreviations: ROI, region of interest.

We fully agree that there are several other (e.g. thoracic spinal cord and cervical spinal cord¹⁰⁻¹²) established surgical windows for T cell tracking in mouse blood vessels besides our 2PM ROI (brainstem/medulla oblongata), each of which has certain advantages and disadvantages. In the last decade, our lab has published more than a dozen studies with exactly this method used here¹³⁻¹⁷ (**Figure for the reviewer 2**). Nonetheless to address the reviewer's concern, we performed an additional immunohistochemical analysis comparing the lesion area in our ROI with the thoracic spinal cord in EAE diseased mice (see schematic in **Figure for the reviewer 3 a, b**). We found a trend towards more infiltrating immune cells in the thoracic spinal cord, but this did not reach the significance level. We confirmed these results with a second method by examining the absolute number of CD4⁺ T cells in the parenchyma of our ROI compared to the rest of the spinal cord using FACS. The quantification (**Figure for the reviewer 3c**) clearly showed a cluster of infiltrated T cells in our ROI, again supporting our 2PM method.

Thus, we were able to prove that T cell trafficking into the CNS does indeed occur in our 2PM ROI, which should serve – and we hope the reviewer agrees – as a sufficient prerequisite for studying this process under the influence of K_{2p}2.1 modulators. Whether there are spatial differences in the kinetics of immune cell extravasation depending on the surgical window used was not the subject of the current study.

Figure for the reviewer 3: (a) Scheme explaining the experimental procedure for immunohistochemical analysis of the lesion area in cross-sections (hematoxylin-eosin staining) of mice suffering from EAE with two representative examples from the brainstem (ROI, left, white) and thoracic spinal cord (SC, right, grey). Cell infiltration was digitally marked by hand and lesion area was determined by the percentage of total parenchyma. Scale bars: 1000 μ m. **(b)** Quantification of lesions area as described in B (t test, $p = 0.405$, $n = 4$ mice from 2 different EAE experiments). Abbreviations: ROI, region of interest; SC, spinal cord. **(c)** The part of the brainstem imaged in the intravital two-photon microscopy setup (ROI) and the spinal cord (SC) of EAE diseased mice were collected and weighted separately and the number of T cells/mg parenchyma ($CD45^+CD3^+CD4^+$ cells) was analysed by FACS after exclusion of doublets and dead cells. Manual counting of all living cells already infiltrating these two parts (ROI and SC) using trypan blue during the workup procedure enabled the calculation of absolute numbers later in flow cytometry (t test, $p = 0.061$, $n = 9$ mice from 2 different EAEs). Abbreviations: ROI, region of interest; SC, spinal cord.

2. Labeling of blood vessel lumen using dextran cannot be used to document “endothelial crossing”. Instead, endothelial cells need to be labeled with fluorescent reporters to document “endothelial crossing”.

We thank the reviewer for this valuable comment. As suggested, we have updated the manuscript to refer to the process in question as “extravasation” of T cells. We would like to emphasize that the use of fluorescent dextran is a common method to analyze exactly this process ^{10,18–23}.

To reinforce this point, we performed 2PM experiments using both an anti-CD105 antibody ^{10,18,24} to visualize the endothelial layer and rhodamine dextran to visualize the blood. The 3D reconstructions depicted in the **Figure for the reviewer 4** shows the blood closely connected to the endothelial layer with no space in between.

Figure for the reviewer 4: Representative 3D reconstructions from intravital two-photon microscopy scans showing that the blood volume in red is tightly associated with the endothelial layer reconstruction in green. Microscopy was performed with EAE diseased mice after intra-arterial injection of FITC dextran to visualise the intraluminal blood volume (red) and Alexa Fluor® 594 anti-mouse CD105 antibody to visualise the endothelial layer (green). Scale bars: 70 μm .

3. Spadin is administered systemically, therefore it is not possible to determine if the observed effects are due to endothelial expression of *Kcnk2*. Can *Kcnk2* be blocked specifically in endothelial cells?

We fully agree that this claim needed further experimental support. Following the suggestion, we performed two further EAE experiments with mice specifically lacking *Kcnk2* in endothelial cells by mating B6.Cg-Tg(Tek-cre)1Ywa/J (Tie2-cre) mice with newly generated loxP-flanked *K_{2P}2.1* mice (*Kcnk2^{fl/fl}*). Mice lacking Cre recombinase expression or loxP sites were used as controls. Endothelial cell specific *Kcnk2^{-/-}* and control animals were treated daily with 100 μl 0.9% NaCl or 100 μl 10 μM spadin in NaCl i.p. Control mice treated with spadin, as well as endothelial cell specific *Kcnk2^{-/-}* mice treated both with spadin and NaCl developed a worse EAE disease course than control mice treated with NaCl (**Figure 7h**). These results demonstrate that *K_{2P}2.1* expression on endothelial cells is crucial for the effect mediated by spadin.

4. The analysis of BBB needs to include tight junction and plasma proteins. Is there increase only in crossing of cells or increased leaks of the BBB?

The impact of *K_{2P}2.1* on BBB integrity and permeability has been already investigated in detail in our previous study ¹. The tight junction proteins ZO-1, claudin-5 and occludin demonstrated no differential expression between WT, *Kcnk2^{-/-}* under naïve and EAE conditions (**Figure for the reviewer 5a**). Additionally, barrier properties were unaffected by *Kcnk2^{-/-}* as shown by trans-endothelial resistance (TER) measurements (**Figure for the reviewer 5b**) and Evans blue injection and extravasation into the CNS (**Figure for the reviewer 5c**).

Figure for the reviewer 5. (a) Relative mRNA expression of ZO-1, Claudin-5 and Occludin in MBMECs from naive WT and *Kcnk2*^{-/-} mice and from WT and *Kcnk2*^{-/-} mice with EAE. Expression values were normalized to naive WT MBMECs. (b) Capacity (left panel) and trans-endothelial resistance (TER) in WT and *Kcnk2*^{-/-} MBMECs. (c) Extravasation of Evans blue into the CNS 24 h after i.v. injection into naive WT and *Kcnk2*^{-/-} mice and EAE mice (left panel). Quantification of Evans blue extravasation into brain (upper graph) and spinal cord (lower graph). Figures from: Bittner S, Ruck T, et al. Endothelial TWIK-related potassium channel-1 (TREK1) regulates immune-cell trafficking into the CNS. *Nat Med.* 2013 Sep;19(9):1161-5. doi: 10.1038/nm.3303. Epub 2013 Aug 11. PMID: 23933981.¹

5. Spadin is administered acutely. What are the effects on clinical signs and histopathology in the spinal cord in the EAE model after chronic administration of spadin or other pharmacologic or genetic approaches to deplete *kcnk2* in endothelial cells?

As already shown in our previous study¹, spadin significantly worsened the EAE disease course in WT but didn't have any additional effect on *Kcnk2*^{-/-} mice. We confirmed the data from the previous publication in endothelial cell-specific *Kcnk2*^{-/-} mice. Therefore, we generated a *Kcnk2*-floxed mouse and crossed these mice with *Tie2*-cre mice. As described above, we treated the mice for 35 days with spadin (i.p.). Of note, spadin administration worsened

the EAE disease course exclusively in WT (*Kcnk2^{fl/fl}xTie2^{cre-}*) and not in *Kcnk2*-knockout (*Kcnk2^{fl/fl}xTie2^{cre+}*) mice (see **Figure 7h**).

6. The EAE experiment is missing controls using Ova-T cells. As only T cells from 2D2 mice are injected, it is unknown whether these responses are specific to encephalitogenic T cells from 2D2 mice or are T cell not recognizing CNS antigens. An Ova T cell control needs to be included. The choice of imaging a single T cell in the brainstem, a region with low T cell infiltration, further impedes addressing these questions.

We greatly appreciate this insightful comment. It has already been reported that administration of OT-II Th17 cells in *Rag^{-/-}* resulted in hardly any T cell infiltration into the CNS. Co-transfer of OT-II Th17 cells with 2d2 cells resulted in increased OT-II T cell infiltration; however, in comparison to 2d2 T cells only 1/20 of OT-II T cells infiltrate the CNS T cells ²⁵, making visualization of the infiltration process challenging. Nevertheless, to investigate whether OT-II T cell infiltration is also affected by spadin treatment, we repeated the in vivo application of spadin and analyzed OT-II.RFP T cell infiltration compared to 2d2.CFP T cell infiltration of the CNS of EAE diseased *Rag^{-/-}* mice. After spadin treatment, we only observed an increase in the absolute cell number of 2d2 T cells, but not of OT-II T Cells in the CNS (**Figure for the reviewer 6**).

Figure for the reviewer 6: Passive EAE was induced by transferring B6.2D2 Th17 cells into *Rag2^{-/-}γc^{-/-}* mice. Each 5×10^6 RFP positive B6OT.Th17 cells together with 5×10^6 CFP positive B6.2D2.Th17 cells were subsequently injected intravenously at disease onset. Mice were treated intraperitoneally with 100 μ l of 10 μ M spadin (treatment group) or 100 μ l NaCl (control group) daily. After 3 days, the whole CNS was harvested and the number of 2D2 (left panel, t test, $p = 0.049$) and OT2 (right panel, t test, $p = 0.9861$) Th17 cells infiltrating the CNS was analysed by FACS. As before, manual counting of all live cells that had already infiltrated the CNS using Trypan Blue during the workup procedure allowed the absolute numbers to be calculated later in flow cytometry (Spadin: $n = 5$ mice; NaCl: $n = 4$ mice each from 2 different EAEs).

7. ICAM-1 deficiency has relatively mild protective effects in EAE and it is unclear whether regulating ICAM-1 alone would be sufficient to modulate trafficking. Are there other molecules regulated by *kcnk2*?

We also quantified the amount of VCAM1 expression in WT and *Kcnk2*^{-/-} MBMECs. Other than ICAM1, VCAM1 expression was not regulated in *Kcnk2*^{-/-} MBMECs (**Figure 2c**). Our data suggest that the ICAM1-mediated effect in *Kcnk2*^{-/-} MBMECs is due to the increase in docking structures. The *Kcnk2*^{-/-} cells show an increase in membrane protrusions in addition to enhanced ICAM1 expression and clustering on those protrusions as shown in **Figure 2d, e**.

8. The number of biological replicates used in the study is not stated. For example in Figure 6, it is mentioned that data are from “n=3-6 EAEs”. Are these mice, vessels, or cells? If they are mice, how many vessels have been sampled per mouse? As the variability in Fig. 6g is very high, are the reported differences statistically significant? Have power calculations been performed for Fig. 6e? It appears that n=3 is not sufficient to draw any conclusions given the large variability. If n represents the number of cells from the same mouse, then the study needs to be repeated in at least n=8 EAE mice and the number of mice needs to be used for statistical analysis as an independent biological replicate.

We apologize for this lack of clarity. As mentioned above, we have now updated the manuscript and depicted the procedure in more detail in the **Figure for the reviewer 2** to clarify this point. We used 7 mice from 3 different EAES. For baseline recording we used 5 mice, spadin treatment and application of the isotype control antibody was performed with 7 mice and spadin treatment with anti-ICAM1 neutralizing antibody was recorded in 6 mice. Further, we increased the number of experiments and now provide a detailed description on biological replicates in the corresponding figure legends (**new Figure 7**).

We gratefully thank the reviewer for his/her constructive and insightful feedback.

Reviewer #3 (Remarks to the Author):

Reviewer report on “K2P2.1 shapes morphology and function of brain endothelial cells via actin network remodeling” by S. Bock et al. (2021)

In this manuscript the authors report a novel role of K2P2.1, a two-pore domain mechanosensitive potassium channel which express in brain microvascular endothelial cells and is an important regulator of immune cell trafficking/transmigration across the blood-brain barrier (BBB). The manuscript explores the role of K2P2.1 in modulating endothelial cells morphology and filamentous actin cytoskeleton remodeling. The authors show that reduction in K2P2.1 expression in microvascular endothelial cells results in altered morphology, increased in membrane protrusions, and increased formation in stress fibers with K2P2.1 channels colocalizing with stress fibers. In addition, the authors showed that absence of K2P2.1 results in remodeling of the actin cytoskeleton with increase in stress fibers formation and enhanced cellular cortical stiffness. Furthermore, the authors observe that activation of key filamentous actin regulating proteins Cofilin1 and Arp2/3 complex. Then, the authors show in vitro enhanced CD4+ T cell adhesion and morphological changes on *Kcnk2* depleted microvascular endothelial cells. Finally, they perform in vivo experiments and observe increased immune CD4+ T cell adhesion and transmigration in mouse brain stem blood vessels when adding Spadin a pharmacological drug that causes inhibition of K2P2.1. The experiments seem to be performed carefully, and in my

view the manuscript shows novel and relevant results. However, there are some aspects as data quantification and some clarifications that the authors need to address.

Broadly speaking, I see a clear contribution of this paper towards understanding mechanosensitive regulators of immune cell trafficking across the blood-brain barrier essential for central nervous system immune and pharmacological therapeutics. The presented research work is interesting, and I feel is of broad interest to the Nature Communications readership and I may recommend its publication after the authors address the following comments:

1) Lines 98 to 100; the sentence “The main determinants for cellular mechanical properties...” will be significantly improved if references are provided/added to this sentence supporting that the actin cytoskeleton and stress fibers are the main determinant of cellular mechanical properties.

We thank the reviewer for this important advice. We now provide references to this sentence. The determinants of mechanical properties, especially in vascular endothelial cells, are nicely described and reviewed in ²⁶⁻²⁹.

2) Lines 159 to 164 and Figure 1e; the cortical stiffness is determined, however the authors report a normalized cortical stiffness instead of the true estimated value. In order to provide the reader a real comparative sense of the stiffness behavior of the endothelial cells it would be great if the authors provide the real stiffness values.

We thank the reviewer for this suggestion. We thought presenting normalized stiffness values could be more intuitive for the broad readership of Nature communications. However, we were also discussing this point in advance (Figure for the reviewer 7). We exchanged the graph and provided real stiffness values for better comparison between the groups in Figure 1f.

Figure for the reviewer 7: Cortical stiffness of untreated and inflamed WT and Kcnk2^{-/-} MBMECs provided as normalized stiffness values (a) and true estimated values (b).

3) Furthermore, following up my previous comment; How the authors extract the stiffness values from the acquired AFM force-distance curves? Do the authors perform a linear regression to fit the force distance curves thus the obtained slope is the measured apparent stiffness? Is the

estimated stiffness in N/m units? The authors fail to properly describe in the Materials and Methods how these calculations were made (in lines 403-405 the authors state the use a nano-indentation analysis software named Punias 3D).

We apologize for this lack of clarity. We amended the respective materials and methods section by the following to clarify the methods (see p 20, 1477-484):

We analyzed the slope of the force distance curves that gives direct information about the force (in pN), which is needed to indent the cell for a given distance (in nm). Here, the force is defined as cell “stiffness”: the stiffer the sample, the higher the deflection of the cantilever, i.e. the steeper the slope of the force distance curve. The contact point of the cantilever was defined by performing a baseline fit. Starting from this point, the first slope of the force distance curves has been analyzed, which could independently be shown to reflect the cell cortex. The analysis was performed using the Protein Unfolding and Nano-Indentation Analysis Software (PUNIAS), using the mode for “nanoindentation”³⁰.

4) Lines 215 to 217; I suggest the authors include the ARP2/3 complex data in the main text Figure 3 since this observation is one of the main findings and an important component of the proposed model in Figure 7.

We agree with the reviewer and updated the figures accordingly. Arp2/3 expression levels are now depicted in **Figure 3d**.

5) Figure 6; statistical analysis and significance labels are missing on panels Figures 6e and 6g. The authors should perform the statistical analysis and add the significance labels.

We thank the reviewer for this advice. We increased the number of replicates in **Figure 7e** (old Figure 6). The significance labels are now depicted in the figures.

6) Supplementary Figure 2; statistical analysis and significance labels seems to be missing in presented data for *Arpc3* and *Arpc4* in Supplementary Fig. 2C.

Due to the low number of replicates the difference in *Arpc3* and *Arpc4* expression is not statistically significant. The p-value for *Arpc3* WT uninflamed and KO uninflamed is 0.155, and after 30 min of inflammation 0.078. The p-value for *Arpc4* 6h after inflammation between WT and KO is 0.77. Unfortunately, we were not able to increase the number of replicates due to limited mouse numbers and breeding restrictions following Covid-19 pandemic and relocation of our lab to a new city.

We greatly appreciate the very positive feedback from the reviewer and are thankful for the important suggestions.

Reviewer #4 (Remarks to the Author):

Comments:

The manuscript by Bock et al. explores the role of the two-pore domain potassium channel K2P2.1 (TREK1) in the immune cell trafficking into the CNS through a number of cell biology and imaging experiments. The key conclusion by the authors is that K2P2.1 mediates the immune cell transmigration across the blood-brain barrier by influencing the actin cytoskeleton structures, completing and deepening the study published by Bittner et al, Nature Medicine 2013.

Originality and significance

The originality of this work lies on the completing the model described in the Nat Med 2013 in which TREK1 activity regulate the cell transmigration through the BBB. This involving regulation of cytoskeleton and actin dynamics through the mechanosensitive K2P2.1 channel.

Although the observations made by the authors are potentially interesting, the analyses provided appear incomplete and more rigorous analyses would be critical to a better understanding of the role of K2P2.1 in BBB and a better emphasizing of the novelty of this work.

Major concerns/points for improvements are as follows:

The authors used an appropriate battery of tools for demonstrate their data, including immunofluorescence staining, atomic force microscope, single cell force spectroscopy proximity ligand binding assays and in vivo two-photon scanning and the utilization of specific knock-out mice.

1. Figure 1b and 2b. K2P2.1 shares high homology with other members of the same family, making the antibody unspecific. Moreover, some of this members are also mechanosensitive. I suggest to the authors adding a negative control using cells from KO animals for Fig 1B and Fig 2B. What is the labelling of the cells in KO mice. This would give an information regarding AB specificity. Moreover, it would interesting to make it with KO cells to see if clusters arranges similarly to WT cells upon inflammation.

We thank the reviewer for this important advice. As suggested, we also included untreated and inflamed *Kcnk2*^{-/-} MBMECs as staining controls. However, in *Kcnk2*^{-/-} MBMECs, residual staining of K_{2P}2.1 was observed, despite the signal intensity is much lower than in untreated WT conditions. The used K_{2P}2.1 primary antibody (Anti-TREK-1 antibody produced in rabbit, T6448, Sigma-Aldrich) recognizes an amino acid sequence at the N-terminal domain of K_{2P}2.1. The *Kcnk2* knock-out mouse has been generated by excision of exon 3 as described in ⁵. For K_{2P}2.1 and other members of the K_{2P} channel family, it has already been shown that truncated forms of the protein can still be expressed, even if there are excisions (by slice variants or genetic modifications) or frame-shift mutations in the gene sequence ⁶⁻⁹. For truncated forms of K_{2P}2.1, it has been shown, that they still can dimerize with full-length proteins, if co-expressed, but show overall reduce the surface expression of K_{2P}2.1 and dominant-negative effects on the channel ^{6,7}. If only truncated forms of K_{2P}2.1 are expressed (as in our case), the ion current is also abolished ⁷. Therefore, the N-terminal residue of the protein could still be translated and recognized by the antibody. Unfortunately, we were not able to exchange this primary antibody by an alternative one (e.g., TREK1-antibody (F-6), sc-398449, Santa Cruz Biotechnology), as this was the only antibody that showed robust signals in WT MBMECs in

all experimental settings (FACS, immunofluorescence, PLA). However, in the case of an un-specific binding to other K_{2P} channels, we would expect a similar signal intensity in the immunofluorescence staining in WT and *Kcnk2*^{-/-} cells. As we observed a reduced signal in immunofluorescence of *Kcnk2*^{-/-} MBMECs, the staining is most likely specific for the N-terminal domain of $K_{2P}2.1$. Consistent with this, we detect a clear shift of the signal of 2.5-log scales in WT MBMECs compared to *Kcnk2*^{-/-} (**Supplementary Figure 1b**) in flow cytometry using the same antibody. Further, we analyzed the mRNA expression of *Kcnk2* was additionally checked in naïve WT and *Kcnk2*^{-/-} mice and WT and *Kcnk2*^{-/-} EAE mice at the peak of disease (**Supplementary Figure 1c**). As described in **Figure 1a**, the expression of *Kcnk2* is significantly reduced under inflammatory conditions in WT MBMECs (*in vitro* and in MBMECs from EAE mice). These findings are consistent with our data on protein level. *Kcnk2* mRNA was not detected in *Kcnk2*^{-/-} mice. Additionally, we included MBMECs from *Kcnk2*^{-/-} mice as a control for the proximity ligation assay of PI(4,5)P₂ and $K_{2P}2.1$ in **Figure 5f**. Here, the signal from the PLA shows a clear negative result. This provides a strong indication, that even if the $K_{2P}2.1$ antibody recognizes the residual N-terminal domain of $K_{2P}2.1$, this part of the channel does not bind to PI(4,5)P₂ in the membrane. In summary these different experiments and controls provide evidence that the used antibody specifically recognizes $K_{2P}2.1$ and does not bind to un-specific targets, such as other K_{2P} channels.

The protrusion formation and ICAM1 clustering is similar in inflamed WT and *Kcnk2*^{-/-} MBMECs both, untreated and inflamed. Mechanistically, we propose the binding of Cfl1 to PI(4,5)P₂ due to downregulation and redistribution of $K_{2P}2.1$. This mechanism is supported by the fact that the colocalization of PI(4,5)P₂ and Cfl1 is increased in *Kcnk2*^{-/-} MBMECs even under untreated conditions as shown in **Figure 5d, e**.

2. qRT-PCR experiments indicate that $K_{2P}2.1$ mRNA is rapidly decreased (30 minutes) under inflammatory stimulus driven by IFN γ and TNF α , with a minimal expression level reached after 24 hours of treatment. Despite this striking decrease at 24h, the protein level of expression seems not to be reduced even this point is difficult to evaluate on the immunofluorescence staining. The authors should evaluate this discrepancy and quantify $K_{2P}2.1$ protein expression and turn-over in the different experimental conditions. Moreover, the authors mentioned that $K_{2P}2.1$ is expressed at the membrane, in the cytosol and close to the nucleus of the MBMECs and they conclude that $K_{2P}2.1$ protein distribution is altered by the IFN γ /TNF α treatment. To better characterize this channel redistribution, the authors should provide images with a better resolution.

We fully agree that it is hard to evaluate the exact protein amount of $K_{2P}2.1$ by immunofluorescence. Therefore, we established a flow cytometry staining for $K_{2P}2.1$. Alternatively, we quantified the protein expression of $K_{2P}2.1$ over time of inflammation by flow cytometry (**Figure 1b, Supplementary Figure 1b**). Consistent with the qRT-PCR data, we detected a decrease in $K_{2P}2.1$ protein expression levels in WT MBMECs over time following inflammation.

Additionally, we improved the resolution of the mentioned images depicted in **Figure 1c** and **Figure 2d**. Under untreated conditions, there is a signal for $K_{2P}2.1$ along actin filaments and also distributed throughout the whole cell (**Figure 1c**). After inflammation, the $K_{2P}2.1$ signal correlates more with the phalloidin staining of actin.

3. AFM experiments highlight the presence of protrusions both in wild-type MBMECs treated with IFN γ /TNF α and in *Kcnk2* mutant cells. Other markers should be used to confirm the presence of these structures. Are these protrusions ezrin-rich? The conclusion that loss of K2P2.1 leads to the formation of membrane protrusions with clusters of ICAM-1 is based on co-immunofluorescence analyses with ICAM1 and K2P2.1. Providing images with a better resolution will allow to evaluate this point. The authors should also deepen the analyze of the docking structures and investigate their different components such as ERM proteins, EBP50, filamin b, etc.

We thank the reviewer for this valuable comment. We improved the resolution of the immunofluorescence staining shown in **Figure 1c** and **2d**. In addition, we included the orthogonal views and a 3D reconstruction of the staining for ICAM1, K_{2P}2.1 and actin (phalloidin). The new images and analyses provide additional support for the presence of membrane protrusions in the absence of K_{2P}2.1 and after inflammation of WT cells. We also provided different assays investigating the adhesion of T cells to the different MBMEC cultures. An increased T cell migration behavior under static conditions in *Kcnk2*^{-/-} MBMECs and an enhanced adhesion force (measured by AFM) provides mechanistic evidence for the formation of membrane protrusions and an increase in “stickiness” of *Kcnk2*^{-/-} MBMECs (**Figure 6**). We also included adhesion assays under flow conditions using mutant bEnd3 cell lines having a knock-out for *Kcnk2* and *Cfl1* and a double knock-out. These results also demonstrate an increased T cell adhesion after knock-out of *Kcnk2*. Since there is already literature describing the role of the ERM proteins for membrane protrusions and immune cell docking structures, e.g. ³¹, we decided – and we hope the reviewer agrees - to focus on the direct role of K_{2P}2.1 on MBMEC morphology and function with a special focus on T cell adhesion and transmigration under (neuro-)inflammatory conditions.

4. To reinforce the main message of the manuscript and its novelty, the authors should provide a more detailed analysis of the molecular pathways whereby K2P2.1 modulates BBB permeability and ion flux-dependent or independent functions of this ion channel.

We greatly appreciate this suggestion. In addition to the analysis of the transcriptome of untreated and inflamed WT and *Kcnk2*^{-/-} MBMECs with a focus on cytoskeletal regulators (**Figure 3, Supplementary Figure 4a, b**) and a detailed analysis of the phosphorylation state of Cfl1 (**Figure 4c, d; Supplementary Figure 4c**), we performed proteomics of MBMECs from naïve WT and *Kcnk2*^{-/-} mice and from mice with EAE at peak of disease. As demonstrated in **Figure for the reviewer 1** and **Supplementary Figure 3**, there was a huge overlap of differentially expressed proteins in WT and *Kcnk2*^{-/-} mice with EAE due to inflammation. Therefore, we decided to focus on the differences between the naïve mice to avoid this strong bias. Comparison of WT and *Kcnk2*^{-/-} brain endothelial cells from naïve mice showed major changes in pathways regulating the cytoskeletal organization (**Figure 4a**). A deeper analysis of differentially regulated proteins revealed regulation of proteins involved in actin binding, stress fiber organization, cell membrane organization and actin-based cell projections (**Figure 4b**). Additionally, proteins involved in extracellular matrix organization and metabolic processes were altered in *Kcnk2*^{-/-} endothelial cells. We also highlighted proteins that are regulated by Ca²⁺, Na²⁺ or K⁺ or that are ion channels themselves and added a corresponding paragraph to the discussion section (p 16, l 381-386). Many of them are linked to cell membrane organization or the regulation of actin-based cell projections.

Other than on transcriptome level, the total amount of Cfl1 was not altered. However, analysis of levels of phosphorylated and dephosphorylated Cfl1 by immunofluorescence and Western

blot showed a shift towards dephosphorylated Cfl1 in *Kcnk2*^{-/-} MBMECs. As discussed in our statement to the editor, we aimed for performing phospho-proteomics in MBMECs from naïve and EAE mice, but we were not able to acquire enough protein. We pooled three mice for each replicate in the proteome analysis to receive 5 µg of protein. In our setup, a minimum of 25 µg protein would have been required to generate robust data on protein phosphorylation. Overall, combining analyses of transcriptome, proteome and Cfl1 phosphorylation by immunofluorescence and Western blot analyses supported a clear link between K₂P2.1 and the actin cytoskeleton in brain microvascular endothelial cells and the critical role of actin depolymerization factors in this interaction.

5. Cofilin-1 was identified as the main cytoskeletal regulator involved here. The authors should provide the immunofluorescence images based on which the Cfl1/P-Cfl1 quantifications have been done (Figure 3c,d).

We agree with the reviewer and added representative images to **Figure 4c** for the quantifications of Cfl1 and pCfl1.

6. Moreover, the involvement of the different components of ARP2/3 complex is shown here by measurement of their mRNA levels of expression. As ARP2/3 activity is regulated by phosphorylation, to better address ARP2/3 participation, the authors should investigate the phosphorylation status and activity of the complex in the different experimental conditions.

Unfortunately, we were not able to address this point. We had limitations regarding mouse numbers. For detailed analysis of phosphorylation status, a minimum protein amount of 25 µg per sample would have been required. We were only able to isolate 5 µg per sample (3 pooled) mice. To acquire the protein amount of 25 µg, we would have had to pool 15 mice (for one replicate). This number of mice wouldn't be manageable due to our animal housing and the animal welfare law, especially under Covid19-related restrictions.

8. The authors used a Proximity Ligation Assay to assess whether Cofilin1 and PI(4,5)P₂ interact in presence of in absence of *Kcnk2*. Can the authors provide the reverse experiments in which they look at TREK1 colocalization with PIP₂?

We thank the reviewer for this excellent idea. Colocalization of K₂P2.1 and PI(4,5)P₂ was identified by proximity ligation assay (PLA) as suggested by the reviewer. We inflamed WT MBMECs for 3 h, 6 h, 24 h and investigated also naïve WT and *Kcnk2*^{-/-} MBMECs as negative controls concerning the interaction of K₂P2.1 and PI(4,5)P₂ (**Figure 5f, g**). The interaction of K₂P2.1 and PI(4,5)P₂ decreases during time of inflammation. Our hypothesis for sterically hindrance of Cfl1 binding to PI(4,5)P₂ is strengthened when considering the interaction of Cfl1 and PI(4,5)P₂ as shown in **Figure 5d, e**, which is also happening in a time-dependent fashion and already increased in untreated *Kcnk2*^{-/-} MBMECs. Of note, a recent publication from Panasawatwong et al. supports our findings and the interaction of K₂P2.1 and PI(4,5)P₂ in the context of the conformational states of K₂P2.1³². (see p 11, l 255-257; p 15, l 343-347, p 16, l 373-381)

9. Authors conclude that K2P2.1 has a direct impact on this interaction. What could be this “impact” at a molecular level?

As discussed and shown in the scheme in **Figure 8 a-c**, K₂P2.1 interacts with PI(4,5)P₂ to get into its mechanosensitive state^{32,33}. The attachment of the C-terminal domain of the channel may prevent association of Cfl1 with PI(4,5)P₂³⁴. At a molecular level, K₂P2.1 could sterically prevent this interaction of Cfl1 and PI(4,5)P₂. Strong evidence for this hypothesis is provided by our interaction studies of both proteins, K₂P2.1 and Cfl1, with PI(4,5)P₂ as shown in **Figure 5 d-g**.

Clarity and context

The way the text is written is clear and easy to understand for general audience. I only found two minor corrections concerning to grammars:

- ICAM1 is intercellular adhesion molecule and not intracellular (page 3)
- Weather for whether (page 11)

We apologize for the mistake and corrected it accordingly.

Figure 7 is hard to understand and does not show permeation of cells, which is supposed to be the main finding.

We apologize for this shortcoming and added an overview demonstrating the permeation of the cells into the CNS to provide additional insight in the proposed mechanism (**Figure 8d**).

We thank the reviewer for the thorough evaluation and the very constructive feedback, which helped us to significantly improve our manuscript.

References

1. Bittner, S. *et al.* Endothelial TWIK-related potassium channel-1 (TREK1) regulates immune-cell trafficking into the CNS. *Nat. Med.* **19**, 1161–1165 (2013).
2. Gurniak, C. B., Perlas, E. & Witke, W. The actin depolymerizing factor n-cofilin is essential for neural tube morphogenesis and neural crest cell migration. *Dev. Biol.* **278**, 231–241 (2005).
3. Kanellos, G. *et al.* ADF and Cofilin1 Control Actin Stress Fibers, Nuclear Integrity, and Cell Survival. *Cell Rep.* **13**, 1949–1964 (2015).
4. Montesano, R. *et al.* Increased proteolytic activity is responsible for the aberrant morphogenetic behavior of endothelial cells expressing the middle T oncogene. *Cell* **62**, 435–445 (1990).
5. Heurteaux, C. *et al.* TREK-1, a K⁺ channel involved in neuroprotection and general anesthesia. *EMBO J.* **23**, 2684–2695 (2004).
6. Rinné, S. *et al.* A splice variant of the two-pore domain potassium channel TREK-1 with only one pore domain reduces the surface expression of full-length TREK-1 channels. *Pflügers Arch. - Eur. J. Physiol.* **466**, 1559–1570 (2014).
7. Veale, E. L., Rees, K. A., Mathie, A. & Trapp, S. Dominant Negative Effects of a Non-conducting TREK1 Splice Variant Expressed in Brain*. *J. Biol. Chem.* **285**, 29295–29304 (2010).
8. Lengyel, M., Czirják, G., Jacobson, D. A. & Enyedi, P. TRESK and TREK-2 two-pore-domain potassium channel subunits form functional heterodimers in primary somatosensory neurons. *J. Biol. Chem.* **295**, 12408–12425 (2020).
9. Royal, P. *et al.* Migraine-Associated TRESK Mutations Increase Neuronal Excitability through Alternative Translation Initiation and Inhibition of TREK. *Neuron* **101**, 232-245.e6 (2019).
10. Haghayegh Jahromi, N. *et al.* A Novel Cervical Spinal Cord Window Preparation Allows for Two-Photon Imaging of T-Cell Interactions with the Cervical Spinal Cord Microvasculature during Experimental Autoimmune Encephalomyelitis. *Front. Immunol.* **8**, 406 (2017).
11. Niesner, R., Siffrin, V. & Zipp, F. Two-Photon Imaging of Immune Cells in Neural Tissue. *Cold Spring Harb. Protoc.* **2013**, pdb.prot073528 (2013).
12. Schläger, C., Litke, T., Flügel, A. & Odoardi, F. In Vivo Visualization of (Auto)Immune Processes in the Central Nervous System of Rodents. in 117–129 (2014). doi:10.1007/7651_2014_150
13. Paterka, M. *et al.* Gatekeeper role of brain antigen-presenting CD11c + cells in neuroinflammation. *EMBO J.* **35**, 89–101 (2016).
14. Siffrin, V. *et al.* In Vivo Imaging of Partially Reversible Th17 Cell-Induced Neuronal Dysfunction in the Course of Encephalomyelitis. *Immunity* **33**, 424–436 (2010).
15. Larochelle, C. *et al.* Pro-inflammatory T helper 17 directly harms oligodendrocytes in neuroinflammation. *Proc. Natl. Acad. Sci.* **118**, (2021).
16. Wasser, B. *et al.* CNS-localized myeloid cells capture living invading T cells during

- neuroinflammation. *J. Exp. Med.* **217**, (2020).
17. Birkner, K. *et al.* β 1-Integrin- and KV1.3 channel-dependent signaling stimulates glutamate release from Th17 cells. *J. Clin. Invest.* **130**, 715–732 (2020).
 18. Coisne, C., Lyck, R. & Engelhardt, B. Live cell imaging techniques to study T cell trafficking across the blood-brain barrier in vitro and in vivo. *Fluids Barriers CNS* **10**, 7 (2013).
 19. Odoardi, F. *et al.* T cells become licensed in the lung to enter the central nervous system. *Nature* **488**, 675–679 (2012).
 20. Lodygin, D. *et al.* A combination of fluorescent NFAT and H2B sensors uncovers dynamics of T cell activation in real time during CNS autoimmunity. *Nat. Med.* **19**, 784–790 (2013).
 21. Schläger, C. *et al.* Effector T-cell trafficking between the leptomeninges and the cerebrospinal fluid. *Nature* **530**, 349–353 (2016).
 22. Flach, A.-C. *et al.* Autoantibody-boosted T-cell reactivation in the target organ triggers manifestation of autoimmune CNS disease. *Proc. Natl. Acad. Sci.* **113**, 3323–3328 (2016).
 23. Lodygin, D. *et al.* β -Synuclein-reactive T cells induce autoimmune CNS grey matter degeneration. *Nature* **566**, 503–508 (2019).
 24. Vajkoczy, P., Laschinger, M. & Engelhardt, B. α 4-integrin-VCAM-1 binding mediates G protein-independent capture of encephalitogenic T cell blasts to CNS white matter microvessels. *J. Clin. Invest.* **108**, 557–565 (2001).
 25. Lee, H.-G. *et al.* Pathogenic function of bystander-activated memory-like CD4⁺ T cells in autoimmune encephalomyelitis. *Nat. Commun.* **10**, 709 (2019).
 26. Fels, J., Jeggle, P., Liashkovich, I., Peters, W. & Oberleithner, H. Nanomechanics of vascular endothelium. *Cell Tissue Res.* **355**, 727–737 (2014).
 27. Fels, J. & Kusche-Vihrog, K. Endothelial Nanomechanics in the Context of Endothelial (Dys)function and Inflammation. *Antioxidants Redox Signal.* **30**, 945–959 (2019).
 28. Pollard, T. D. & Cooper, J. A. Actin, a central player in cell shape and movement. *Science* **326**, 1208–1212 (2009).
 29. Katoh, K., Kano, Y. & Ookawara, S. Role of stress fibers and focal adhesions as a mediator for mechano-signal transduction in endothelial cells in situ. *Vasc. Health Risk Manag.* **4**, 1273–1282 (2008).
 30. Maase, M. *et al.* Combined Raman- and AFM-based detection of biochemical and nanomechanical features of endothelial dysfunction in aorta isolated from ApoE/LDLR^{-/-} mice. *Nanomedicine Nanotechnology, Biol. Med.* **16**, 97–105 (2019).
 31. Barreiro, O. *et al.* Dynamic interaction of VCAM-1 and ICAM-1 with moesin and ezrin in a novel endothelial docking structure for adherent leukocytes. *J. Cell Biol.* **157**, 1233–1245 (2002).
 32. Panasawatwong, A., Pipatpolkai, T. & Tucker, S. J. Transition between conformational states of the TREK-1 K^{2P} channel promoted by interaction with PIP₂. *Biophys. J.* (2022).

doi:10.1016/j.bpj.2022.05.019

33. Chemin, J. *et al.* A phospholipid sensor controls mechanogating of the K⁺ channel TREK-1. *EMBO J.* **24**, 44–53 (2005).
34. Zhao, H., Hakala, M. & Lappalainen, P. ADF/cofilin binds phosphoinositides in a multivalent manner to act as a PIP(2)-density sensor. *Biophys. J.* **98**, 2327–2336 (2010).

Point-to-point-reply

Thank you for your response and further support via e-mail in response to our appeal regarding the abovementioned decision. We would also like to thank the reviewers for their time as well as their reasonable and constructive feedback. Despite our reservations concerning certain remarks by reviewers #1 and #2—comments you noted as the primary basis for rejecting our manuscript—we have nonetheless made every effort to comprehensively address each concern. We have outlined the additional experiments, their added value and conclusions in the attached point-by-point reply. In combination with the positive feedback from reviewers #3-5, we are confident that we were able to address all the remaining concerns

Following the reviewer's concerns and discussions with you, we undertook the considerable challenge of creating a cofilin 1 (Cfl1) knock-out in primary mouse brain microvascular endothelial cells (MBMECs) via CRISPR/Cas9 gene editing. Given the lack of established protocols, we anticipated substantial technical difficulties. Nevertheless, we successfully cloned the CRISPR/Cas9 construct, which included a GFP fluorescent detection marker. Despite successfully cloning the CRISPR/Cas9 construct, which included a GFP marker, and extensive optimization of electroporation, lipofectamine, and calcium phosphate transfection methods, we were unable to produce stable Cfl1-knock-out MBMECs. Transfection efficiency remained insufficient, as evidenced by weak GFP signals, and the few cells showing low GFP levels displayed replication deficits. Although multiple strategies were tested, it ultimately proved impossible to establish a viable MBMEC Cfl1 knock-out using CRISPR/Cas9.

To explore alternative approaches, we conducted an extensive literature search and identified a study by Kanellos et al. (2015, PMID: 26655907), which demonstrated successful Cfl1 silencing using siRNA in primary isolated murine keratinocytes and squamous cell carcinoma cells. However, no published protocols existed for Cfl1 knock-down in primary brain endothelial cells such as MBMECs. We ventured to transfer and optimize the Kanellos et al. protocol to our mouse endothelial cell background. Using validated siRNAs from Horizon Discovery, we optimized transfection conditions and timing, and achieved effective and reproducible Cfl1 silencing in MBMECs.

The labor- and resource intensive establishment of this siRNA-based knock-down protocol for primary MBMECs represents a major methodological advancement. Our optimized protocol preserves MBMEC identity by avoiding de-differentiation due to long culture times, while generating sufficient cell numbers for adhesion experiments, immunofluorescent staining, and knock-down validation via quantitative RT-PCR. By successfully standardizing the investigated cell type across all *in-vitro* experiments, we provide a robust foundation for elucidating the role of K_{2p}2.1 and cofilin1 as a downstream effector in brain endothelial cell morphology and function through its impact on actin dynamics.

We would like to emphasize the substantial effort and resources required to establish this novel approach, which now provides a valuable framework for future studies on primary brain endothelial cells (see Figure 1 for the editor).

Figure 1 for the editor: Cfl1 knock-down protocol in primary MBMECs. Experimental approaches for knock-down of Cfl1 in MBMECs by (1) generating a CRISPR/Cas9 plasmid, followed by different transfection protocols and (2) siRNA knock-down.

To proof the cell-specific deletion of *kcnk2* in endothelial cells in *Kcnk2^{fl/fl}Tie2^{Cre+}* animals - addressing the concerns of reviewer #2 - we performed a BaseScope experiment, a highly specialized RNA in situ hybridization assay, to investigate *kcnk2* expression in brain endothelial cells of *Kcnk2^{fl/fl}Tie2^{Cre+}* and *Kcnk2^{fl/fl}Tie2^{Cre-}* mice. Spatial context and single cell resolution revealed highly reduced signal in CD31⁺ endothelial cells from Tie2^{Cre+} animals, while signal was preserved in non-endothelial cells.

In parallel, we stained brain slices from naïve and EAE *Kcnk2^{fl/fl}Tie2^{Cre+}* and *Kcnk2^{fl/fl}Tie2^{Cre-}* mice to assess the impact of endothelial cell-specific K_{2p}2.1 knock-out on adhesion molecules and cytoskeletal proteins. ICAM1, a well-established adhesion molecule, exhibited a strongly upregulated expression in EAE mice compared to naïve controls, with an additional effect of the K_{2p}2.1 knock-out (see **Figure 2a for the editor**). To analyze cytoskeletal regulation, we stained for Arp2/3 and Cfl1. Arp2/3 expression remained unchanged in both EAE and K_{2p}2.1 TREK1 knockout conditions, in line with long term in-vitro inflammation (see Figure S3). Cfl1 was upregulated in *Kcnk2^{fl/fl}Tie2^{Cre+}* animals. Notably, this increase appears to be driven specifically by the endothelial TREK1 knock-out, as evidenced by the comparison to *Kcnk2^{fl/fl}Tie2^{Cre-}* EAE mice (see **Figure 2b for the editor**).

Figure 2 for the editor: Impact of endothelial-specific K_{2p}2.1 KO on adhesion molecule and cytoskeletal protein expression. CD31 staining (green) marks endothelial cells in brain slices from *Kcnk2^{fl/fl}Tie2^{Cre+}* and *Kcnk2^{fl/fl}Tie2^{Cre-}* mice. **(a)** Representative staining of ICAM1 expression (red) in microvessels under naïve and inflammatory (EAE) conditions. ICAM1 expression is upregulated in EAE, with a trend toward a stronger increase in *Kcnk2^{fl/fl}Tie2^{Cre+}* mice compared to *Kcnk2^{fl/fl}Tie2^{Cre-}* siblings. **(b)** Representative stainings of the cytoskeletal proteins Arp2/3 and Cfl1 in microvessels under naïve and inflammatory (EAE) conditions. Arp2/3 expression (magenta) remains unchanged under both inflammatory conditions and in *Kcnk2^{fl/fl}Tie2^{Cre+}* mice, indicating no effect of endothelial K_{2p}2.1 knock-out. In contrast, Cfl1 staining (red) shows a pronounced increase in *Kcnk2^{fl/fl}Tie2^{Cre+}* mice, specifically under inflammatory conditions, compared to *Kcnk2^{fl/fl}Tie2^{Cre-}* siblings.

The comments regarding our statistical analysis, questioned by reviewer #2, have been addressed by a detailed list of the biological and technical replicates in the attached point-to-point reply.

As part of our appeal, we have also reorganized the experiments. In our view, the revised manuscript now more clearly underscores the core finding that K2P2.1 influences brain endothelial cell morphology and function by remodeling the actin network.

Please find attached a detailed **point-by-point reply**.

We sincerely hope you will take our arguments into consideration and remain optimistic about a positive decision on our manuscript. We have devoted substantial effort to this project and are confident that both its scope and methods align with the profile of *Nature Communications*. Our work

delivers robust, cutting-edge research with the potential to captivate a broad readership and catalyze significant advancements in subsequent studies.

Sincerely,

Prof. Tobias Ruck, Dr. Stefanie Lichtenberg, Dr. Laura Vinnenberg and Prof. Sven Meuth

Point-by-point reply:

Reviewers' comments:

Reviewer #1 (Remarks to the Author):

The revised manuscript # NCOMMS-21-01217A addresses some of the concerns raised by several reviewers. The authors have done a tremendous job to elucidate some of the cell biological mechanisms that link K2p 2.1 potassium channel to the actin cytoskeleton in endothelial cells that plays an important role in membrane ruffling and recruitment of immune cells.

Despite these major efforts, there are still several concerns for the revised study.

We thank the reviewer for his/her important and constructive feedback, which significantly improved our manuscript.

1. The major concern is that the study still remains largely an *in vitro* study where the authors have used bEND3.0 cells, an endothelioma cell line that does not resemble at all brain endothelial cells. Therefore, the relevance to the *in vivo* remains limited. The *in vivo* validation of these targets to show that they are upregulated under inflammation (EAE) or in *Kcnk2*^{-/-} cells in mice is not performed. The validation of the screen between the wild-type and mutant cells *in vivo* is critical since the MBMECs are very different to *in vivo* brain endothelial cells. Therefore, the study remains limited in scope.

Please see answer for concern #2, where we respond to both points of criticism.

2. In Figure 6, the authors perform *in vitro* studies of T cell interactions with the wild-type, *Kcnk2*^{-/-} cells, *Kcn2*^{-/-} *Cfl1*^{+/-} cells and *Csfl1*^{+/-} bEnd3.0 cells. Why was knock-down performed in bEnd3.0 cells and not primary brain endothelial cells? As mentioned above bEND3.0 cells are an endothelioma cell line that does not resemble at all brain endothelial cells. Therefore, the value of this experiments is very limited.

Thank you once again for dedicating your time to reviewing our manuscript and for providing constructive and valuable feedback. We are, however, somewhat puzzled by the rationale behind this concern. We employed bEnd.3 cells to knock down cofilin 1, as suggested by reviewer #1 during the first round of revisions. As noted in our previous point-by-point response, *Cfl1*^{-/-} mice and double *Cfl1*^{-/-}/*Kcnk2*^{-/-} mice are not viable, making it unfortunately impossible to generate a mouse line for *in vivo* experiments. Nonetheless, to address these concerns, we have successfully developed a protocol for effective *Cfl1* knock-down via siRNA in MBMECs (primary mouse brain microvascular endothelial cells). We believe that this a major improvement of our manuscript.

We took on the significant challenge of establishing a cofilin 1 (*Cfl1*) knock-out in primary mouse brain microvascular endothelial cells (MBMECs) using CRISPR/Cas9 gene editing. Given the lack of established protocols in the literature, we were aware of the technical difficulties associated with this approach. Nevertheless, we successfully cloned the CRISPR/Cas9 construct, which included a GFP fluorescent detection marker. However, despite extensive optimization using electroporation, lipofectamine and calcium phosphate transfection methods, we were unable to generate stable *Cfl1*-knock-out MBMECs. The transfection efficiency remained insufficient, as indicated by weak GFP signals, and the few cells with low GFP expression exhibited replication deficits. Despite testing

multiple strategies, we were ultimately unable to achieve a viable MBMEC Cfl1 knock-out using the proposed CRISPR/Cas9 strategy.

To explore alternative approaches, we conducted an extensive literature search and identified a study by Kanellos et al. (2015, PMID: 26655907), which demonstrated successful Cfl1 silencing using siRNA in primary isolated murine keratinocytes and squamous cell carcinoma cells. However, no published protocols existed for Cfl1 knock-down in primary brain endothelial cells such as MBMECs. We ventured to transfer and optimize the Kanellos et al. protocol to our mouse endothelial cell background. Using validated siRNAs from Horizon Discovery, we optimized transfection conditions and timing, achieving effective and reproducible Cfl1 silencing in MBMECs.

The labor- and resource intensive establishment of this siRNA-based knock-down protocol for primary MBMECs represents a major methodological advancement. Our optimized protocol preserves MBMEC identity by avoiding de-differentiation due to long culture times, while generating sufficient cell numbers for adhesion experiments, immunofluorescent staining, and knock-down validation via quantitative RT-PCR. By successfully standardizing the investigated cell type across all *in-vitro* experiments, we provide a robust foundation for elucidating the role of K_{2P}2.1 in brain endothelial cell morphology and function through its impact on actin dynamics.

We would like to emphasize the substantial effort and resources required to establish this novel approach, which now provides a valuable framework for future studies on primary brain endothelial cells (see **Figure for the reviewer 1**).

Figure for the reviewer 1: Cfl1 knock-down protocol in primary MBMECs. (a) Experimental approaches for knock-down of Cfl1 in MBMECs by (1) generating a CRISPR/Cas9 plasmid, followed by different transfection

protocols and (2) siRNA knock-down. **(b)** Validation of Cfl1 knock-down in MBMECs by using two different siRNA by qRT-PCR. mRNA expression levels of Cfl1 in untreated and inflamed (TNF α /IFN γ for 24 h) WT and Kcnk2 $^{-/-}$ MBMECs. Data are shown as n-fold change, normalized to the respective NTC WT MBMEC control condition. NTC = non-target-control siRNA; Cfl1^{pool} = Cfl1 SMARTpool siRNA; Cfl1^{m4} = mix of four different Cfl1 siRNAs, (N=3). **(c)** T cell adhesion under low flow (0.25 dyn/cm²) to untreated and inflamed (TNF α /IFN γ for 24 h) WT and Kcnk2 $^{-/-}$ MBMECs transfected with different siRNAs. Total number of T cells adhering to the endothelial cells within 30 min of acquisition. NTC = non-target-control siRNA; Cfl1^{pool} = Cfl1 SMARTpool siRNA; Cfl1^{m4} = mix of four different Cfl1 siRNAs; (N=8-10).

Figure 7 of the revised manuscript validates the gene-silencing effect by quantitative RT-PCR expression analysis and immunofluorescence staining. In cell adhesion assays under low physiological flow, an increased number of WT T cells were attached to inflamed MBMECs of Kcnk2 $^{-/-}$ compared to WT mice. The important regulatory role of Cfl1 in this process was confirmed by a decreased amount of adherent T cells on both inflamed MBMEC layers of WT and Kcnk2 $^{-/-}$ mice under siRNA knock-down of Cfl1. We decided to show only one batch of Cfl1 siRNA to maintain the consistency of the manuscript.

MBMECs are commonly used in *in vitro* studies to characterize the physiology, morphology and function of mouse brain microvascular endothelial cells, e.g., in (Wimmer *et al.*, 2019; Samus *et al.*, 2020; Castro Dias *et al.*, 2021; Epping *et al.*, 2022; Marchetti *et al.*, 2022).

Through the generation of endothelial-cell-specific Kcnk2 knock-out mice and by performing 2-photon imaging in EAE, we could carry out an *in vivo* experiment to prove our hypothesis. We used freshly isolated MBMECs from naïve WT and Kcnk2 $^{-/-}$ mice for the proteome analysis (**Figure 4a-c**).

page 9, line 197 ff:

“Proteome profiling reveals changes in actin network proteins and cytoskeletal organization in Kcnk2 $^{-/-}$ MBMECs

To elucidate the underlying molecular mechanisms upon knock-out of K2P2.1, we analyzed the proteome of Kcnk2 $^{-/-}$ and WT MBMECs. In total 273 differentially regulated proteins were detected (Fig. 4A). Network analysis revealed changes in different pathways related to cytoskeletal regulation and actin filament organization, including actin-based membrane projection, but also metabolic processes and organization of the extracellular matrix in Kcnk2 $^{-/-}$ MBMECs (Fig. 4B). Analysis of interactions and networks of those differentially expressed proteins, revealed interconnections to the Rho/ROCK pathway and different actin depolymerizing factors. We also searched for proteins that are already linked to the function of ion channels. Some of the regulated proteins were dependent on ion channel functions, however, there was no clear connection to a specific channel or channel type (Fig. 4C).”

3. Figure 4 - The assessment of phosphorylated versus non-phosphorylated Cfl1 data are not convincing.

We examined Cfl1 phosphorylation through two distinct, independent methods, which, in our view, provide a robust validation of our findings. To make targeted improvements, we respectfully request clarification on which specific aspects of the data are deemed unconvincing. In its current form, the comment lacks constructive detail

Reviewer #2 (Remarks to the Author):

The revised manuscript by Bock et al has improved in response to the Reviewers' comments. A major improvement in the manuscript is in vivo validation in newly generated endothelial-specific knock out mice for *kcnk2*.

We thank the reviewer for this positive feedback and the thoughtful criticism.

Although the generation of the *Kcnk2f/fl* mice is described in the Methods, Figures are not included to demonstrate the targeting strategy, DNA southern blot analysis and immunohistochemical validation of the cell-specific targeting in endothelial cells.

Please see response to concern #1 below.

Another major concern is the overall rigor of the study and methods of statistical analysis used to derive the conclusions. Most panels are described by double N numbers, such as Fig. 2b (N=3, n=27-34), Fig. 4d (N=4, n=8), etc. The statistics section indicates that "N represents the number of MBMEC preparations; n the number of analyzed single cells or independent coverslips. Statistics was calculated on single cell data, if n is indicated in the figure legend or on mean values, if only N is given". The majority of the study is based on statistical analysis on technical replicates or individual cells. No criterion is provided why in some few cases the N number of biological replicates is used, while in the majority of the study statistics are performed with individual cells. Furthermore, the Figure Legends do not include sufficient information for Reviewers to evaluate the rigor and methods of statistical analysis used in the study. The Figure legends need to be rewritten to clearly define the N used for statistical analysis and eliminate the double "N, n". Source Data can be included to report all the collected data. All statistics performed with individual cells and technical replicates, will need to be repeated with biological replicates (either number of mice or number of independent biological experiments). The exact method of statistical analysis including the post-hoc correction test also needs to be included for each panel. Some examples are given below, but statistics need to be repeated and legends need to be corrected throughout the manuscript.

We regret that the differentiation between experimental replicates and individual cells caused confusion. We have revised the manuscript accordingly, providing a more detailed explanation of the numbers and statistical analyses. Notably, this point was not raised by any other reviewer, although it has been classified here as a major concern.

As specified in the materials and methods section regarding statistical analysis, "N" indicates the number of MBMEC preparations, while "n" denotes the number of analyzed single cells or independent coverslips. Statistics were calculated on single cell data, if n is indicated in the figure legend or on mean values, if only N is given. Statistical analysis of normalized data was performed on the respective raw data (RT-qPCR data and cortical stiffness measurements)." (p. 32). "n" does not reflect the number of technical replicates, but the number of technically independent experiments that can be statistically analyzed in an independent manner. "N" represents the number of individual cell preparations or mice. Throughout the manuscript "n" was used for analyses of EAE, flow chamber, microscopy and AFM experiments and "N" for all quantitative expression analyses (e.g. qPCR and proteomics).

We now include detailed information on the methods of statistical analysis used in the study in **Suppl. Table 2**(or see below response to concern #3).

1. The newly generated *Kcnk2f/fl* mice are only described in the text in Supplemental Methods with no data shown. Figures need to be included to fully disclose the data on the generation of the *Kcnk2f/fl* mice including targeting strategy, DNA southern blot analysis, etc. Importantly, immunohistochemistry

data need to be included to characterize the cell-specific depletion of *Kcnk2* in endothelial cells. What is the percentage of endothelial cells that do not express *Kcnk2*? The IHC analysis needs to be performed before and after EAE induction.

We apologize for not including a scheme of the targeting method and the southern blot analysis in the supplementary methods section. The targeting of the *Kcnk2* gene was performed by insertion of LoxP sites by CRISPR/Cas9 as shown in detail in **Figure for reviewer 2**. We determined the correct insertion of the LoxP sites by Southern blot as provided in **Figure for reviewer 3**.

Figure for reviewer 2: Targeting of the exon 4 of mouse *Kcnk2* gene. The intronic and intergenic regions are shown as line, exons are shown as filled boxes with numeration shown above. The arrows above correspond to the LoxP sequences, and arrows below corresponding to restriction endonuclease sites *Bam*HI (B) and *Pst*I (P). The black box (HR) corresponds to the Southern probe (*Kcnk2_temp1*). The expected sizes of restriction DNA fragments are labeled above arrows. (A) Wild type locus. (B) DNA template (*Kcnk2_temp1*) structure. (C) Genomic locus after the homologous recombination.

Figure for reviewer 3: Southern blot analysis of DNA from tail biopsy. **(A)** Southern blot analysis of tail biopsy DNA's isolated from F1 mice offspring and hybridized with the HR probe (*Kcnk2_templ*). Enzymatic digestion *Pst*I (wt allele 3,6 kb, targeted allele 2,1 kb + 1,6 kb), was used to detect the homologous recombination and presence of the first LoxP site in intron 3 of the *kcnk2* gene locus. Animals 291, 293, 295, 315, 327, 334 and 336 contain correctly targeted 5' region of the DNA template. Positions of the size marker (in bp) are shown on the right. **(B)** Southern blot analysis of tail biopsy DNA's isolated from F1 mice using the enzymatic digestion *Bam*HI (wt allele 4,7 kb, targeted allele 2,6 kb + 2,3 kb), to detect the homologous recombination and presence of the second LoxP site in intron 4 of the *Kcnk2* gene locus. Animals 291, 293, 295, 327, 334 and 336 contain correctly targeted 3' region of the DNA template.

We thank the reviewer for the comment regarding the IHC control of the cell-specific targeting of *Kcnk2*. **Fig. S1** of the revised manuscript shows the characterization of *kcnk2* expression in brain endothelial cells of *Kcnk2^{fl/fl}Tie2^{Cre+}* and *Kcnk2^{fl/fl}Tie2^{Cre-}* mice. BaseScope analysis of the exon 4 floxed region of *Kcnk2* demonstrates markedly diminished signal in CD31⁺ endothelial cells from *Tie2^{Cre+}* animals while signal in non-endothelial cells is retained.

2. The breeding strategy of the *Kcnk2^{fl/fl}* mice for EAE experiments need to be included in the Methods. Were littermate controls used for the EAE study?

To avoid or minimize germline deletion of the floxed allele in Cre-lox experiments, *Tie2-Cre* females were mated with floxed males. In addition, the genotyping protocol included the detection of germline deletion or recombination. We apologize for not stating explicitly that we used Cre negative littermate controls in these experiments. This information was included in the Supplementary Methods section.

3. Figure 1. What does the N refer to? Number of mice, biological replicates, etc? What is the method

of statistical analysis used for each panel? In Figure 1f, if statistics have been performed with technical replicates (n=45-49), they need to be repeated with biological replicates.

We thank the reviewer for this and the following comments, which improved clarity and comprehensibility of the manuscripts' statistics. N represents the number of biological replicates as explained in the materials and methods section (revised manuscript p.30, l. 669-670).

In **Figure 4e** (1f in previous version) of the revised manuscript, four independent MBMEC preparations were used to measure the cortical stiffness of single cells. Therefore, values are depicted for single cell measurements and also statistics were calculated on single cells as already published in Peters *et al.*, 2012, Jeggle *et al.*, 2013, Fels *et al.*, 2022. Technical replicates in this assay are represented as six force-distance curves measured on each individual cell. The average of those technical replicates representing the stiffness values for each cell that is depicted in the graph.

We summarize the numbers of replicates and statistical method for each experiment in the following table to clarify this issue. For reasons of transparency, the figure numbers of the revised (1st column) and original (2nd column) manuscript are listed. The details of the now excluded experiments are also listed for the reviewers:

Figure revised manuscript	Figure original manuscript	Biological replicates (N)	Number of individual cells or independent experiments (n)	Technical replicates per experiment	Statistical method	Tested on
1a	7a	scheme				
1b	7b	3 EAEs	2-3 mice		representative images	Individual mice (n)
1c	7c	3 EAEs	2-3 mice		representative images	
1d	7d	3 EAEs	2-3 mice		representative images	
1e	7e	3 EAEs	total: 7 mice; 3 independent EAEs; 5 mice baseline (11 T cell tracks), 7 mice isotype + spadin (96 T cell tracks), 6 mice anti-ICAM1 + spadin (21 T cell tracks)		1way ANOVA, Bonferroni corr.	Individual mice (n)
1f	7h	3 independent EAEs	Total 8-9 mice		2-way ANOVA, Bonferroni corr.	Individual mice (n)
1g	7g	3 independent EAEs	Total 8-9 mice		2-way ANOVA, Bonferroni corr.	Individual mice (n)

2a	1a	4-6 MBMEC preparations		triplicates	Kruskal-Wallis test, Dunn's multiple comparison test	MBMEC preparation (N)
2b	1b	6 MBMEC preparations		duplicates	Kruskal-Wallis test, Dunn's multiple comparison test	MBMEC preparations (N)
2c	6d	scheme				
2d	6e	6-8 MBMEC preparations	2 individual flow chambers, 2 independent T cell preparations		1way ANOVA, Bonferroni corr.	individual flow chambers/ T cell preparations (n)
3a	2d	3 MBMEC preparations	2 coverslips	4 ROIs	representative images	
3b	2e	3 MBMEC preparations	2 coverslips	4 ROIs	representative images	
3c	2a	3 MBMEC preparations	4-8 dishes		representative images	
3d	2b	3 MBMEC preparations	4-8 dishes, 27-34 individual areas		Kruskal-Wallis test, Dunn's multiple comparison test	Individual areas (n)
3e	6h	4 MBMEC preparations	2 dishes, 3-5 individual T cells each	20-50 force-distance curves/ T cell	Kruskal-Wallis test, Dunn's multiple comparison test	Individual T cells (n)
4a	4b	5 MBMEC preparations	4 mice pooled		Student's t-test, Benjamini-Hochberg correction for multiple testing, FDR 0.01	MBMEC preparations (N)
4b	4c	5 MBMEC preparations	4 mice pooled			
		5 MBMEC preparations	4 mice pooled			
4d	1d	3 MBMEC preparations	4-8 dishes (12-22 individual areas)		representative images	
4e	1e	3 MBMEC preparations	4-8 dishes (12-22 individual areas)		1way ANOVA, Bonferroni corr.	Individual areas (n)

4f	1f	4 MBMEC preparations	45-59 individual cells (total)	6 force distance curves per cell	1way ANOVA, Bonferroni corr.	Individual cells (n)
5a	3a	4 MBMEC preparations		quadruples	Significantly differentially regulated genes, 1way ANOVA Bonferroni corr.	MBMEC preparations (N)
5b	3c	4 MBMEC preparations		quadruples	1way ANOVA, Bonferroni corr.	MBMEC preparations (N)
5c	S4a	4 MBMEC preparations		quadruples	1way ANOVA, Bonferroni corr.	MBMEC preparations (N)
5d	S4b	4 MBMEC preparations		quadruples	1way ANOVA, Bonferroni corr.	MBMEC preparations (N)
5e	4c	4 MBMEC preparations	2 coverslips		representative images	
5f	S4c	4 MBMEC preparations	2 coverslips	4 ROIs	1way ANOVA, Bonferroni corr.	Individual coverslips (n)
6a	5a	4 MBMEC preparations	2 coverslips	4 ROIs	representative images	
6b	5b	4 MBMEC preparations	2 coverslips	4 ROIs	1way ANOVA, Bonferroni corr.	Individual coverslips (n)
6c	5c	4 MBMEC preparations	2 coverslips	4 ROIs	1way ANOVA, Bonferroni corr.	Individual coverslips (n)
6d	5d	4 MBMEC preparations		duplicates, 4 ROIs each	representative images	MBMEC preparations (N)
6e	5e	4 MBMEC preparations		duplicates, 4 ROIs each	1way ANOVA, Bonferroni corr.	MBMEC preparations (N)
6f	5f	5 MBMEC preparations		quadruples, 4 ROIs each	representative images	MBMEC preparation (N)s
6g	5g	5 MBMEC preparations		quadruples, 4 ROIs each	1way ANOVA, Bonferroni corr.	MBMEC preparations (N)
7a	New	12 MBMEC preparations	2 coverslips	4 ROIs	representative images	
7b	New	3 MBMEC preparations		quadruples	2way ANOVA, multiple	MBMEC preparations (N)

						comparisons Bonferroni corr.	
7c	New	12 MBMEC preparations / one T cell preparation per MBMEC preparation				2way ANOVA, multiple comparisons Bonferroni corr.	MBMEC preparations (N)
8	8	scheme					
S1a	New	3 independent EAEs	Total mice 5-6			representative images	
S1b	New	3 independent EAEs	Total mice 5-6	2 ROIs		2-way ANOVA, Bonferroni corr.	Individual mice (n)
S2a	S2a	5 MBMEC preparations	2 coverslips	4 ROIs		representative images	
S2b	S2b	5 MBMEC preparations	2 coverslips	4 ROIs		1way ANOVA, Bonferroni corr.	Individual coverslips (n)
S2c	S2c	5 MBMEC preparations	2 coverslips	4 ROIs		1way ANOVA, Bonferroni corr.	Individual coverslips (n)
S3a	3d	4 MBMEC preparations		quadruples		1way ANOVA, Bonferroni corr.	MBMEC preparations (N)
S3b	New	3 independent EAEs	Total mice 5-6			representative images	
S-Tab 1	S-Tab 1	4 MBMEC preparations		quadruples		Significantly differentially regulated genes, 1way ANOVA Bonferroni corr.	MBMEC preparations

4. Figure 2: Have the statistics been performed with N=3 or n=27-34? What was the method of statistical analysis used?

The detailed information is now provided in **Suppl. Table 2** above the original Figure 2b refers to **Figure 3d** of the revised manuscript.

The algorithm and statistics used for the analysis of membrane protrusions has been adapted from (Franz *et al.*, 2016). The analysis of protrusions was performed within the region of interest, represented by an area of 400µm². 27-34 different regions have been imaged and analyzed accordingly. Statistics have been calculated on the different ROIs (n=27-34) using the Kruskal-Wallis test with Dunn's correction for multiple comparisons.

5. Figure 5b, c: “N=4, n=8”. The bar graphs show 8 data points, assuming that n=8 was used for statistical analysis. In contrast, in Figure 5g “N=5, n=20”, the bar graph shows 5 data points. How do the authors choose the N for statistical analysis?

For the analysis of stress fibers in **Figure 6b and c** (Figure 5b and c of the original manuscript), the two coverslips of each MBMEC preparation were treated independently in different culture dishes. The PLA shown in Figure 5g was performed using technical (side-by-side) replicates. The N for statistical analysis has been chosen according to the independent treatment of the replicate. Technical replicates in direct side-by-side assays were used to calculate an average for the respective data point used for analysis. As technical replicates in **Figure 6b and c** (Figure 5b and c of the original manuscript), we used four regions of interest that were imaged and analyzed on the individually treated coverslips.

6. Figure 6: The majority of the Figure includes n=17-19 data points. Are these numbers of individual cells? Are results statistically significant if N=biological replicates and/or does the P value change if statistics are performed with biological replicates?

In **Figure 3e** (Figure 6h of the original manuscript) the individual data points represent individual T cell-MBMEC adhesion forces. As already stated for the AFM-based methods, we use the analyses that are widely accepted and commonly used (Peters *et al.*, 2012; Jeggle *et al.*, 2013; Fels *et al.*, 2022). Each data point has been calculated as an average of 20-50 force-distance curves per T cell, representing the technical replicate. We performed measurements on different MBMEC and T cell preparations.

Statistical significance remains when calculated using single MBMEC preparations:

Bonferroni's multiple comparisons test	Individual cells			Averaged MBMEC preparations		
	Mean Diff,	95,00% CI of diff,	Adjusted P Value	Mean Diff,	95,00% CI of diff,	Adjusted P Value
WT vs. WT infl	-0,4253	-0,5374 to -0,3131	<0,0001	-0,4096	-0,5299 to -0,2893	<0,0001
WT vs. KO	-0,1204	-0,2375 to -0,003256	0,0403	-0,1230	-0,2434 to -0,002692	0,0439
WT vs. KO infl	-0,1450	-0,2656 to -0,02441	0,0094	-0,1624	-0,2827 to -0,04202	0,0067
WT infl vs. KO	0,3049	0,1878 to 0,4220	<0,0001	0,2866	0,1662 to 0,4069	<0,0001
WT infl vs. KO infl	0,2803	0,1597 to 0,4009	<0,0001	0,2472	0,1269 to 0,3676	0,0002
KO vs. KO infl	-0,02463	-0,1498 to 0,1006	>0,9999	-0,03933	-0,1597 to 0,08100	>0,9999

7. Other than the intravital imaging, were there any other experiments or quantification methods blinded in the study? Were the EAE experiments scored in a blinded manner? Were the drugs administered in a blinded manner? Blinding and randomization for the preclinical pharmacology studies need to be described.

We regret any lack of clarity. All experiments were conducted under randomized and blinded conditions —both in data acquisition (including scoring of EAE, drug administration) and analysis—and randomization was e.g. done by changing, arbitrary recording and treatment order of the various groups.

8. Figure 7h “(h) EAE disease course of endothelial cell-specific Kcnk2^{-/-} and control mice with or without spadin treatment (N=8). Data from 7 mice from 3 different EAEs (baseline: n=5, spadin and isotype: n= 7, spadin and anti-ICAM1: n=6)”. What does the N=8 refer to? If N=8 mice, how is this possible from “7 mice from 3 different EAEs”, which should be 7X3=21 mice? Does baseline refer to

the *Kcnk2^{fl/fl}/Tie2^{cre-}* + ctr group? The Figure legend does not correspond to the graph shown in Fig. 7h and the number of the conditional KO mice used in the study is not disclosed.

We apologize for this misleading figure legend. The last sentence corresponds to Figure 1 a-e (Figure 7a-g of the original manuscript). We corrected this accordingly.

9. Which method of statistical analysis was used for the EAE experiment in Fig 7h? P values need to be indicated for every data point that is statistically significant.

In figure 1f (Figure 7h of the original manuscript) we used 2-way ANOVA with Bonferroni's multiple comparison test to analyze the EAE disease course. We added the p values to the table below:

	day 1-14	day 15-16	day 17-24	day 25-35
Kcnk2^{fl/fl}/Tie2^{cre-} + ctr vs. Kcnk2^{fl/fl}/Tie2^{cre-} + spadin	ns	*	ns	*
Kcnk2^{fl/fl}/Tie2^{cre-} + ctr vs. Kcnk2^{fl/fl}/Tie2^{cre+} + ctr	ns	ns	ns	*
Kcnk2^{fl/fl}/Tie2^{cre+} + ctr vs. Kcnk2^{fl/fl}/Tie2^{cre+} + spadin	ns	ns	ns	ns
Kcnk2^{fl/fl}/Tie2^{cre-} + spadin vs. Kcnk2^{fl/fl}/Tie2^{cre+} + spadin	ns	ns	ns	ns

10. Supplementary Figure 2c: The Western blot image is of very poor quality and needs to be replaced with a high quality image. Furthermore, the western blot image is not representative of the quantification shown.

We agree and excluded all western blot results based on this comment.

Reviewer #3 (Remarks to the Author):

The authors have satisfactorily addressed all my comments. This is an interesting and relevant scientific manuscript performed in a careful and meticulous manner. Therefore, I recommend the revised manuscript version for publication in Nature Communications.

We thank the reviewer for this very positive feedback and value the constructive comments and time spent on our manuscript.

Reviewer #4 (Remarks to the Author):

The manuscript NCOMMS-21-01217A by Bock et al., entitled « K2P2.1 shapes morphology and function of brain endothelial 1 cells via actin network remodeling », deepens the conclusions that these authors have made in Bittner et al., Nature Medicine 2013. Blood Brain Barrier (BBB) dysfunction is a common feature in several neurological diseases. Whereas previously the authors demonstrated that TREK1 potassium channel (K2P2.1, encoded by *Kcnk2* gene) is expressed in mouse brain microvascular endothelial cells (MBMECs) and that either the genetic deletion of *Kcnk2* in mice or the pharmacological inhibition of K2P2.1 by Spadin facilitates lymphocyte migration into the CNS by upregulating cellular adhesion molecules ICAM1 and VCAM1, revealing a role for TREK1 in the Blood Brain Barrier (BBB) function, the present paper describes the molecular events leading to it.

Notably the authors described the detailed molecular pathways whereby TREK1 modulates BBB permeability.

The authors made a good work in response to the comments. The new version of the paper is clear and easy to understand making it accessible for general audience, the experiments are well designed and the quality of the results is now really high. Because of the pathologic states induced by dysfunction in BBB, this study is of general interest by characterizing TREK1 function on BBB maintenance. Together the results support TREK1 as a promising drugable target to treat such diseases.

The authors responded well to the concerns rose during the first round of reviewing, notably:

In the first version of their manuscript, the authors made use of different imaging techniques to investigate TREK1 localization and actin cytoskeleton characterization but some images for quantification were missing or not of optimal quality. They now fully completed their data by providing pictures with a better resolution and by adding quantifications. They also improved the clarity of the procedures and reinforced their conclusions by including new controls in their analyses. In addition, they performed new experiments (proteomics, generation and analysis of mutant MBMEC lines, use of conditional KO mice for *Kcnk2*) that further document the role of TREK1 in cytoskeleton regulation.

We sincerely appreciate the considerable time and effort the reviewer has devoted to evaluating our work, and we thank for his/her very positive feedback.

Specific minor comments are listed below :

- Results section, Title 1st paragraph : « down-regulation of *Kcnk2* gene expression... » instead of « modulation of *K2P2.1* function » would be more appropriate.

We amended the manuscript accordingly.

Reviewer #5 (Remarks to the Author):

The manuscript presented by Bock et al., entitled " *K2P2.1* shapes morphology and function of brain endothelial cells via actin network remodeling", describes a study which includes proteomics analyses by data-independent acquisition (DIA) mass spectrometry (MS) using an Orbitrap Exploris 480 instrument and variable size precursor isolation windows. The proteomics experiments and results are well described. Please be sure to release the PRIDE repository with identifier PXD031051, as it is not yet public.

We thank the reviewer for the positive feedback and apologize for this issue. The data will be public as soon as the manuscript will be accepted for publication. We missed to add the accession details for the reviewers. You can access the data at the PRIDE repository as a reviewer with the following login:

Username: reviewer_pxd031051@ebi.ac.uk

Password: h8Zrymf7

References:

Castro Dias, M. et al. (2021) 'Brain endothelial tricellular junctions as novel sites for T cell diapedesis across the blood–brain barrier', *Journal of Cell Science*, 134(8). doi: 10.1242/jcs.253880.

- Dzamukova, M. *et al.* (2022) 'Mechanical forces couple bone matrix mineralization with inhibition of angiogenesis to limit adolescent bone growth', *Nature Communications*, 13(1), p. 3059. doi: 10.1038/s41467-022-30618-8.
- Epping, L. *et al.* (2022) 'Activation of non-classical NMDA receptors by glycine impairs barrier function of brain endothelial cells', *Cellular and Molecular Life Sciences*, 79(9), p. 479. doi: 10.1007/s00018-022-04502-z.
- Fels, B. *et al.* (2022) 'Effects of Chronic Kidney Disease on Nanomechanics of the Endothelial Glycocalyx Are Mediated by the Mineralocorticoid Receptor', *International Journal of Molecular Sciences*, 23(18), p. 10659. doi: 10.3390/ijms231810659.
- Franz, J. *et al.* (2016) 'Nanoscale imaging reveals a tetraspanin-CD9 coordinated elevation of endothelial ICAM-1 clusters', *PLoS ONE*, 11(1). doi: 10.1371/journal.pone.0146598.
- Greene, C. *et al.* (2022) 'Microvascular stabilization via blood-brain barrier regulation prevents seizure activity', *Nature Communications*, 13(1), p. 2003. doi: 10.1038/s41467-022-29657-y.
- Jeggle, P. *et al.* (2013) 'Epithelial sodium channel stiffens the vascular endothelium in vitro and in liddle mice', *Hypertension*, 61(5), pp. 1053–1059. doi: 10.1161/HYPERTENSIONAHA.111.199455.
- Marchetti, L. *et al.* (2022) 'ACKR1 favors transcellular over paracellular T-cell diapedesis across the blood-brain barrier in neuroinflammation in vitro', *European Journal of Immunology*, 52(1), pp. 161–177. doi: 10.1002/eji.202149238.
- Peters, W. *et al.* (2012) 'Nanomechanics and sodium permeability of endothelial surface layer modulated by hawthorn extract WS 1442', *PLoS ONE*, 7(1). doi: 10.1371/journal.pone.0029972.
- Samus, M. *et al.* (2020) 'Actin-Binding Protein Cortactin Promotes Pathogenesis of Experimental Autoimmune Encephalomyelitis by Supporting Leukocyte Infiltration into the Central Nervous System', *The Journal of Neuroscience*, 40(7), pp. 1389–1404. doi: 10.1523/JNEUROSCI.1266-19.2019.
- Wimmer, I. *et al.* (2019) 'PECAM-1 stabilizes blood-brain barrier integrity and favors paracellular T-cell diapedesis across the blood-brain barrier during neuroinflammation', *Frontiers in Immunology*, 10(APR). doi: 10.3389/fimmu.2019.00711.

Point by point reply

Main Reviewer Comment (Author Checklist):

For the two special cases (Figures 3e and 4f) with MBMECs and T cells, we ask that you please specify how many of each were used for each experiment (the "N" would be two numbers) in the Main Text and in your statistics Table S2. We also ask that you report the results of statistical tests performed with the data split by MBMEC. That is, please add new statistical tests on the MBMEC numbers (because the MBMEC batches might have a systematic bias, e.g. they may be particularly adhesive), so that the readers can have both analyses.

We thank the reviewer for this important and constructive comment regarding the statistical analysis of Figures 3e and 4f. In response, we have carefully revised the figure legend, Supplementary Table S2 and our Source Data File to clearly indicate the number of MBMEC preparations ("N") and the number of individual T cells or endothelial cells ("n") used per condition. These specifications are now consistently reported to ensure full transparency and facilitate reproducibility.

Within the field's standards for single-cell AFM-based force spectroscopy, individual cells represent the primary biological resolution ^(1,2). Nevertheless, in accordance with the reviewer's request, we have performed an additional statistical analysis using averaged values per MBMEC preparation to assess potential batch effects. The results of this new analysis were consistent with our original findings, supporting the conclusion that the observed effects are robust and not attributable to systematic differences between MBMEC batches. To ensure transparency and clarity, we now provide both layers of analysis in the Source Data File:

- **Statistics based on individual cells (n)** – reflecting the nature of the single-cell measurements obtained via AFM-based force spectroscopy.
- **Statistics based on MBMEC preparations (N)** – using averaged values from individual batches to control for potential batch-specific effects.

Moreover, the comparison of standard deviations between individual cells and MBMEC preparation means revealed higher variability among cells than between preparations (see Figure for the reviewer 1). This supports the interpretation that no systematic batch effect was present.

Figure for the reviewer 1: Comparison of standard deviations in single cell spectroscopy measurements (CellHesion, 3e, and Cortical stiffness, 4f) calculated on single cell level (n) and the MBMEC preparations (N).

We hope these clarifications and additional analyses fully address the reviewer's concerns and further enhance the transparency and rigor of our manuscript.

Reviewer #1 (Remarks to the Author)

I appreciate the efforts that the authors have put in the revised manuscript (# NCOMMS-21-01217B-Z) to address the concerns raised by the reviewers.

1. It is great to see that the authors have addressed the concern from Reviewer # 1 and have used the siRNA knockdown of Cfl1 in primary mouse brain endothelial cells for their experiments in Figure 7 which have strengthened the conclusions of the study since they have moved away from the bEnd3.0 endothelioma cell line.
2. In addition, the authors have done an excellent job validating some of the actin regulatory factors in control and EAE WT and Kcn2fl/fl Tie2-Cre mice.

We greatly appreciate the positive feedback.

3. The concern raised by Reviewer # 2 about the "meaning" of N versus n in many figure legends still remains unresolved. The authors should make the effort to include a couple of additional sentences in each figure legends to explain what N or n means rather than making the reader guess. In addition the statistical method used for the analysis should be put in the respective figure legends. This will improve the understanding of the data described in the respective figures.

We appreciate the emphasis on clarifying the distinction between biological and technical replicates. In the revised version of our manuscript, we have now consistently used **N** to denote biological replicates (individual MBMEC preparations) and used **n** for reflecting the nature of the single-cell measurements obtained via AFM-based force spectroscopy (Fig. 3e and 4f). We added the information regarding the entity of the biological replicate to the figure legends. Furthermore, we have included the specific statistical methods used in each experiment directly within the respective figure legends. Further experimental details, including repetitions and technical replicates are stated in the methods and summarized in Table S2.

4. Statistics concerns raised by Reviewer # 2 still remains an issue. I feel that the statistics are overinflated if they are performed on the n, rather than N (biological replicates). This seems to be the case for Figs 3d, 4e, 4f, 5f, 6b, 6c, 6e, S2b, S2c. The authors should perform the statistical analysis on biological, not technical, replicates. The technical replicates measure the same sample multiple times, assessing experimental variability, while biological replicates measure different samples, capturing biological variation which is used for statistical analysis.

- **Correction of Figure 1e:**

Thank you for pointing out the discrepancy in Figure 1e. Upon review, we realized that a preliminary version of the graph was mistakenly included in the previous submission. We have now corrected this by including the final, updated version of Figure 1e. Specifically, this experiment was conducted using in total seven mice across three independent EAE experiments. Due to randomization of the experimental groups, we were able to analyze:

- N = 5 mice for the baseline condition,
- N = 7 mice for the isotype + spadin condition,
- N = 6 mice for the anti-ICAM1 + spadin condition.

These numbers are now accurately reflected in both the updated figure and revised Table S2.

- **Revised Figures and Statistical Reanalysis:**

In response to Reviewer #1's concerns regarding Figures 3d, 4e, 4f, 5f, 6b, 6c, 6e, S2b, and S2c, we have thoroughly revised the statistical analysis. For Figures 3d, 4e, 5f, 6b, 6c, 6e, and S2b/c, we now present results based on biological replicates. Specifically, we calculated the mean of each independent MBMEC preparation. To address the reviewer's concern about the overinflation of the statistical analysis, we re-calculated the statistics based on N. These updates are clearly documented in the revised Table S2.

- **Special Cases – Figures 3e and 4f:**

The analysis in Figures 3e and 4f differs due to the specialized methods used. The CellHesion and cortical stiffness measurements by atomic force microscopy are widely accepted methods for single cell analysis. Moreover, the comparison of standard deviations between individual cells and MBMEC preparation means revealed higher variability among cells than between preparations (see Figure for the reviewer 1). This supports the interpretation that no systematic batch effect was present.

Reviewer #3 (Remarks to the Author):

The revised manuscript addresses the concerns raised by several reviewers. The amount of data is impressive, experiments are well performed and controlled, imaging data is of good quality, and results convincingly support the main authors conclusions. Issues raised by Reviewer #2 regarding proper statistics and reproducibility has been satisfactorily addressed. The authors provided a comprehensive table detailing the manuscript statistics used in every figure. Altogether, the results presented in this study are of high-quality and convincing, thus I recommend acceptance for publication of this revised manuscript version.

We sincerely thank the reviewer for the positive and encouraging feedback. We are grateful for the acknowledgment of the quality and reproducibility of our data, as well as the improvements made in response to the previous comments. We appreciate the reviewer's recommendation for acceptance.

Reviewer #4 (Remarks to the Author):

Regarding the number of replicates and statistical analyses: The authors have provided a detailed response to the statistical concerns, with appropriate clarification regarding the distinction between biological (N) and technical (n) replicates. They state that statistical analyses have been recalculated based solely on biological replicates, which aligns with standard practice for ensuring valid inference. Sample sizes are explicitly reported, and the statistical methods used are now documented in the manuscript and supplementary materials. In experiments involving single-cell measurements, the authors justify their approach by referencing established literature employing similar methodologies. While this aligns with common practices in the field, it is worth noting that in some cases, a large number of measurements were obtained from a relatively small number of biological preparations. This raises potential concerns about pseudoreplication, particularly if statistical independence was assumed at the level of individual cells rather than the biological preparation. Although the authors acknowledge this issue and cite relevant precedent, it would be helpful to provide further clarification on how the statistical analyses accounted for this nested data structure. That said, their engagement with this concern reflects a thoughtful and responsible approach to data interpretation.

Additionally, in some experiments where the number of biological replicates is low (e.g., N=4–7), a brief discussion on the statistical power to detect meaningful effects may help contextualize the reliability of the findings.

Overall, the revisions thoughtfully address the main statistical concerns. While a few minor uncertainties remain, especially regarding the treatment of nested data and the robustness of inferences drawn from small sample sizes, the statistical reasoning is generally sound. The response is well considered and, with a few additional clarifications, could be further strengthened. To conclude, the authors have clearly made thoughtful efforts that have significantly enhanced the rigor of their work.

We are very pleased that the reviewer found the previous revisions satisfactory and that they contributed to improving the overall quality of the manuscript. Some minor statistical questions remained unanswered, which we are happy to address: we agree that the relatively low number of biological replicates in some experiments (e.g., N=4–7) may limit the statistical power to detect smaller effect sizes. However, the observed differences in these experiments were consistent across replicates and reached statistical significance, supporting the robustness of the findings. Together with the detailed explanation and evaluation of the single-cell measurements included in the revised version, we are confident that the revisions convincingly support the main conclusions of our study.

References:

1. Fels, J. et al. (2025). SECS, drugs, and Rac1&Rho: regulation of EnNaC in vascular endothelial cells. *Pflügers Archiv - European Journal of Physiology*, 477(7), 977–992. <https://doi.org/10.1007/s00424-025-03093-5>
2. Vahldieck, M. et al. (2023). Endothelial Glycocalyx and Cardiomyocyte Damage Is Prevented by Recombinant Syndecan-1 in Acute Myocardial Infarction. *The American Journal of Pathology*, 193(4), 474–492. <https://doi.org/10.1016/j.ajpath.2022.12.009>